

# Challenges associated with the climatic interpretation of water stable isotope records from a highly resolved firn core from Adélie Land, coastal Antarctica

5 Sentia Goursaud[1,2], Valérie Masson-Delmotte[1], Vincent Favier[2,3], Suzanne Preunkert[2,3], Michel Legrand[2,3], Bénédicte Minster[1], Martin Werner[4]

[1]LSCE (UMR CEA-CNRS-UVSQ 8212-IPSL), Gif-sur-Yvette, France

10 [2]Univ. Grenoble Alpes, Laboratoire de Glaciologie et Géophysique de l'Environnement (LGGE), 38041 Grenoble, France

[3]CNRS, Laboratoire de Glaciologie et Géophysique de l'Environnement (LGGE), 38041 Grenoble, France

[4] Alfred Wegener Institute, Helmholtz Centre for Polar and Marine Research, Bremerhaven, Germany

*Correspondence to*: Sentia Goursaud (sentia.goursaud@lsce.ipsl.fr)



## Abstract.

A new 21.3 m firn core was drilled in 2015 at a coastal Antarctic high accumulation site in Adélie Land (66.78 °S; 139.56 °E, 602 m a.s.l.). The core was dated by annual layers counting based on non-sea-salt sulfate and methanesulfonate summer peaks, refined by a comparison between the reconstructed surface mass balance (hereafter, SMB) and the closest available stake data. The mean reconstructed SMB of 75.2 ± 15.0 cm w.e. $y^{-1}$ is consistent with local stake data, and remarkably high for coastal East Antarctica. The resulting inter-annual and sub-annual variations in isotopic records ($\delta^{18}O$ and deuterium excess, hereafter d-excess) are explored for 1998 – 2014 and are systematically compared with a couple of climatic time series: an updated database of Antarctic surface snow isotopic composition, SMB stake data, meteorological observations from Dumont d'Urville station, sea-ice concentration based on passive microwave satellite data, precipitation outputs of atmospheric reanalyses, climate and water stable isotope outputs from the atmospheric general circulation model ECHAM5-wiso, as well as air mass origins diagnosed using 5-days back-trajectories.

The mean isotopic values (-19.3 ± 3.1 ‰ for $\delta^{18}O$ and 5.4 ± 2.2 ‰ for d-excess) are consistent with other coastal Antarctic values. No significant isotope- temperature relationship can be evidenced at any timescale, ruling out a simple interpretation of in terms of local temperature. An observed asymmetry in the $\delta^{18}O$ seasonal cycle may be explained by the precipitation of air masses coming from Indian and Pacific/West Antarctic Ice Sheet sectors in autumn and winter times, recorded in the d-excess signal showing outstanding values in austral spring versus autumn. Significant positive trends are observed in the annual d-excess record and local sea-ice extent (135 °E-145 °E) over the period 1998-2014.

However, processes studies focusing on resulting isotopic compositions and particularly the d-excess-$\delta^{18}O$ relationship, evidenced as a potential fingerprint of moisture origins, as well as the collection of more isotopic measurements in Adélie Land are needed for an accurate interpretation of our signals.



## 1. Introduction

### *Motivation for new coastal Antarctic firn cores*

Polar ice cores are exceptional archives of past climate variations. In Antarctica, many deep ice cores have been drilled and analyzed since the 1950s. For instance, Stenni et al. (2017b) compiled water stable isotope data from 112 ice cores spanning at least part of the last 2000 years. Most deep ice cores were drilled in the central Antarctic plateau where low accumulation rates and ice thinning give access to long climate records. In today's context of rapid global climate change, it is of paramount importance to also document recent past climate variability around Antarctica. Many Antarctic regions still remain undocumented due to the lack of accumulation and water stable isotope records from shallow ice cores or pits (Masson-Delmotte et al., 2008; Jones et al., 2016). An accurate knowledge of changes in coastal Antarctic surface mass balance (hereafter, SMB), an evaluation of the ability of climate models to resolve the key processes affecting its variability, and thus an improved confidence in projections of future changes in coastal Antarctic surface mass balance is important to reduce uncertainties on the ice sheet mass balance and its contribution to sea level change (Church et al., 2013).

Meteorological observations have been conducted since 1957 in manned and automatic stations (Nicolas and Bromwich, 2014), and considerable efforts have been deployed to compile and update the corresponding dataset (Turner et al., 2004). This network is marked by gaps in spatio-temporal coverage (see Goursaud et al., 2017a) as well as systematic biases of instruments such as thermistors (Genthon et al., 2011). Satellite remote sensing data have been available since 1979 and provide large-scale information for changes in Antarctic sea ice and temperature (Comiso et al., 2017), but do not provide sufficient accuracy and homogeneity to resolve trends at local scales (Bouchard et al., 2010). Coastal shallow (20 – 50 m long) firn cores are thus essential to provide continuous local to regional climate information spanning the last decades at sub-annual resolution. They complement stake area observations of spatio-temporal variability in surface mass balance (hereafter, SMB; Favier et al., 2013), which also help assessing the representativeness of a single record.

Since the 1990s, efforts have been made to retrieve shallow ice cores in coastal Antarctic areas (e.g. Mayewski et al., 2005). Most of these efforts have been focused on the Atlantic sector, in Dronning Maud Land (e.g. Isaksson and Karlén, 1994; Graf et al., 2002; Altnau et al., 2015) and the Weddell Sea Sector




(Mulvaney et al., 2002). Fewer annually resolved water stable isotope records have been obtained from ice cores in other regions, such as the Ross Sea sector (Bertler et al., 2011), Law Dome (Morgan et al., 1997; Delmotte et al., 2000; Masson-Delmotte et al., 2003), Adélie Land (Yao et al., 1990; Ciais et al., 1995; Goursaud et al., 2017b), and Princess Elizabeth region (Ekaykin et al., 2017). New coastal drilling

efforts have recently been triggered in the context of the ASUMA project (improving the Accuracy of SUrface Mass balance of Antarctica) from the French Agence Nationale de la Recherche, which aims to assess spatio-temporal variability and change in SMB over the transition zone from coastal Adélie Land to the central East Antarctic Plateau (towards Dome C).

*Climatic interpretation of water stable isotope records*

Water stable isotope ($\delta^{18}$O, $\delta$D) records from central Antarctic ice cores have classically been used to infer past temperature changes (e.g. Jouzel et al., 1987). The isotope-temperature relationship was nevertheless shown not to be stationary and to vary in space (Jouzel et al., 1997), calling for site-specific calibrations relevant for various timescales (Stenni et al., 2017b). In coastal regions, several studies

showed no temporal isotope-temperature relationship at all between water stable isotope records in firn cores covering the last decades and near-surface air temperature measured at the closest station. This is for instance the case in Dronning Maud Land, near the Neumayer station (three firn cores, for which the longest covered period is 1958-2012, Vega et al., 2016), in the Ross Sea sector (one snow pit covering the period 1964-2000, Bertler et al., 2011), and in Adélie Land, close to DDU (one firn core covering the

period 1946-2006, Goursaud et al., 2017b). While several three-dimensional atmospheric modelling studies have suggested a dominant role of large-scale atmospheric circulation on the variability of coastal Antarctic snow $\delta^{18}$O (e.g. Noone and Simmonds, 2002; Noone, 2008), understanding the drivers of coastal Antarctic $\delta^{18}$O variability remains challenging (e.g. Schlosser et al., 2004; Dittmann et al., 2016; Bertler et al., 2018; Fernandoy et al., 2018). While distillation processes are expected theoretically to relate

condensation temperature with precipitation isotopic composition, a number of deposition processes can distort this relationship: changes in moisture sources (Stenni et al., 2016), intermittency or seasonality of precipitation (Sime et al., 2008), boundary layer processes affecting the links between condensation and surface air temperature (Krinner et al., 2008), as well as several post-deposition processes, such as the



effects of winds (Eisen et al., 2008), snow-air exchanges (Casado et al., 2016; Ritter et al., 2016), and diffusion processes in snow and ice (e.g. Johnsen, 1977). Nevertheless, all these processes remain poorly quantified until now. As a result, comparisons between firn core records with precipitation records or simulations have to be performed carefully.

Changes in the atmospheric water cycle can also be investigated using a second order parameter, deuterium excess (hereafter, d-excess). The definition given by Dansgaard (1964) as d-excess $= \delta D - 8$ x $\delta^{18}O$ aims to remove the effect of equilibrium fractionation processes to identify differences in kinetic fractionation between the isotopes of hydrogen and oxygen. In Antarctica, spatial variations of d-excess have been documented through data syntheses, showing an increase from the coast to the plateau

(Masson-Delmotte et al., 2008; Touzeau et al., 2016), but temporal variations of d-excess (seasonal cycle, inter-annual variations) remain under-documented and poorly understood.

Theoretical isotopic modeling studies show that d-excess depends on evaporation conditions, mainly through the impacts of relative humidity (hereafter RH), and sea surface temperature (hereafter SST) on kinetic fractionation at the moisture source (Merlivat and Jouzel, 1979; Petit et al., 1991; Ciais et al.,

1995), and the preservation of the initial vapor signal during transportation towards polar regions (e.g. Jouzel et al., 2013; Bonne et al., 2015). The effect of wind speed on kinetic fractionation is secondary and thus has been neglected in climatic interpretations of d-excess. Some studies usually privileged one variable (RH or SST). For instance, glacial – interglacial d-excess have classically been interpreted to reflect past changes in moisture source SST, neglecting RH effects or assuming co-variations of RH and

SST (Vimeux et al., 1999; Stenni et al., 2001; Vimeux et al., 2001). Recent measurements of d-excess in water vapor from ships have evidenced a close relationship between d-excess and oceanic surface conditions, especially RH, at sub-monthly scales (Uemura et al., 2008; Pfahl and Sodemann, 2014; Kurita et al., 2016). Other recent studies have suggested that evaporation at sea-ice margins may be associated with a high d-excess value due to low RH effects, a process which may not be well captured in

atmospheric general circulation models (e.g. Kurita, GRL, 2011; Steen-Larsen et al, 2017). Several authors have thus identified the potential to identify changes in moisture sources using d-excess (Ciais et al., 1995; Delmotte et al., 2000; Sodemann and Stohl, 2009). The comparison between multi-year isotopic precipitation datasets with the identification of air mass origins using back-trajectories showed however



a complex picture, with no trivial relationship between the latitudinal air mass origin and d-excess (Schlosser et al., 2008; Dittmann et al., 2016). A few studies have also explored sub-annual d-excess variations, and suggested that seasonal d-excess signals cannot be explained without accounting for seasonal changes in moisture transport (e.g. Delmotte et al., 2000). These features have been explored

through the identification of back-trajectory clusters and their relationship with $\delta^{18}$O-d-excess relationships, including phase lags (Markle et al., 2012; Caiazzo et al., 2016; Schlosser et al., 2017). Most of these d-excess studies have been performed using firn records and not precipitation samples. We stress that the impact of post-deposition processes on d-excess remain poorly documented and understood. While relationships between moisture origin and d-excess should in principle be conducted on vapour

measurements to circumvent the uncertainties associated with deposition and post-deposition processes, the available vapour water stable isotope records from Antarctica only cover one or two summer months (Casado et al., 2016; Ritter et al., 2016) and do not yet allow to explore the relationships between moisture transport and seasonal or inter-annual isotopic variations. Also, state-of-the-art atmospheric general circulation models equipped with water stable isotopes such as ECHAM5-wiso can capture d-excess

spatial patterns in Antarctic snow, but they fail to correctly reproduce its seasonal variations (Goursaud et al., 2017a). Finally, the understanding of the climatic signals preserved in d-excess is limited by the available observations. This motivates the search for highly resolved d-excess records from coastal Antarctic firn cores.

*This study*

In this study, we focus on the first highly resolved firn core drilled in coastal Adélie Land, at the TA192A site (66.78 ° S; 139.56 ° S, 602 m a.s.l., hereafter named "TA"). Only two ice cores and one snow pit were previously studied for water stable isotopes in this region, without any d-excess record: the S1C1 ice core (14 km from the TA, 279 m a.s.l., see Goursaud et al., 2017b), the D47 highly resolved pit (78

km from the TA, 1550 m a.s.l., see Ciais et al., 1995) and the Caroline ice core (Yao et al., 1990). The climate of coastal Adélie Land is greatly influenced by katabatic winds (resulting in a very high spatial variability of accumulation), and by the presence of sea-ice (Périard and Pettré, 1993; König-Langlo et al., 1998), including the episodic formation of winter polynya (Adolphs and Wendler, 1995), which lead

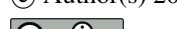

to nearby open water during winter time. The regional climate is well documented since March 1957 at the meteorological station of Dumont d'Urville, where multi-year atmospheric aerosol monitoring has also been performed (e.g. Jourdain and Legrand, 2001). The spatio-temporal variability of regional SMB has also been monitored at annual time scale since 2004 through stake height and snow density

measurement over a 156-km stake-line (91 stakes) (Agosta et al., 2012; Favier et al., 2013). The TA firn core was analysed at sub-annual scale for water isotopes ($\delta^{18}O$ and $\delta D$) and chemistry (Na, $SO_4^{2-}$, and MSA), and dated. Using these records, we explore: (i) the links between the TA isotopic signals, local climate and atmospheric transport, (ii) the possibility to extract a sub-annual signal from such a highly-resolved core, and (iii) how to interpret the d-excess signal of coastal Antarctic ice cores.

In this manuscript, we first present our material and methods (Section 2), then describe our results (Section 3) and compare them with other Antarctic records and the outputs of the ECHAM5-wiso model in our discussion (Section 4), before summarizing our key findings and formulating suggestions for future studies (Section 5).

## 2. Material and method

### 2.1 Field work and laboratory analyses

We present here the results of one firn core drilled at the TA site (66.78 ° S; 139.56 ° S, 602 m a.s.l.), located at 25 km from Dumont d'Urville station (hereafter, DDU) and at 14 km from the S1C1 ice core (Goursaud et al., 2017b; see Fig. 1). The 21.3 m long firn core was drilled on the 29th of January in 2015, when the daily surface air temperature and wind speed were -8.5°C and 3.9 m/s respectively, at the D17

station (9 km from the drilling site).

The FELICS (Fast Electrochemical lightweight Ice Coring System) drill system was used (Ginot et al., 2002; Verfaillie et al., 2012). Firn core pieces were then sealed in polyethylene bags, stapled and stored in clean isothermal boxes. At the end of the field campaign, the boxes were transported in a frozen state to the cold-room facilities of the Institute of Environmental Geoscience (IGE, Grenoble, France). Every

core piece was weighted and its length measured in order to produce a density profile. The cores were sampled at 4 cm resolution, leading to a total of 533 samples for oxygen isotopic ratio and ionic




concentrations following the method described in Goursaud et al. (2017b). Samples devoted to ionic concentration measurements were stored in the cold room until concentrations of sodium (hereafter $Na^+$), sulfate (hereafter $SO_4^{2-}$), and methane sulfonate (hereafter MSA) were analyzed by ion chromatography equipped with a CS12 and an AS11 separator column, respectively. Samples devoted to oxygen isotopic

ratio were sent to LSCE (Gif-sur-Yvette, France) and analyzed following two methods. First, $\delta^{18}O$ was measured by the $CO_2/H_2O$ equilibration method on a Finnigan MAT252, using two standards calibrated to SMOW/SLAP international scales, with an accuracy of 0.05 ‰. Second, $\delta^{18}O$ and $\delta D$ were also measured using a laser cavity ring-down spectroscopy (CRDS) PICARRO analyzer, using the same standards, leading to an accuracy of 0.2 ‰ and 0.7 ‰ for $\delta^{18}O$ and $\delta D$ respectively. The resulting accuracy

of d-excess, calculating using a quadratic approach, is 1.7 ‰.

## 2.2 Datasets

*Instrumental data*

To assess potential climate signals archived in our firn core records, we extracted meteorological data to explore regional climate signals, and outputs of atmospheric models to explore synoptic scale climate

signals. The regional climate is well documented since 1957 thanks to the continuous meteorological monitoring    at    Dumont    d'Urville    station (https://donneespubliques.meteofrance.fr/?fond=produit&id_produit=90&id_rubrique=32), with one gap between 1959 and January 1960. We extracted near-surface temperature, humidity, surface pressure, wind speed and wind direction data computed monthly and annual averages over the periods 1957-2014 and

20 1998-2014.

The monthly average sea-ice concentration was extracted from the Nimbus-7 Scanning Multichannel Microwave Radiometer (SMMR) and Defense Meteorological Satellite Program Special Sensor Microwave/Imagers - Special Sensor Microwave Image/Sounder (DMSP SSM/I-SSMIS) passive microwave data (http://nsidc.org/data/nsidc-0051), over the 50 -90 °S latitudinal ridge (Cavalieri et al.,

1996). We then averaged sea ice concentration data over four longitudinal sectors: (i) "local" (135°E – 145°E), (ii) "Indian" (100°E – 145°E), (iii) "Amundsen" (160°E – 205°E) and (iv) "regional" (100°E – 205°E).



SMB measurements from stake point data were obtained from the Glacioclim observatory (https://glacioclim.osug.fr/). We extracted data from the three stakes closest to the TA drilling site, namely "18.3" (66.77°S, 139.57°E; 1.04 km from the drilling TA site), "19.2" (66.77°S, 139.56°E; 83 m from the drilling TA site), and "20.3" (66.78°S, 139.55°E; 1.00 km from the drilling TA site), all spanning the period 2004-2014.

*Database of surface snow isotopic composition*

In order to compare the d-excess record from the TA firn core with available Antarctic values, we have updated the database of Masson-Delmotte et al. (2008), by adding 26 new data points from precipitation and firn core measurements provided with d-excess (see Table 1). This includes data from five ice cores from the database constituted by the Antarctica2k group (Stenni et al., 2017a). Altogether, the updated database includes 777 locations. This includes 64 coastal sites at an elevation lower than 1000 m a.s.l., (with 19 new datasets).

*Atmospheric reanalyses and backtrajectories*

Unfortunately, because of the katabatic winds around DDU, no instrumental method allows reliable measurements of precipitation (Grazioli et al., 2017a). We use outputs from ERA-interim reanalyses (Dee et al., 2011b), which were shown to be relevant for Antarctic surface mass balance (Bromwich et al., 2011), to provide information on DDU precipitation intra-annual variability. We extracted these outputs from the grid point (0.75° x 0.75°, ~ 80 km x 80 km, point centered at 66.75 °S and 140.25 °E) closest to the TA drilling site over the period 1998-2014, at a 12-hours resolution, and calculated daily, monthly and annual average values.

In order to identify the origin of air masses, back-trajectories were computed using the HySPLIT (Hybrid Single-Particle Lagrangian Integrated Trajectory) model. It is an atmospheric transport and dispersion model developed by the National Oceanic and Atmospheric Administration's (NOAA) Air Research Laboratory (Draxler and Hess, 1998), based on a mixing between Lagrangian and Eulerian approaches (Stein et al., 2015). We set the arrival point at the coordinates of the TA drilling site, at an initial height of 1500 m a.g.l., and used the NCEP/NCAR Global Reanalysis ARL archived data for forcing the



meteorological conditions, as the ERA-interim reanalyses are not available in the required extension. Earlier studies (e.g. Sinclair et al., 2010; Markle et al., 2012) highlighted good performances of NCEP outputs when compared with Antarctic station data after 1979. For instance, previous studies showed that the mean sea level pressure simulated at DDU and averaged on a 5 year running window, was well

captured in NCEP reanalyses after 1986 (correlation coefficient > 0.8, bias < 4 hPa and RMSE < 5 hPa) (Bromwich and Fogt, 2004; Bromwich et al., 2007). Also, Simmons et al. (2004) showed quasi-equal twelve months running mean of 2-meter temperatures for the Southern Hemisphere between the European Re-Analyses ERA-40, the NCEP/NCAR and the Climatic Research Unit CRUTEM2v products. We thus run daily five-days back-trajectories from January 1998 to December 2014. Each back-trajectory was

analyzed for the geographical position of the last simulated point (the estimated start of the trajectory, 5 days prior arrival at DDU), and classified into one of the following four regions, represented on Figure 2 and defined by their longitude (hereafter, lon) and latitude (hereafter, lat) as follows : (i) "Indian": 0 °E<lon < 160 °E and lat > -66 °, (ii) "Plateau": 0 °E < lon < 180 °E and lat<-66°, (iii) "Ross-Sea Secor": 180 °E <lon< 240° E and lat > -75 °, and finally (iv) "WAIS + Pacific": 180 °E <lon<240 °E lat<-75, and

240 °E <lon< 360 °E (no condition for the latitude).

*Atmospheric general circulation and water stable isotope modeling: ECHAM5-wiso*

The potential relationships between large-scale climate variability and regional precipitation isotopic composition was investigated through outputs of a nudged simulation performed with the atmospheric

general circulation model ECHAM5-wiso (Roeckner et al., 2003), equipped with stable-water isotopes (Werner et al., 2011). We chose this model due to demonstrated skills to reproduce spatial and temporal patterns of water stable isotopes in Antarctica (Masson-Delmotte et al., 2008; Werner et al., 2011; Goursaud et al., 2017a; Steen-Larsen et al., 2017), and in Greenland (Steen-Larsen et al, 2017).

In this study, we use the same simulation than Goursaud et al. (2017a), in which the large-scale circulation

(winds) and air temperature were nudged to outputs of the ERA interim reanalyses (Dee et al., 2011a). The skills of the model were assessed over Antarctica for the period 1979-2013. The model was run in a T106 resolution (i.e. ~ 110 km x 110 km horizontal grid size),



## 3. Results

### 3.1 Firn core chronology

*Ice core dating*

The firn core was dated using an annual layer counting method (Fig. 3). As in Goursaud et al. (2017b),

we used concentrations in MSA and non-sea-salt (nss) $SO_4^{2-}$ (hereafter $nssSO_4^{2-}$).

We have explored the validity of an approach using a definition of $nssSO_4^{2-}$ based on a sulfate to sodium mass ratio of 0.25 inferred from summer observations only. The multiple year study of size-segregated aerosol composition conducted at the coast of TA (the DDU station) has demonstrated that sea-salt aerosol is depleted in sulfate with respect to sodium in winter, with a sulfate to sodium mass ratio of 0.13 from

May to October instead of 0.25 (i.e. the sea-water composition) in summer (Jourdain and Legrand, 2002). Even at the high plateau station of Concordia, Legrand et al. (2017a) showed that sea-salt aerosol is depleted in sulfate in winter (sulfate to sodium ratio of 0.13 from May to October). We resampled the sulfate time series recorded in the TA with 12 points per year and inferred seasonal average values from averages over the corresponding subsets of points, as previously done for isotopic records (see Section

3.2). We then calculated $nssSO_4$ (hereafter noted as $nssSO_4^*$) using a sulfate to sodium ratio of 0.25 for points associated to months from November to February and 0.13 for points associated from months from March to September. Note that when ignoring the change in sulfate to sodium ratio in winter, (i.e. applying a sulfate to sodium ratio of 0.25 for all the points of the year), the mean nssSO4 value is lower by 18.2 %, decreasing from $36.5 \pm 12.3$ ppb to $43.1 \pm 11.8$ ppb for $nssSO_4^*$ (see Figure S1 in the Supplementary

Material). We thus applied a calculation of $nssSO_4$ for all points of our firn core, only using the sulfate to sodium ratio obtained from summer observations, as $[nssSO_4^{2-}] = [SO_4^{2-}] - 0.25 [Na^+]$.

Summer (December-January) peaks were identified (i) from $nssSO_4$ values higher than 100 ppb, synchronous with MSA peaks (with no threshold), and (ii) for $nssSO_4$ values higher than 200 ppb (with or without a simultaneous MSA peak). Double $nssSO_4$ peaks were counted as one summer (e.g. 2012,

2003, and 2001). The outcome of layer counting allowed us to estimate annual layer thickness, which, combined with the density profile, allowed us to estimate annual SMB in the firn core. This estimated SMB was then compared with stake area data. The three stake data closest to the TA firn core ("18.3", "19.2" and "20.3", not shown) depict the same inter-annual variability (pairwise coefficient correlations,





r>0.93 and p-values, p<0.001), giving confidence in the use of these measurements to characterize the inter-annual variability of local SMB. The comparison with the stake data shows that our initial layer counted chronology results in a mismatch in the measured versus estimated SMB for year 2008 (See Fig. 4a). This mismatch can be resolved by identifying one more summer peak in the chemical records (thin

green line, Fig. 3). The revised firn core SMB record from this revised chronology shows correlation coefficients between the stake data and the TA firn core varying from 0.64 for the "20.3" to 0.83 for the "19.2" (p<0.05), with coherent inter-annual variability (See Fig. 4b).

Peaks in $\delta^{18}O$ or d-excess were not used in our layer counting, so that our age scale is independent of a climatic interpretation of water stable isotopes (e.g. assumption of synchronicity between temperature

seasonal cycles and water stable isotope records). We note an uncertainty in layer counting of 3 years when comparing the outcome of layer counting using chemical records with $\delta^{18}O$ peaks.

As a result, we consider that the "best guess" chronology results from the annual layer counting based on nssSO$_4$ and MSA refined with the comparison with stake data, giving a total of 18 summer peaks (green vertical lines, Fig. 3). In the following, we thus use the dated firn core records covering the complete

period 1998-2014. We note that our chronology is more robust for the period 2004-2014, for which stake area SMB data are available.

*Potential post-deposition effects*

In order to test whether the available d-excess records are not affected by post-deposition effects, one

may apply calculations of diffusion (e.g. Johnsen et al., 2000; van der Wel et al., 2015; Jones et al., 2017). However, many records are not available as depth profiles, and annual accumulation rate data are missing, precluding a systematic approach. We thus applied a simple approach to quantify how the seasonal $\delta^{18}O$ and d-excess amplitudes vary through time in firn records, as an indicator of potential post-diffusion effects. For this purpose, we calculate the ratio between the mean amplitude of the most recent three

complete seasonal cycles (2011-2014 for TA) and the average seasonal amplitude for the whole record (1998-2014 for TA). If seasonal cycles are stable through time, and if there is no significant smoothing due to post-deposition effects, we should obtain a ratio of 1. However, it is expected to be above 1 in the case of large post-deposition smoothing. We obtain a ratio of 0.5 for TA $\delta^{18}O$ data, possibly reflecting



the inter-annual variability of the $\delta^{18}$O seasonal amplitude. We repeated the same exercise with all the 8 other sub-annual $\delta^{18}$O records from our database (see Table S2 in the Supplementary Material.). Discarding an outlier (NUS 08-7), all ratios are between 1.0 and 2.9. Ratios based on d-excess amplitudes are similar to those found for $\delta^{18}$O (see Table S3 in the Supplementary Material). For the TA firn core,

we again obtain a ratio of 0.5 (see Section 3.3). We also note high ratios for d-excess data in the NUS 08-7. Except for the ratios calculated in the WDC06A, which notably differs for d-excess (1.1) compared to $\delta^{18}$O (2.9), other ratios for d-excess data vary between 1.0 and 1.4, with 20 % maximum difference compared to the corresponding ratio for $\delta^{18}$O data.

These results suggest that (i) potential post-deposition effects in the TA can be neglected.

Notwithstanding, a potential loss of seasonal amplitude in the other average time series compared to the most recent seasonal cycles cannot be discarded, and has to be considered in the comparison of seasonal amplitudes, from one core to the other, in the comparison with the seasonal amplitude of precipitation $\delta^{18}$O time series, and with ECHAM5-wiso outputs.

## 3.2 Mean values

*Mean climate from instrumental data*

Before reporting the mean values from TA records, we describe the available meteorological data. A time-averaged statistical description of the available meteorological data measured at DDU, the station closest to the drilling site, is given in Table 2 for the whole available measurement period prior to 2015 (1957-2014), and over the period covered by our TA records, 1998-2014. For all the considered parameters

(near-surface temperature, wind direction, wind speed, humidity, and surface pressure), the time-averaged values differ by less than 8% (the maximum deviation being for the wind direction) over the period 1998-2014 compared to the whole available period. Standard deviations calculated over these two time periods also differ by less than one respective standard deviation unit, except for wind direction which shows much less variability over the recent period. We conclude that the local climate of the period 1998-2014

is representative of the multi-decadal climate state since 1957. In ERA-interim, the average precipitation is 53.7 ± 7.6 cm w.e. y$^{-1}$ and the average SMB (calculated as precipitation minus evaporation or sublimation) is 40.5 ± 8.3 cm w.e. y$^{-1}$ over the period 1998-2014.



Finally, we compare the statistical description of the sea-ice concentration for the four aforementioned regions (see Section 2.2) over the available period 1979-2014, with the period covered by the TA firn core 1998-2014 (see Table 3). We note that the mean difference between the two periods is maximum for the local sea-ice concentration (135 °E – 145 °E), with 8.9 % difference, whereas it is remains below 1.5

% for the other sectors. The extrema (minimum and maximum values) varies of 0.5 % on average (all regions included) from one sector to another, with a maximum difference of 2.9 %. As a result, the mean sea-ice concentrations of the period 1998-2014 are also representative for the last decades over large sectors from Amundsen to Indian Ocean.

*Mean values recorded in the TA firn core*

Time-averaged values calculated from TA records are reported in Table 4. The average SMB is $75.2 \pm 15.0$ cm w.e. $y^{-1}$. Stake data points from Glacioclim show that this site of high accumulation is located in an area of large spatial variability. This features is confirmed by the values given by (i) the stake data closest to the TA site ("19.2", 100 m for the TA site and associated to a $76.6 \pm 25.8$ cm w.e. $y^{-1}$ mean

accumulation rate) compared to further stake data ("18.2", 1.04 km from the TA site and associated to a $47.7 \pm 15.7$ cm w.e. $y^{-1}$ mean accumulation rate), (ii) our mean SMB reconstruction from the S1C1 ice core almost four times lower than for the TA ($21.8 \pm 6.9$ cm w.e. $y^{-1}$ (Goursaud et al., 2017b)) (iii) and meso-scale fingerprints such as the SMB estimated for coastal Adélie Land by Pettré et al. (1986), based on measurements at stakes located from 500 m to 5 km from the coast and the estimated SMB simulated

by ERA-interim 85.7 % lower than for the TA (see Table 2).

The TA average $\delta^{18}O$ value is $-19.3 \pm 3.1$ ‰, close to the average S1C1 $\delta^{18}O$ of $-18.9 \pm 1.7$ ‰, and the average TA d-excess is $5.4 \pm 2.2$ ‰. Compared to the 64 points located at an elevation lower than 1000 m a.s.l. from our database, the TA $\delta^{18}O$ and d-excess average values are slightly higher than the average low-elevation records ( $-22.7 \pm 8.8$ ‰ and $4.8 \pm 2.3$ ‰ for $\delta^{18}O$ and d-excess respectively, see Fig. 5).

Finally, the average TA concentrations in $Na^+$, MSA and $nssSO_4$ are of $126.0 \pm 276.5$ ppb, $4.5 \pm 5.6$ ppb and $36.5 \pm 44.2$ ppb respectively. Note that the $Na^+$ average concentration value is affected by strong peaks in 2003 and 2004 with annual values of 369.4 and 388.5 ppb. Excluding these two peaks, the





average Na$^+$ concentration is reduced to 93.2 ppb (with a standard deviation of 38.6 ppb). These concentrations will be discussed later in Section 4.4.

To summarize our findings, the TA records encompass a period (1998-2014) representative of multi-decadal mean climatic conditions. The isotopic mean values appear lower than the average of other Antarctic low elevation records (as shown in Fig. 4). The local SMB is remarkably high for East Antarctica, consistent with stake measurements performed close to the TA site.

## 3.3 Inter-annual variations

In the following, we refer to seasons as follows: summer (December to February), autumn (March to May), winter from (June to September) and spring (October and November). This cutting was defined based on the mean seasonal cycle of temperature, showing the highest values from December to February, and a plateau of low values from May to September (see Fig. 8a). In the TA records, resampled with 12 points per year, we identified seasonal average values by calculating averages over the corresponding subsets of points (e.g. for autumn, we select from the 3$^{rd}$ to the 5$^{th}$ points out of the 12 resampled points within the year). We are fully aware that this is a simplistic approach, assuming a regular distribution of precipitation year round, and that our chronology is more accurate for summer than for other seasons, due to the layer counting method. We nevertheless checked that the distribution of precipitation simulated by ERA-interim within each year is rather homogeneous (see Table S4 in the Supplementary Material).

*Trends in time-series*

We report here the analysis of potential trends from 1998 to 2014, and the identification of remarkable years. Figure 6 and 7 display the time series of meteorological variables, sea ice concentration and TA records over 1998-2014. Sea-ice concentrations from all selected sectors in this study (see Section 2.2) are correlated (r>0.56 and p>0.05), particularly the "local" and the "Indian" sea-ice concentrations (r=1.0 and p<0.05), "local" sector being part of the "Indian" sector. As a consequence, although we explored sea-ice records from all sectors described in Section 2.3, we thus only depict the local sea-ice concentration (i.e. in the 135-145 °E sector) on Fig. 6. On Figures 6 and 7, we chose not to display





standard deviations for readability, but they are reported in the Supplementary Material (see Fig. S5 and S5).

Significant increasing trends are detected in the annual values of TA d-excess (0.11 ‰ y$^{-1}$, r=0.61 and p<0.05) as well as of local and Indian Ocean sea-ice concentrations (0.10 % y$^{-1}$, r=0.53 and r=0.52 respectively, p<0.05). The sea ice trend is the largest in summer, they disappear if we discard the value observed in 2013.

*Pairwise linear regressions between variables*

We performed pairwise linear regressions for all records (meteorological and firn core records), using on one side, annual averages and on the other side, monthly or seasonal values. As previously observed by Comiso et al. (2017), we report a significant anti-correlation between annual regional sea-ice concentration (i.e. 100 – 205 °E, but not with other sectors) and DDU near-surface air temperature (r=-0.56 and p<0.05). This relationship is strongest in autumn (r = -0.75 and p<0.05), where it holds for sea ice in all sectors, and disappears in spring or summer. Confirming earlier studies (Minikin et al., 1998), we observe a close correlation between annual concentrations of MSA and nssSO$_4$ (r=0.76, p<0.05). Statistically significant linear relationships appear between the isotopic signals ($\delta^{18}$O and d-excess) and nssSO$_4$ in spring (r=0.65 and r=0.55 respectively, p<0.05), and only between d-excess and nssSO$_4$ in autumn (r=0.65 and p<0.05). We find no relationship between $\delta^{18}$O and temperature at all. Our record depicts a significant anti-correlation between annual values of TA SMB and d-excess (r=-0.59 and p<0.05).

In order to understand the specificities of the TA record, we explored the temporal correlation between $\delta^{18}$O and d-excess from all available Antarctic records (see Table 5, cells in bold for significant relationships), using all data points (in order to be able to exploit non-dated depth profiles) as well as inter-annual variations, when available. When focusing on significant results, we note that most precipitation datasets depict an anti-correlation between d-excess and $\delta^{18}$O. Although we are cautious with the short DDU precipitation time series (with only 19 points, and 0.05<p-values<0.10, cell in italitcs in Table 5), it shows a positive relationship, similar to the one identified in the TA record. At the inter-annual scale, significant results correspond to anti-correlations between d-excess and $\delta^{18}$O for 3 out of 9

precipitation records. We conclude that the positive correlation observed in the TA records is unusual in an Antarctic context.

*Remarkable years*

Using only annual SMB, water stable isotope and chemistry TA records, we finally searched for remarkable years, defined here as deviating from the 1998-2014 mean value by at least 2 standard deviations. We highlight three remarkable years (red-shaded for high values and blue-shaded for low values, Fig. 6 and 7):

→ Year 2007: very low SMB in in TA data.

→ Year 2009: remarkably high $\delta^{18}O$ values.

→ Year 2011: high MSA and wind speed values.

We had previously noted that years 2003 and 2004 are associated with very high $Na^+$ values and add that year 1999 experienced low $nssSO_4$ values and year 2013 high local sea-ice concentration. The remarkable large sea-ice concentration was also observed at a larger scale by Reid et al. (2015).

To summarize, we identify increasing trends in d-excess and sea ice concentration, no significant correlation between TA $\delta^{18}O$ and temperature, and an anti-correlation between d-excess and SMB. We also note 2 remarkable years in SMB ("dry" 2007) and $\delta^{18}O$ ("high" 2009). Finally, no systematic relationships are identified between chemistry and water stable isotope signals (e.g. parallel trends, inter-annual correlation, and remarkable years).

## 3.4 Intra-annual scale

*Mean cycles*

The high resolution of the TA record allows us to describe the mean seasonal cycles (see Fig. 8), and to explore all seasonal cycles to explore how they vary from year to year.

Among the meteorological variables, only near-surface temperature, relative humidity and sea level pressure show a clear seasonal cycle. Temperature (Fig. 8a) is minimum in July and maximum in January, while relative humidity and pressure (Fig. 8b and 8c respectively) are minimum in spring (in November



and October respectively), and a maximum in winter (in August and June respectively), as reported in earlier studies (Pettré and Périard, 1996). The average seasonal cycles of wind speed and wind direction are flat but marked by large inter-annual variations (Fig. 8d and 8e). Finally, the local sea-ice concentration shows a rapid advance from March to June, a plateauing from June to October and a rapid retreat from October to November, with a minimum in February (see Fig. 8c), as previously reported by Massom et al. (2013).

In the TA firn core, Na, nssSO$_4$, and MSA show symmetric cycles with minima in winter and maxima in summer (by construction of our time scale) (see Fig8 g-i), consistent with previous studies of aerosols and ice core signals (e.g. Preunkert et al., 2008). The $\delta^{18}$O seasonal cycle is surprisingly asymmetric (Fig. 8j), with a maximum in December and a minimum in April, thus not in phase with the seasonal cycles of local sea ice concentration (Fig. 8f) nor DDU temperature (Fig. 8a). The mean d-excess seasonal cycle (Fig. 8k) is marked by a maximum in February, two months after the $\delta^{18}$O maximum, and a minimum in October, six months after the $\delta^{18}$O minimum. We then calculated the mean of the isotopic seasonal amplitudes, preferentially to the amplitude of the mean seasonal cycle, due to the different timing of peaks from one year to another. The mean $\delta^{18}$O seasonal amplitude is 8.6 ± 2.1 ‰, more than three times higher than found in the S1C1 ice core, and close to the DSSA mean $\delta^{18}$O seasonal cycle of 8.0 ± 2.8 ‰. The mean d-excess seasonal amplitude is 6.5 ± 2.8 ‰, close to the DSSA value of 5.3 ± 1.0 ‰. Compared with other precipitation and firn/ice core isotopic data from other regions of Antarctica (see Table S7 in the Supplementary Material), the average seasonal amplitude obtained from TA $\delta^{18}$O is closest to the one obtained at KM, BI sites in Dronning Maud Land, and Vernadsky or Rothera in Peninsula, but is much larger than identified from NUS 08-7 or WDC06A, and significantly smaller than at Halley (by a factor of almost 2), Neumayer (a factor of 2.3), Dome C or Dome F (a factor of ~4). In addition to DSSA, the average seasonal amplitude obtained from TA d-excess is also comparable to the one obtained in the KM, BI, and the IND25B5 firn cores in Dronning Maud Land, but is systematically higher (by a factor higher than 3) than in precipitation datasets. This calls for systematic comparisons of d-excess seasonal amplitudes in precipitation and snow data.

Due to their common symmetric aspect, significant positive linear relationships emerge from the mean seasonal cycles of (i) temperature, nssSO$_4$ and MSA (r>0.93 and p<0.05), (ii) nssSO$_4$ and MSA (r=0.97

and $p<0.05$), (iii) $nssSO_4$ and $Na^+$ ($r=0.93$ and $p<0.05$) and finally (iv) $\delta^{18}O$ with $nssSO_4$ ($r=0.75$ and $p<0.05$). Due to the asymmetry of water stable isotope seasonal cycles, no linear relationship is detected between the seasonal cycles of temperature, $\delta^{18}O$, and d-excess. Finally, the seasonal cycle of d-excess is clearly anti-correlated with all sea ice concentration indices (local, Indian, Amundsen and regional), with

correlation coefficients varying between -0.83 and -0.80.

*Inter-annual variability of peaks*

Over the whole period covered by the TA firn core (1998-2014), the seasonal cycle of $\delta^{18}O$ shows a large inter-annual variability (see Table 6). $\delta^{18}O$ maximum values occur primarily in summer (41 % of the

time) and winter (41 %), and more rarely in spring (12 %) and in autumn (6%). The same feature is observable for d-excess, which most of the time has its maximum in summer (38 %) and winter (43 %), and more scarcely in spring (6%) and in autumn (13%).

In summary, the TA water stable isotope seasonal cycles displays an asymmetry, with higher isotopic

values in austral spring than in austral autumn. The TA d-excess seasonal cycle is anti-correlated with local sea-ice concentration. Finally, the TA isotopic seasonal cycles show a high inter-annual variability from one seasonal cycle to another one, with no recurrent pattern between those of $\delta^{18}O$ and d-excess.

## 3.5 Influence of synoptic weather on TA records: insights from ECHAM5-wiso simulation and back-trajectories

In order to explore the influence of the synoptic scale weather on TA records, we explore outputs of ECHAM5-wiso and back-trajectory calculations, driven by atmospheric reanalyses. None of the associated atmospheric simulations does resolve local processes such as katabatic winds or sea breeze. We used the ECHAM5-wiso model outputs to explore the following questions: (i) Do ECHAM5-wiso outputs show similarities with the corresponding observed variables for their inter-annual variability,

trends and remarkable years? (ii) What are the simulated seasonal cycles for $\delta^{18}O$ and d-excess? (iii) What are the simulated relationships between surface air temperature and $\delta^{18}O$, and $\delta^{18}O$ and d-excess at the seasonal and inter-annual scales?



*ECHAM5-wiso similarities with the corresponding observed variables*

For inter-annual variations, the annual means of near-surface temperature measured at DDU and the simulated 2-meter temperature (hereafter 2m-T) are significantly correlated (slope of $0.85 \pm 0.19$, r=0.76 and p<0.05). This relationship holds when selecting any season. It is the strongest in winter (slope of 1.3

$\pm 0.1$ °C °C$^{-1}$, r=0.96 and p<0.05), and the weakest in summer (slope of 0.96 $\pm$ 0.3 °C °C$^{-1}$, r=0.69 and p<0.05). The simulated wind speed is only significantly related to the observed wind speed during spring and summer (slope of $0.36 \pm 0.15$ m.s$^{-1}$ (m.s$^{-1}$)$^{-1}$ and $0.21 \pm 0.08$ m.s$^{-1}$ (m.s$^{-1}$)$^{-1}$ respectively, r=0.54 with p<0.05 for both relationships). For $\delta^{18}O$, model outputs are only correlated with the TA record in winter (slope of $0.25 \pm 0.11$ ‰ ‰$^{-1}$, r=0.53 with p<0.05), and no correlation is identified for d-excess. We found

no significant trend in any model output over 1998-2013.

In terms of remarkable years, ECHAM5-wiso shows a low $\delta^{18}O$ value in 1998 and a high d-excess level in 2007 (see S8 in the Supplementary Material). Only year 2007 is remarkable in both the data (low reconstructed SMB) and the model. We thus explored more deeply the model. The high d-excess value was simulated the 7$^{th}$ of May (see Table S9a in the Supplementary Material). When comparing from the

6$^{th}$ to the 8$^{th}$ of May in 2007, with daily averages over the period 1979-2013, the model simulates similar near-surface temperature, but particularly low precipitation, and wind components (zonal and meridional). Despite the small precipitation amount, the daily isotopic anomaly is sufficiently large to drive the annual anomaly (see Fig. S9b in the Supplementary Material). We nevertheless remain cautious with this value which could be due to a numerical artefact.

Neither $\delta^{18}O$ – temperature, nor d-excess - SMB relationships are identified in ECHAM5-wiso seasonal or annual outputs for precipitation and d-excess. Likewise, no significant relationship could be identified between d-excess and SMB simulated by ERA using both annual and seasonal values. A systematic positive correlation is identified between d-excess and $\delta^{18}O$, except in summer. It is the strongest in austral spring, with a correlation coefficient of 0.75 (with a slope of 0.61 ‰ ‰$^{-1}$).

*Simulated seasonal cycles for $\delta^{18}O$ and d-excess*

We now explore the simulated seasonal variations in $\delta^{18}O$ and d-excess over the period of simulation (1998-2013, see Table 6). The peaks in the simulated $\delta^{18}O$ predominantly occur in spring and summer





(25 % and 63 % respectively), while it only happens 12 % of the time in winter and never in autumn. The simulated d-excess peaks most often in autumn (69 %) and secondarily in winter (31 %), but never during the other seasons. As a result, the model simulates more regular isotopic seasonal cycles with $\delta^{18}$O maxima during spring to summer seasons, and d-excess maxima during autumn to winter seasons, than
identified in the TA record.

*Origin of air masses*

Finally, we used the HYSPLIT back-trajectory model to count the proportion (in percentage) of air mass back-trajectories, based on daily calculations over the period 1998-2014, and averaged at the annual and
seasonal scale, from four different regions (see Section 2.3): the plateau, the Indian Ocean, the Ross Sea sector (hereafter RSS), and the West Antarctic Ice Sheet with the Pacific Ocean (hereafter WAIS+Pacific), as displayed on Fig. 9. In average, the highest proportion of air masses comes from the Indian Ocean (49.8 ± 6.4 % over the period 1998-2014) and the East Antarctic plateau (34.3 ± 3.8 % over the period 1998-2014), while a small proportion of air masses come from the WAIS and the Pacific Ocean
sectors (9.6 ± 3.6 % over the period 1998-2014), and from the RSS (6.3 ± 2.4 % over the period 1998-2014).

Inter-annual variations in back trajectories (Fig. 9a and 9b) reveal a positive trend for the fraction of air masses coming from WAIS+Pacific (slope of 6.5 ± 2.1 % $y^{-1}$, r=0.63 and p<0.05), and remarkable years: 1999, which was identified as a remarkable high $\delta^{18}$O value and low $nssSO_4$ in our TA records, is here
associated with a minimum of back-trajectories from the Plateau, and year 2011 which was associated with particular high MSA in our TA record, shows a minimum of back-trajectories from the Indian region and maxima of back-trajectories from the Ross and the WAIS+Pacific regions.

The seasonal cycles of back-trajectories per region is shown on Figures 9c and 9d. The percentage of back-trajectories coming from the Plateau display peaks in autumn and spring (March and November),
those from the Ross Sea sector in winter and summer (January and June), those from the Indian Ocean in autumn and winter (May and August), and finally those from the WAIS+Pacific region in spring (November). We note a significant linear correlation between the seasonal cycles of the percentage of $\delta^{18}$O and back-trajectories coming from the Ross sea sector (r=0.68 and p<0.05), and from the




WAIS+Pacific region (r=0.59 and p<0.05), and between the seasonal cycles of d-excess and the percentage of back-trajectories coming from the WAIS+Pacific region (r=-0.68 and p<0.05).

Finally, we associated each daily back-trajectory to daily precipitation $\delta^{18}$O and d-excess values simulated by ECHAM5-wiso in the precipitation, and classified the time series for each variables by back-

trajectories sectors. We then computed the corresponding seasonal cycles (see Fig. 10). The mean $\delta^{18}$O value is slightly higher for air masses coming from the Indian sector (-20.6 ‰ compared to -21.9± 0.2 for the other sectors).

The asymmetry in $\delta^{18}$O is particularly well marked for air masses coming from the Ross and WAIS+Pacific regions, with peaks in August and September respectively (resulting in a winter amplitude

more than twice higher compared to the Indian and Plateau sectors), and correspond to higher precipitation amounts during these months during the winter season. The d-excess mean seasonal cycles substantially differ by their amplitude: for air masses coming from WAIS+Pacific, it is 11.8 ‰, with outstanding values in March and October (minima) and in May (higher than the mean plus two standard deviations), whereas it varies between 3.2 ‰ and 3.6 ‰ for the other sectors.

The back-trajectory of the 7$^{th}$ of May in 2007 (shown to be remarkable of simulated d-excess by ECHAM5-wiso) was identified as coming from WAIS+Pacific.

To summarize, we found a mismatch between ECHAM5-wiso outputs and the TA data for d-excess variations. There are no similarities for trends, for seasonal cycles, or for inter-annual isotopic variations. ECHAM5-wiso produces a significant relationship between $\delta^{18}$O and temperature, a feature which is not

identified in the TA record. However, both TA records and ECHAM5-wiso depict an unusual feature in 2007, with dry conditions and high d-excess values. The comparison between TA records and air mass back trajectories suggests that the asymmetry in the $\delta^{18}$O seasonal cycle is due to the precipitation of air masses coming from the WAIS+Pacific region, and that an increased occurrence of (rare) air masses coming from the WAIS+Pacific region is associated with high d-excess values.





## 4. Discussion

### 4.1 SMB

The estimated SMB of East Antarctica does not show a clear trend since 1900 (Favier et al., 2017). Recent studies (Altnau et al., 2015; Vega et al., 2016; Ekaykin et al., 2017) report negative SMB trends in coastal areas contrary to positive trends for the plateau. For our study period (17 years for the TA record, 16 years for the ECHAM5-wiso simulation), a slightly increasing but not significant trend (Agosta et al., 2012), the TA firn core, the ERA or ECHAM5-wiso data.

In Adélie Land, a quality controlled SMB dataset has been developed (Favier et al., 2013), but the drivers of SMB spatio-temporal variability remain unexplored (Favier et al., 2017). This is related to the challenges in monitoring 1) precipitation in windy areas 2) sublimation of precipitating snow flakes (Grazioli et al., 2017b) in the katabatic flow, 3) and the amounts of surface erosion or deposition according to surface wind convergence or divergence, of drifting snow fluxes, and of sublimation of the drifting snow particles (Gallée et al., 2013; Amory et al., 2016; Amory et al., 2017). The low correlation (over 1998-2006) between TA192A annual accumulation and from the first shallow ice core ("S1C1" (Goursaud et al., 2017b) ), collected 14 km from TA192A site, demonstrates this complexity, even though this mismatch may be explained by age scale uncertainties. The S1C1 reconstructed accumulation was also weakly correlated with stake data and model outputs, reflecting the random snow accumulation amounts due to the presence or absence of sastrugis, and the potential occurrence of annual erosion at S1C1 site. Here, the TA accumulation record is highly correlated not only with the closest stake data, but also with the ECHAM5-wiso model output over the period 2004-2014, showing the robustness of our reconstruction for this period. The fact that the TA record, the ECHAM5-wiso output for the corresponding grid point, and the "156 km" stake area data are pairwise correlated ($0.79 \leq r \leq 0.90$ and $p<0.05$), indicates that the TA firn core captures a 100 km- scale regional signal. The differences between the local and regional SMB signal are (see Fig.3b): (i) a higher local SMB average compared to the regional SMB, and (ii) the shift of minimum peak of 2007 in the local signal (i.e. the TA firn core and the "19.2" stake data) to 2008 in the regional signal (see the 2007-2008 plateau in the "156 km" network and the 2008 minimum value simulated by the ECHAM5-wiso model in Fig. 3b).



As a conclusion, the absence of similarity between the TA and the S1C1 accumulation reflects the uncertainty in the S1C1 dating resulting from the large spatial variability and from more frequent erosion processes occurring at the S1C1 site (see Fig. S10 in the Supplementary Material). More ice core records within a 100 km area will allow reducing uncertainties in the interpretation of ice core signals, in particular

on the link with the atmospheric variability.

## 4.2 The $\delta^{18}$O – temperature relationship in coastal Antarctic regions

Several studies have shown that the annual $\delta^{18}$O – temperature relationship is weak in coastal regions. As an example, over Dronning Maud Land, Isaksson and Karlén (1994) found a weaker correlation between $\delta^{18}$O records and Halley temperature for coastal ice cores, i.e. for sites below 1000 m a.s.l., with

a correlation coefficient of 0.56 compared to a correlation coefficient of 0.91 for site above 1000 a.s.l. More recently, Abram et al. (2013) reported a coefficient correlation of 0.52 for the relationship between $\delta^{18}$O recorded in the James Ross Island ice core (at a high of 1524 m a.s.l., with a mean reconstructed SMB of 63 cm w.e. y$^{-1}$), and the near-surface temperature measured at Esperanza station (n=56 and p<0.0001,). Closer to Adélie Land, in Victoria Land, Bertler et al. (2011) found a correlation coefficient

of 0.35 between the $\delta^{18}$O recorded in the Victoria Lower Glacier ice core (at a high of 626 m a.s.l. ) and the summer near-surface temperature measured in Scott Base station (n=30 and p<0.0005). In this study, we find no relationship between the near-surface temperature from DDU and the $\delta^{18}$O recorded in the TA firn core, based on annual averages. Similarly, no relationship had been identified in the S1C1 core (Goursaud et al., 2017b). The ECHAM5-wiso model, over the period 1979-2013, selecting the grid point

corresponding to the location of DDU (as well as the TA and S1C1 drilling sites), produces however a significant but weak relationship between the 2m-T and precipitation-weighted $\delta^{18}$O, with a slope of 0.27 ‰ (°C)$^{-1}$ and a correlation coefficient of 0.35 (p<0.05).

Our study shows that changes in air mass trajectories (dynamics) may dominate over thermodynamical controls (condensation temperature) on coastal Adélie Land $\delta^{18}$O signal, as shown by the asymmetry of

the $\delta^{18}$O seasonal cycle recorded in the TA firn core (see Section 3.4 and Fig. 8). The coupling of calculations of air masses back-trajectory and ECHAM5-wiso outputs suggests that $\delta^{18}$O outstanding high





values occurring during winter time would be brought by air masses coming from the WAIS+Pacific sector (see Section 3.5 and Fig. 10).

Water stable isotope, a fingerprint of changes in air mass origins

In the TA firn core, the mean d-excess is 5.4 ± 1.0 ‰, close to the 4.7 ± 0.4 ‰ value simulated by the

ECHAM5-wiso model for the "coastal Indian" region defined in Goursaud et al. (2017a, different definition than in this study), and the 5.2 ± 0.6 ‰ value for the grid point corresponding to the TA drilling site. Inter-annual variations of TA d-excess (see Fig. 7) are anti-correlated with TA reconstructed SMB, a feature not depicted by ECHAM5-wiso. We suggest that air masses associated with small/large precipitation amounts are associated with different trajectories and moisture sources, and that this signal

is preserved in d-excess, as lead by the following hints:

    (i)     the positive trends both for the TA d-excess and the percentage of air masses coming from the WAIS+Pacific region

    (ii)    an anti-correlation between the seasonal cycles of the TA d-excess and percentages of air masses coming from the WAIS+Pacific region

(iii)    the high simulated d-excess amplitude simulated by ECHAM5-wiso for air masses coming from the WAIS+Pacific sector, reflecting outstanding values occurring in autumn and winter times

    (iv)    The particular case of the 7[th] of May in 2007 with very high d-excess values simulated by ECHAM5-wiso, corresponding to an air mass trajectory from the WAIS+Pacific sector

As pointed earlier, the last item is however to take with cautious as it could be due a numerical artefact. Earlier studies showed empirically that the relationship between d-excess and $\delta^{18}O$, and mainly the phase lag between signals within the seasonal cycle may indicate variations of the origin of the moisture source. This phage lag was shown to be of ~ 3-4 months over coastal regions such as Law Dome (Masson-Delmotte et al., 2003), Dronning Maud Land (Vega et al., 2016) and in the Ross Sea sector (Sinclair et

al., 2014). By contrast, most studies identified an anti-phase over the East Antarctic plateau (e.g. Ciais et al., 1995; Landais et al., 2012), and at D47, situated close to the TA drilling site (Ciais et al., 1995). We thus focus on the outcome of the running linear regression between d-excess and $\delta^{18}O$ over 12 points all along the core (see Fig. 11). We focus on the periods (53.3%) when a significant linear relationship is



identified (i.e. p<0.05). The time-averaged correlation coefficient is $0.71 \pm 0.45$, which is consistent with the results obtained from the annual averages (varying from 0.51 in autumn to 0.75 in spring, see Section 3.3). The time-averaged slope is $0.83 \pm 0.83$ ‰ ‰$^{-1}$. These positive values prevail for 91.5 % of the significant linear regressions. However, we observe remarkable deviations from this overall relationship.

Particularly, linear regressions within years 2002 and 2007 show slopes lower than the time-averaged minus two standard deviations (with a minimum value of -1.46 ‰ ‰$^{-1}$ in 2007), and others within year 2011 show surprisingly very high slopes up to 6.9 ‰ ‰$^{-1}$. The years 2007 and 2011 were also previously noticed: the mean d-excess simulated by ECHAM5-wiso for the year 2007 was shown to be driven by the high value occurring the $7^{th}$ of May, associated to air masses coming from the WAIS+Pacific sector; and

year 2011 is associated with a minimum of annual back-trajectories percentage from the Indian region and maxima of back-trajectories from the Ross and the WAIS+Pacific regions. As a result, the $\delta^{18}$O-d-excess relationship may be a fingerprint of changes in air mass origins, and particularly of the occurrence of precipitation of air masses coming from the WAIS+Pacific sector.

We undertook the same exercise with outputs of the ECHAM5-wiso model (see Fig. S11 in the

Supplementary Material), where only 19.2 % of the simulated linear regressions are significant (i.e. p<0.05). All significant relationships have negative correlation coefficients and slopes of time-averaged values $-0.72 \pm 0.25$ ‰ ‰$^{-1}$ and $-0.39 \pm 0.23$ ‰ ‰$^{-1}$ respectively (consistently with what obtained within each year, see Section 3.4). Moreover, these significant relationships do not occur during the remarkable years 2002, 2007 and 2011 identified in the TA firn core.

As a result, we propose that remarkable anomalies in d-excess / $\delta^{18}$O running linear relationships provide an isotopic fingerprint associated with changes in dominant air mass trajectories. But a more comprehensive mechanistic study would be necessary to quantify the fractionation processes associated with different moisture source and transport characteristics.

### 4.3 Limits associated with model-data isotopic comparisons

We note a mismatch between ECHAM5-wiso outputs and the data (see Section 3.5 and Fig. S8 in the Supplementary Material). This could be related to (i) to post-deposition processes associated with wind scoring or snow metamorphism not resolved in ECHAM5-wiso, (ii) the key role of very local atmospheric





circulation effects related to katabatic wind processes, not resolved in large-scale atmospheric reanalyses and simulations, (iii) or the difficulties of ECHAM5-wiso to resolve the processes associated with the ocean boundary vapour d-excess, a mismatch already identified in the Arctic (Steen-Larsen et al., 2017). The first issue is related with the robustness of records from a single coastal firn core. Several studies

have evidenced signal to noise limits (e.g. Graf et al., 2002; Mulvaney et al., 2002). Given the high SMB estimated from TA, diffusion effects can be ignored (Frezzotti et al., 2007), and the estimated inter-annual variations in TA SMB are closely correlated not only with the stake data the closest to the drilling site, but also with the 156 km network stake data and to precipitation from the corresponding grid point of ECHAM5-wiso within a 100 x 100 km area. This finding supports an interpretation of the TA record to

be representative of a regional SBM signal (100 km scale). However, we cannot have any conclusion of the signal to noise aspects of the water stable isotope records, given the lack of coherency between the inter-annual variability in the TA and S1C1 $\delta^{18}$O records for the few years of overlap (with unfortunately no remarkable year in this period), and the lack of any other d-excess record within hundreds of kilometers.

The second source of uncertainty lies in the mismatch between inter-annual variations from coastal Adélie Land meteorological observations and the TA records, with ECHAM5-wiso outputs. For instance, we only see high correlation for the surface air temperature inter-annual variations for winter, and weak correlation for wind speed in spring and summer. These items suggest limitations in the skills of either atmospheric reanalyses or the ECHAM5-wiso model to correctly capture the processes responsible for

local climate variability. We had previously reported the capability of ECHAM5-wiso to correctly simulate observed large-scale features of water stable isotopes and SMB across Antarctica, for spatio-temporal patterns identified from datasets spanning the last decades such as mean values, amplitudes and phases of mean seasonal cycles, amplitude of inter-annual variance, strength of isotope-temperature relationships, d-excess versus $\delta^{18}$O relationships) (Goursaud et al., 2017a). We thus highlight here

specific challenges related to the Antarctic coastline, where local processes associated with katabatic winds, open water (e.g. polynya), and local boundary layer processes (e.g. snow drift) may affect isotopic records without being resolved at the resolution of reanalyses and ECHAM5-wiso simulation.





Our study therefore depicts limited understanding of the drivers of seasonal and inter-annual variability in coastal Adélie Land hydrological cycle, and thus calls for more isotopic measurements (along ice cores, in precipitation and in vapour) in Adélie Land to reduce uncertainties.

## 4.4 Chemistry

We now compare the chemical concentrations recorded in the TA firn core compared to the costal S1C1 firn core (Goursaud et al., 2017a). On the common covered period (1998-2006), we observe that mean chemical concentrations are slightly lower in the TA than the S1C1 firn core from 30 % for $Na^+$, to 50 % for MSA (see Table S11 in the Supplementary Material). This decrease with the increasing distance from the coast (or elevation above sea level) is consistent with atmospheric studies showing a decrease of levels

from the coast to the plateau for sea-salt (Legrand et al., 2017b) and sulfur aerosols (Legrand et al., 2017a). No significant linear regression emerges from any chemical species, highlighting a high spatial variability and/or the uncertainty in the dating of the S1C1 firn core.

Finally, we initially processed chemical measurements in our firn core, to support the isotopic records not only for dating, but also to identify air mass origins, making the hypothesis of three possible cases: (i) Air

masses formed near the sea ice margin may be associated with relatively high d-excess and $\delta^{18}O$ values, due to respectively a high kinetic fractionation due to evaporation under low humidity levels, and limited distillation effects. Such a configuration should be associated to low sea-salt concentrations due to reduced sea-salt emissions when summer sea-ice is present at the site as shown by atmospheric studies (Legrand et al., 2016). (i) By contrast, air masses formed over the Ocean in the absence of sea-ice may

be associated with high $\delta^{18}O$ values, low d-excess and high sea-salt concentrations. (iii) Finally, air masses from central Antarctica may be associated with depleted $\delta^{18}O$ values and high d-excess, while air masses from ocean regions may lead to intermediate $\delta^{18}O$ and d-excess values, due to distillation effects, and evaporation under relatively humid conditions, but with low sea-salt concentrations.

The period from December 2003 to February 2004, associated with $Na^+$ values higher than the mean plus

five standard deviations, probably caused marine advections, is not distinguishable in the isotopic records. And none of the three aforementioned cases was systematically observed.




To make it short, taking into account the definition of summer observations only does not alter our results. The sea-salt and sulfur concentrations measured along the TA records are slightly lower compared to the S1C1 firn core, consistently with the coast-to-plateau depletion previously observed in atmospheric measurement. Unfortunately, we could not use the sea-salt measurements to support our hypotheses regarding the air mass origins associated with isotopic compositions.

## 5. Conclusions and perspectives

In this study, we report the analysis of the first highly resolved firn core drilled in Adélie Land covering the very recent period 1998-2014, with a sub-annual resolution. The chronology was based on chemical tracers (Na, $nssSO_4$ and MSA) and adjusted by one year based on stake area information. Three $\delta^{18}O$ peaks found no counterparts in the chemical records. The high estimated SMB rate of $74.1 \pm 14.1$ cm w.e. $y^{-1}$ gives access to sub-annual records and limits the effects of diffusion (e.g. Johnsen, 1977). The good consistency of the estimated annual SMB variations with observations on stakes reflects that high accumulation amounts are needed to insure that small scale SMB random variability caused by presence of sastrugis, dunes and barchans is negligible when compared to the mean accumulation value. This condition allows avoiding the erosion of seasonal or annual layers, which would lead to removal of the annual cycle of the recorded signal. For this reason, getting long term observations on distributed stake networks around a drilling site, or ground penetrating radar data is crucial to accurately select a drilling site, by retrieving the location of mesoscale accumulation maxima, and by rejecting zones with potential erosion.

Using an updated database of Antarctic surface snow isotopes, we showed that not only $\delta^{18}O$ but also d-excess mean TA values are in line with the range of coastal values in other locations.

Neither in the TA – DDU dataset, nor in the ECHAM5-wiso output do we see any significant relationship between inter-annual variations in $\delta^{18}O$ and local surface air temperature. The anti-correlation between annual reconstructed SMB and TA d-excess leads us to suggest that changes in large-scale atmospheric transport could lead to an explanation for this feature. Particularly back-trajectory simulations from HYSPLIT and atmospheric outputs from ECHAM5-wiso at the seasonal cycle show the occurrence of air masses coming from the WAIS+Pacific sector during autumn and winter times, corresponding to high




simulated d-excess values. Also, the identification of remarkable years both in back-trajectory percentages and in the relationships between d-excess and $\delta^{18}O$ also lead us to evidence a potential in the d-excess - $\delta^{18}O$ to identify remarkable features in moisture transport.

We cannot explain at this stage the observed positive trends in the TA d-excess and local sea-ice extent

5  over the period 1998-2014. We suggest that an improved understanding of the drivers of moisture transport towards coastal Adélie Land can benefit from the interpretation of water stable isotope tracers, especially d-excess, through mechanistic studies and the exploration of global atmospheric models. Ways forward include a better documentation of the spatio-temporal variability in SMB and water stable isotopes using a matrix of coastal firn core records spanning the last decades; a better documentation of

10  the relationships between precipitation and ice core records through the monitoring of the isotopic composition of surface vapour, and precipitation snow and firn (Casado et al., 2016; Ritter et al., 2016); and the implementation of water stable isotopes in regional models resolving the key missing processes linked for instance with katabatic winds, boundary layer processes, wind drift (Gallée et al., 2013).



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




**Table 1: Site, latitude (in °), longitude (in °E) and reference of new d-excess data added to Masson-Delmotte et al. (2008). The table is divided in three parts. The upper part of the table correspond to precipitation, the middle part correspond to ice cores extracted from the Antarctica2k database (Stenni et al., 2017b), and finally the last part of the table are new data compared to our prior database (Goursaud et al., 2017a). Finally, data associated with "*" were not provided with a dating while data in italic have a sub-annual resolution. Note that DE08-2 and D15 ice cores were not dated.**

| Site | Latitude | Longitude | Reference |
|---|---|---|---|
| Frei (south shetland islands)* | -62.20 | 301.04 | (Fernandoy et al., 2012) |
| O'Higgins (north peninsula)* | -63.32 | 302.10 | (Fernandoy et al., 2012) |
| *Halley* | *-75.58* | *333.50* | *(Rozanski et al., 1993)* |
| *Base tte. Marsh* | *-62.12* | *301.44* | *(Rozanski et al., 1993)* |
| *Rothera Point* | *-67.57* | *291.87* | *(Rozanski et al., 1993)* |
| *Vernadsky* | *-65.08* | *296.02* | *(Rozanski et al., 1993)* |
| *Vostok* | *-78.5* | *106.9* | *(Landais et al., 2012)* |
| *DDU* | *-66.7* | *140.00* | *Pers. Comm., Jean Jouzel* |
| *Neumayer* | *-70.7* | *351.60* | *(Schlosser et al., 2008)* |
| *Dome F* | *-77.3* | *39.7* | *(Fujita and Abe, 2006)* |
| *Dome C* | *-75.1* | *123.4* | *(Schlosser et al., 2017)* |
| EDC Dome C | -75.10 | 123.39 | (Stenni et al., 2001) |
| *NUS 08-7* | *-74.12* | *1.60* | *(Steig et al., 2013)* |
| NVFL-1 | -77.11 | 95.07 | (Ekaykin et al., 2017) |
| *WDC06A* | *-79.46* | *247.91* | *(Steig et al., 2013)* |
| *IND 25B5 Coastal DML* | *-71.34* | *11.59* | *(Rahaman et al., 2016)* |
| OH-4* | -63.36 | 302.20 | (Fernandoy et al., 2012) |
| OH-5* | -63.38 | 302.38 | (Fernandoy et al., 2012) |
| OH-6* | -63.45 | 302.24 | (Fernandoy et al., 2012) |
| OH-9* | -63.45 | 302.24 | (Fernandoy et al., 2012) |
| OH-10* | -63.45 | 302.24 | (Fernandoy et al., 2012) |
| *KC* | *-70.52* | *2.95* | *(Vega et al., 2016)* |



| | | | |
|---|---|---|---|
| *KM* | *-70.13* | *1.12* | *(Vega et al., 2016)* |
| *BI* | *-70.40* | *3.03* | *(Vega et al., 2016)* |
| *GIP* | *-80.10* | *159.30* | *(Markle et al., 2012)* |
| DE08-2 | -66.72 | 112.81 | (Delmotte et al., 2000) |
| *DSSA* | *-66.77* | *112.81* | *(Delmotte et al., 2000)* |
| *D15\** | *-66.86* | *139.78* | *Pers. Comm., Jean Jouzel* |
| *TA192A* | *-66.78* | *139.56* | *This study* |



**Table 2: Number of points ("n"), time averages ("μ"), standard deviation ("σ"), minimum ("min") and maximum ("max") values of all the monthly meteorological observations at Dumont d'Urville from Météo France over the period 1957-2014 (with a gap between March 1959 and January 1960, included) and over the period 1998-2014 for near-surface temperature ("Ts", in °C), wind speed ("ws", in m/s), wind direction ("wd", in °E), and relative humidity ("RH", in %), and of annual precipitation and accumulation (precipitation minus evaporation/sublimation) from ERA-interim reanalyses**
5 **("Prec. ERA' and "Accu. ERA" respectively, both in mm w.e. y$^{-1}$).**

| | 1957-2014 | | | | 1998-2014 | | | | | |
| | Ts (°C) | ws (m/s) | wd (°E) | RH (%) | Ts (°C) | ws (m/s) | wd (°E) | RH (%) | Prec. ERA (cm w.e. y$^{-1}$) | Accu ERA (cm w.e. y$^{-1}$) |
|---|---|---|---|---|---|---|---|---|---|---|
| n | 730 | 719 | 383 | 717 | 202 | 201 | 201 | 199 | | |
| μ | -10.9 | 9.6 | 145.3 | 61.4 | -11.1 | 9.1 | 133.8 | 60 | 53.7 | 40.5 |
| σ | 6 | 2.1 | 22.4 | 7.6 | 6.0 | 1.8 | 15.8 | 8.3 | 7.6 | 8.3 |
| min | -23.5 | 4.9 | 90 | 34 | -22.1 | 4.8 | 1.5 | 34.0 | 39.3 | 24.7 |
| max | 1 | 19.5 | 220 | 86 | 0.9 | 14.1 | 172.4 | 85.5 | 70.3 | 56.1 |



**Table 3: Number of points ("n"), time averages ("μ"), standard deviation ("σ"), minimum ("min") and maximum ("max") values of monthly meteorological sea ice concentration over the periods 1979-2014 and 1998-2014 extracted from the Nimbus-7 Scanning Multichannel Microwave Radiometer (SMMR) and Defense Meteorological Satellite Program Special Sensor Microwave/Imagers - Special Sensor Microwave Image/Sounder (DMSP SSM/I-SSMIS) passive microwave data (http://nsidc.org/data/nsidc-0051) (Cavalieri et al., 1996), and for the four regions defined as "Indian": 0 °E<lon < 160 °E and lat > -66 °, "Plateau": 0 °E < lon < 180 °E and lat<-66°, "Ross-Sea Secor": 160 °E <lon< 190° E and lat > -75 °, and "WAIS + Pacific": 190 °E <lon (see Section 2.2).**

| | 1979-2014 | | | | 1998-2014 | | | |
|---|---|---|---|---|---|---|---|---|
| | Indian | local | Amundsen | Regional | Indian | local | Amundsen | Regional |
| n | 432 | 432 | 432 | 432 | 204 | 204 | 204 | 204 |
| μ | 41.1 | 47.0 | 56.6 | 42.9 | 41.6 | 51.2 | 57.4 | 43.4 |
| σ | 6.8 | 7.7 | 11.5 | 8.7 | 6.6 | 8.0 | 11.6 | 8.8 |
| min | 31.0 | 38.2 | 36.3 | 28.7 | 31.1 | 38.4 | 36.6 | 28.9 |
| max | 55.4 | 66.0 | 74.2 | 58.8 | 53.8 | 65.2 | 73.9 | 57.7 |



**Table 4: Time averages ("μ") and standard deviation ("σ") of the reconstructed accumulation (in cm w.e.) and of the signals recorded in the TA192A ice core obtained from the resampling of the isotopic and chemical variables) for $\delta^{18}$O (in ‰), d-excess (in ‰), Na, MSA, and nssSO$_4$ (in ppb), over the 17 annual values.**

| | Accumulation (cm w.e.) | $\delta^{18}$O (‰) | d-excess (‰) | Na$^+$ (ppb) | MSA (ppb) | nssSO$_4$ (ppb) |
|---|---|---|---|---|---|---|
| μ | 75.2 | -19.3 | 5.4 | 126.0 | 4.5 | 36.5 |
| σ | 15.0 | 3.1 | 2.2 | 276.5 | 5.6 | 44.2 |



**Table 5: d-excess versus δ¹⁸O linear relationship of data from our database provided with d-excess over the whole time series (left side of the table) and over annual averages ('inter-annual scale', right side of the table): slope (in ‰‰⁻¹), correlation coefficient ('r'), p-value ("p-val") and standard error of the slope ('stderr' in ‰‰⁻¹). Cells in bold show significant relationship (p<0.05) and the cell in italic is to be taken with caution (see "DDU" line, p<0.10). Inter-annual relationships could not be computed for not dated data (D15 and OH ice cores) as well as for too short time monitored precipitation data, and thus appear as empty cells.**

| | Whole time series | | | | | Inter-annual scale | | | | |
|---|---|---|---|---|---|---|---|---|---|---|
| | Number of points | Slope (‰‰⁻¹) | R | p-val | stderr | Number of points | Slope (‰‰⁻¹) | r | p-val | stderr |
| **EDC Dome C** | **140** | **-0.41** | **-0.31** | **0.00** | **0.11** | **623** | **-0.41** | **-0.31** | **0.00** | **0.11** |
| **NUS 08-7** | **2413** | **-0.36** | **-0.28** | **0.00** | **0.08** | **626** | **-0.36** | **-0.28** | **0.00** | **0.08** |
| NUS 07-1 | 299 | 0.17 | 0.25 | 0.15 | 0.12 | 299 | 0.17 | 0.25 | 0.15 | 0.12 |
| **NVFL-1** | **233** | **0.47** | **0.40** | **0.00** | **0.02** | **233** | **0.47** | **0.40** | **0.00** | **0.02** |
| **WDC06A** | **41120** | **0.25** | **0.25** | **0.00** | **0.00** | **2056** | **0.05** | **0.06** | **0.01** | **0.02** |
| IND 25B5 | 1297 | 0.17 | 0.08 | 0.45 | 0.22 | 140 | 0.17 | 0.08 | 0.45 | 0.22 |
| BI | 404 | 0.01 | 0.02 | 0.70 | 0.03 | **17** | **-0.57** | **-0.57** | **0.01** | **0.20** |
| **KC** | **343** | **0.22** | **0.23** | **0.00** | **0.05** | 48 | 0.05 | 0.06 | 0.70 | 0.12 |
| KM | 425 | 0.04 | 0.06 | 0.24 | 0.04 | **18** | **-0.86** | **-0.57** | **0.01** | **0.29** |
| DSSA | 161 | -0.03 | -0.04 | 0.58 | 0.05 | **6** | **-0.41** | **-0.74** | **0.04** | **0.15** |
| DE08-2 | 58 | -0.11 | -0.08 | 0.57 | 0.19 | 58 | -0.11 | -0.08 | 0.57 | 0.19 |
| D15-1 | 126 | -0.03 | -0.03 | 0.72 | 0.07 | | | | | |
| D15-2 | 191 | -0.06 | -0.10 | 0.19 | 0.04 | | | | | |
| OH4 | 318 | -0.11 | -0.11 | 0.13 | 0.07 | | | | | |
| OH5 | 213 | -0.05 | -0.04 | 0.45 | 0.07 | | | | | |





| | | | | | | | | | |
|---|---|---|---|---|---|---|---|---|---|
| OH6 | 124 | 0.01 | 0.01 | 0.86 | 0.08 | | | | |
| OH9 | 232 | -0.05 | -0.04 | 0.57 | 0.08 | | | | |
| OH10 | 190 | -0.10 | -0.04 | 0.58 | 0.17 | | | | |
| *DDU* | *19* | *0.48* | *0.41* | *0.08* | *0.26* | | | | |
| **Dome C** | **501** | **-1.48** | **-0.84** | **0.00** | **0.04** | **4** | **-2.75** | **-0.98** | **0.02** | **0.42** |
| **Dome F** | **351** | **-1.60** | **-0.89** | **0.00** | **0.05** | | | | |
| **Halley** | **532** | **-0.20** | **-0.18** | **0.00** | **0.05** | 49 | -0.23 | -0.12 | 0.41 | 0.28 |
| **Marsh** | **19** | **-0.86** | **-0.51** | **0.03** | **0.35** | | | | |
| Neumayer | 336 | -0.06 | -0.07 | 0.21 | 0.05 | 19 | 0.28 | 0.31 | 0.20 | 0.21 |
| **Rothera** | **194** | **-1.00** | **-0.58** | **0.00** | **0.10** | **18** | **-1.21** | **-0.86** | **0.00** | **0.18** |
| **Vernadsky** | **372** | **-1.33** | **-0.57** | **0.00** | **0.08** | **35** | **-1.89** | **-0.69** | **0.00** | **0.29** |
| **Vostok** | **27** | **-0.73** | **-0.63** | **0.00** | **0.18** | | | | |



**Table 6: Fraction of annual maxima (in %) of δ¹⁸O and d-excess identified during each season: austral summer ("DJF", i.e. from December to February), austral autumn ("MAM", i.e. from March to May), austral winter ("JJAS", i.e. from June to September) and austral spring ("ON", i.e. from October to November) in the data over the period 1998-2014, and in the ECHAM5-wiso simulation ("model") over the period 1998-2013. The analysis is based on resampled data (ice core) and monthly values (model)**

| Variable | Source | Period | DJF | MAM | JJAS | ON |
|---|---|---|---|---|---|---|
| $\delta^{18}O$ | data | 1998-2014 | 41 | 6 | 41 | 12 |
| | model | 1998-2013 | 63 | 0 | 12 | 25 |
| d-excess | data | 1998-2014 | 41 | 12 | 41 | 6 |
| | model | 1998-2013 | 0 | 69 | 31 | 0 |





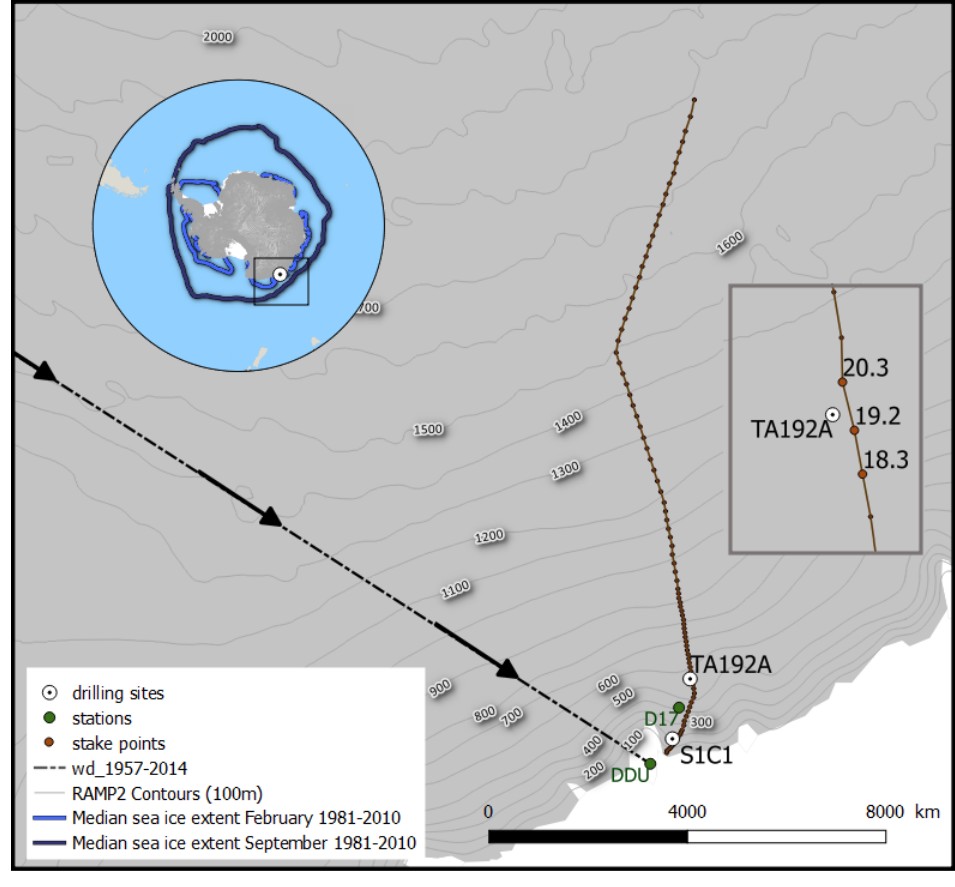

**Figure 1: Map showing the location of the drilling sites of the S1C1 and TA192A firn cores (black points), the Dumont d'Urville and D17 stations (green points), and the stake points (in brown. included the three closest stake points from the TA192A. namely the**
5  **"18.3", the "19.2" and the "20.3, and the "156 km" stake line), the mean wind direction over the periods 1957-2014 (in black). Isohypses (grey lines) shown on the main figure are simulations from Digital elevation model, large scale resolution. Radarsat Antarctic Mapping Project Digital Elevation Model Version 2 (Liu et al., 2001). The map of Antarctica on the top left displays the mean February and September sea ice extent over the period 1981-2010 extracted from the Nimbus-7 Scanning Multichannel Microwave Radiometer (SMMR) and Defense Meteorological Satellite Program Special Sensor Microwave/Imagers - Special Sensor**
10  **Microwave Image/Sounder (DMSP SSM/I-SSMIS) passive microwave data (http://nsidc.org/data/nsidc-0051) (Cavalieri et al., 1996) (light blue. and dark blue lines respectively), and the zoomed area (grey rectangle), while the grey rectangle in the middle right zooms the area around the TA192A drilling site in order to show the three closest stake locations.**





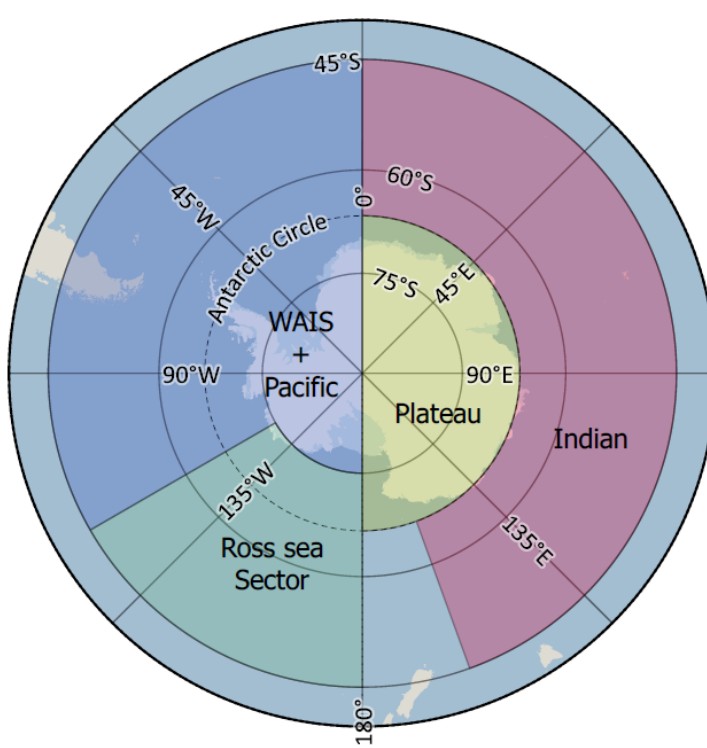

**Figure 2: Representation of the sectors used to classify the last point of the simulated back-trajectories by HYSPLIT over the period 1998-2014, defined as follows: (i) "Indian": 0 °E<lon < 160 °E and lat > -66 °, (ii) "Plateau": 0 °E < lon < 180 °E and lat<-66°, (iii) "Ross-Sea Secor": 180 °E <lon< 240° E and lat > -75 °, and finally (iv) "WAIS + Pacific": 180 °E <lon<240 °E lat<-75, and 240 °E <lon< 360 °E (no condition for the latitude).**





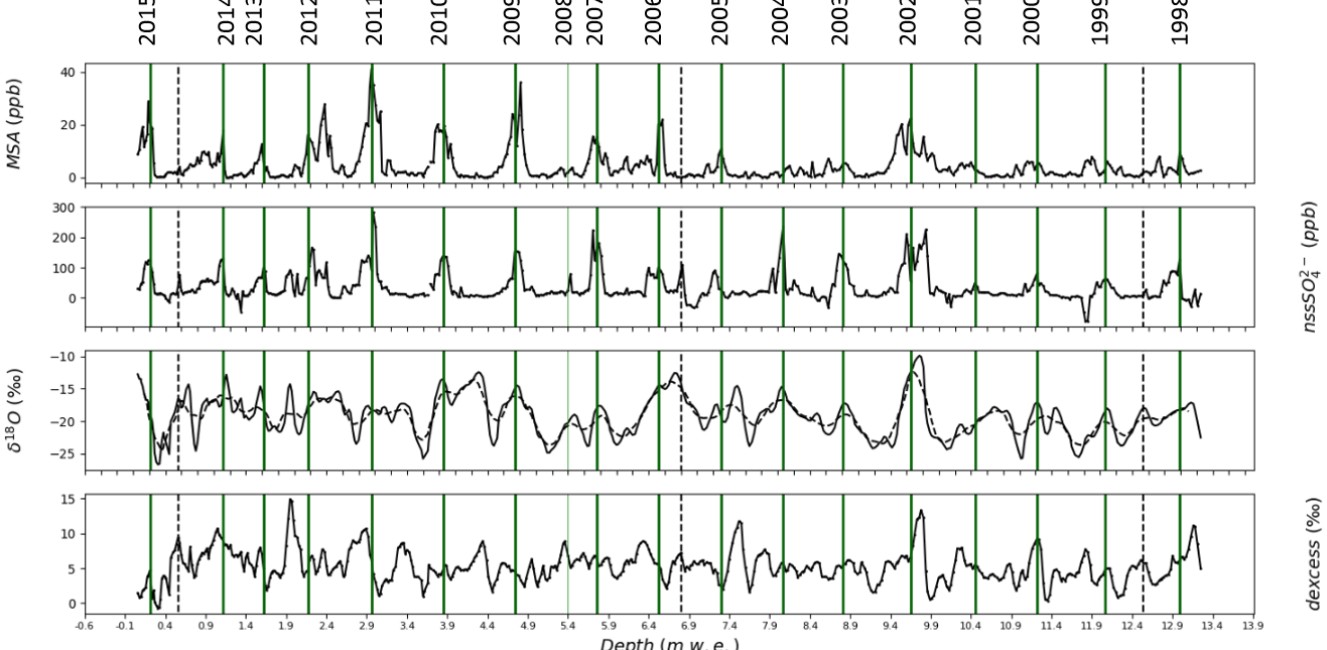

**Figure 3: Identification of annual layers in the TA192A ice core based on the dual identification of nssSO$_4^{2-}$ and MSA summer peaks, and comparison of estimated accumulation with annual accumulation measured at the closest stakes (not shown). The thick vertical green lines correspond to summer peaks identified from chemical signals, while the thin vertical green line shows the additional identification of summer 2008 added to the counted summers from the comparison between the estimated accumulation and the accumulation record from the closest stake data (see Fig. 3). Vertical dashed lines highlight equivocal summer peaks with a sometimes small signal in only one of the chemical records. Water stable isotope records were not used to build the time scale. Note that double peaks in both δ$^{18}$O and d-excess occur repeatedly within one counted year.**







**Figure 4: Annual accumulation (cm w.e. y⁻¹) estimated from the layer counting in the TA192A firn core (blue line), measured at the closest stake point "19.2" (orange line), from the 156 km network stake data (green line) and extracted from the ECHAM5-wiso model (red line), before (a) and after (b) adding the identification of summer 2008 in a time interval of low accumulation rates and skipped in the initial layer counting approach due to the lack of a signal in the MSA record (thin green line in Fig. 2). Correlations between time series (TA192A series inferred from the second dating) are inserted in the lower plot, all linear relationships being significant (p<0.05).**



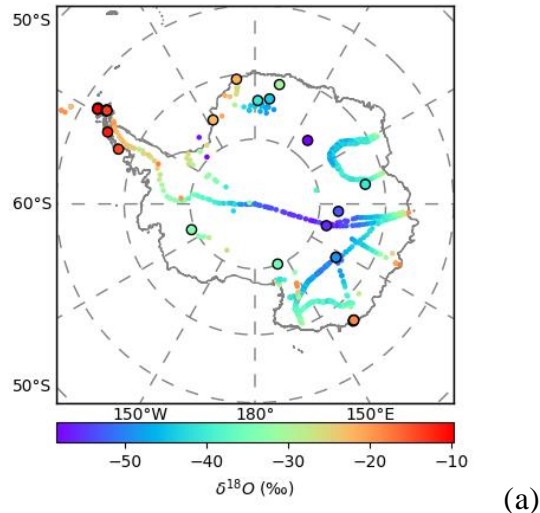
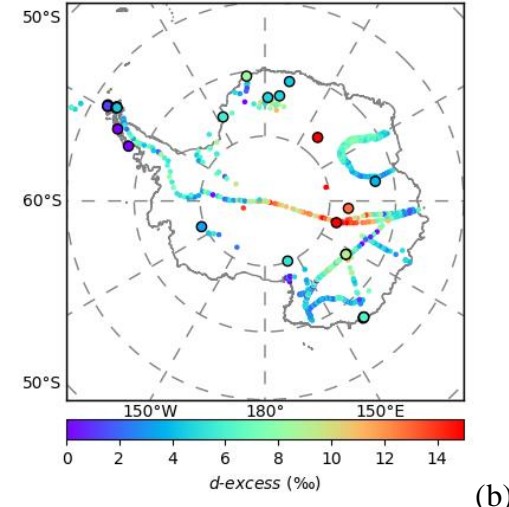

Fig. 5: Spatial distribution of δ¹⁸O (a. in ‰) and d-excess (b. in ‰) in surface Antarctic snow based on our updated database combining data from precipitation, surface snow, pits and shallow ice cores. Bigger points with a black edge correspond to new data compared to Masson-Delmotte et al. (2008).





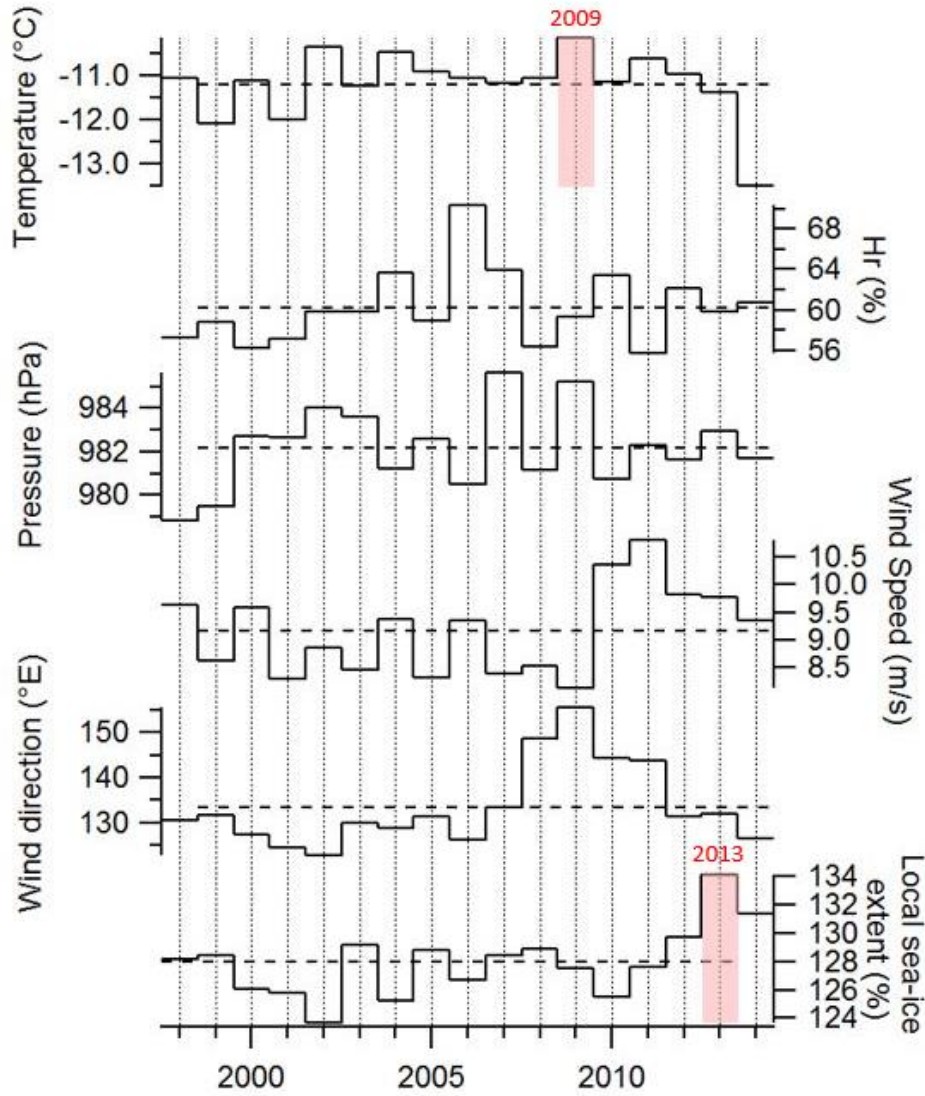

**Figure 6: Meteorological time series over the period 1998-2014 averaged at the inter-annual scale. Near-surface temperature (in °C), relative humidity (in %), sea level pressure (in hPa). wind speed (in m/s) and direction (°E) were provided by Meteo France. The local sea-ice concentration (in %) is extracted in the 135°E-145°E sector (with a latitudinal range of 50°S-90°S) from the Nimbus-7 Scanning Multichannel Microwave Radiometer (SMMR) and Defense Meteorological Satellite Program Special Sensor Microwave/Imagers - Special Sensor Microwave Image/Sounder (DMSP SSM/I-SSMIS) passive microwave data (Cavalieri et al., 1996). Horizontal dashed lines correspond to the climatological averages over 1998-2014 for each parameter. Remarkable years. i.e. associated with values deviating by at least 2 standard deviations from the climatological mean value are highlighted with a red shading (positive anomalies) or blue shading (negative anomalies). The same figure with standard deviations is available in the Supplementary Material (Fig. S5).**





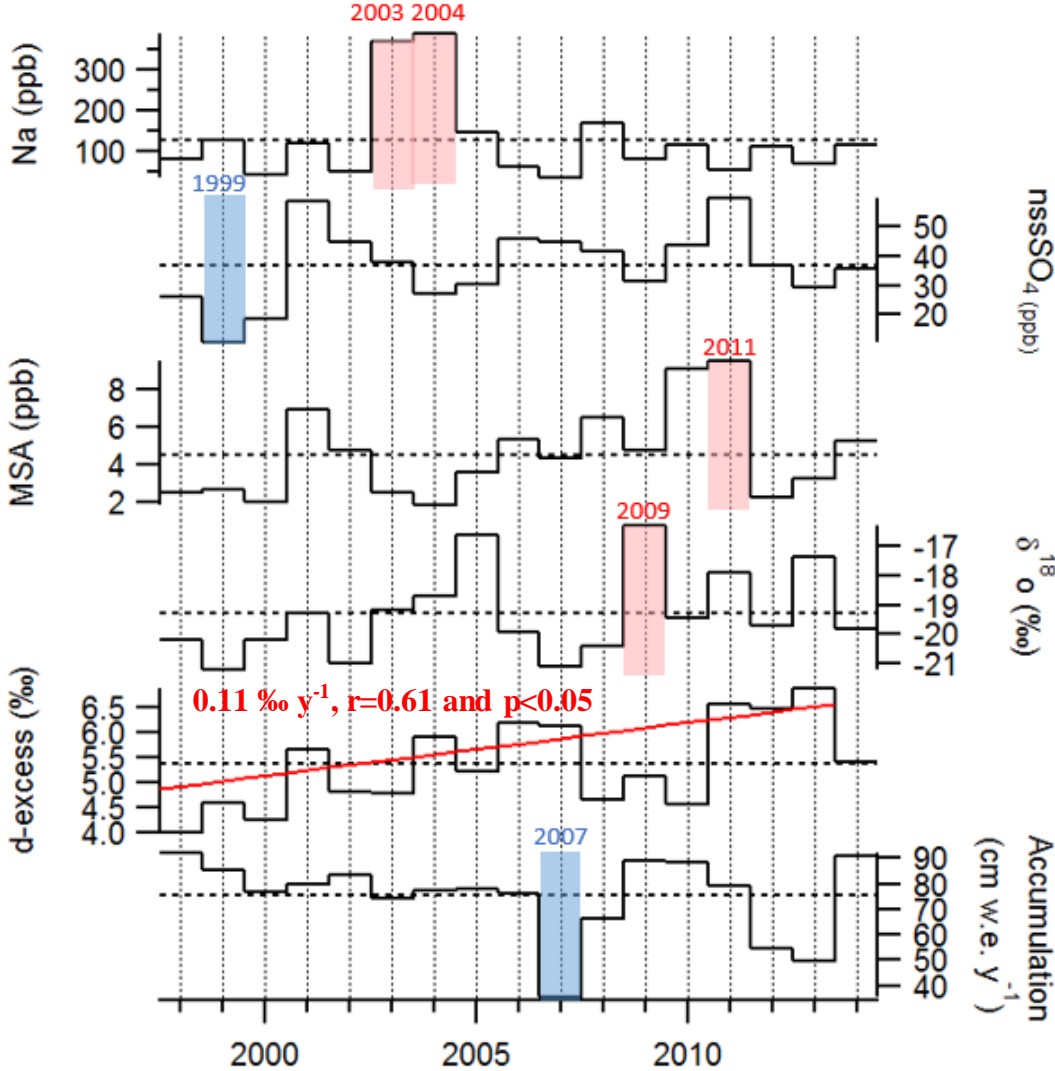

**Figure 7: Dated TA192A ice core annually averaged records over the period 1998-2014: accumulation (in cm w. e. y$^{-1}$), concentrations of Na+ (in ppb), nssSO$_4$ (in ppb), MSA (in ppb), δ$^{18}$O (in ‰) and d-excess (in ‰). Horizontal dashed lines correspond to 1998-2014 average values. Remarkable years (i.e. associated with values deviating by at least 2 standard deviations from the climatological mean value) are highlighted with a red shading (positive anomalies) or blue shading (negative anomalies). The same figure with standard deviations is available in the Supplementary Material (Fig. S6).**





**Figure 8: Mean seasonal cycles over the period 1998-2014. Meteorological observations are averaged from daily data, for near-surface temperature (a. in °C), relative humidity (b. in %), surface pressure (c, in hPa), wind speed (d, in m/s), wind direction (e, in °E), and local sea-ice concentration. i.e. averaged over a 135°E-145°E ridge (f. in %). Seasonal cycles from ice core records are averaged from the resampled time series for Na (g, in ppb), nssSO$_4$ (h, in ppb), MSA (i, in ppb), δ$^{18}$O (j, in ‰) and d-excess (k, in ‰). The inter-annual standard deviation is highlighted with the grey shading.**





**Figures 9: Results of daily back-trajectories calculations over the period 1998-2014 with (i) percentage (in %) of the sum of back-trajectories passing over each defined region: "Indian" (corresponding to 0° E<longitude<160° E and latitude>-66 °), "Ross" (corresponding to -160° E<longitude<120° E and latitude>-75 °), "Plateau" (corresponding to 0° E<longitude<180° E and latitude<-66 °) and "WAIS + Pacific" (corresponding to -150° E<longitude<0 °E) (see Section 2.3); and (ii) averages at the annual scale (a and c) and at the mean seasonal scale (c and d). On the bottom panels . horizontal dashed lines correspond to the mean value and vertical solid lines to standard deviations. Remarkable years. i.e. associated with values deviating by at least 2 standard deviations from the climatological mean value are highlighted with a red shading (positive anomalies) or blue shading (negative anomalies).**

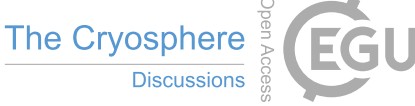



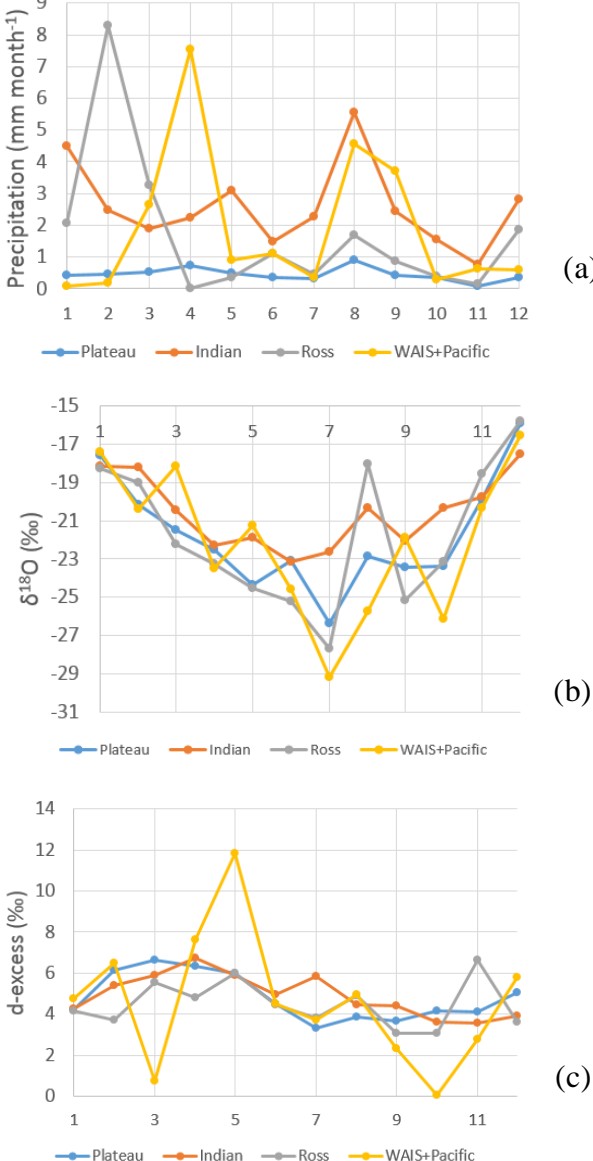

**Figure 10: Seasonal cycles of precipitation (in mm month$^{-1}$. a), $\delta^{18}$O (in ‰, b) and d-excess (in ‰, c) simulated by ECHAM5-wiso by back-trajectories regions "Indian" (corresponding to 0° E<longitude<160° E and latitude>-66 °), "Ross" (corresponding to -160° E<longitude<120° E and latitude>-75 °), "Plateau" (corresponding to 0° E<longitude<180° E and latitude<-66 °) and "WAIS + Pacific" (corresponding to -150° E<longitude<0 °E) (see Section 2.3).**



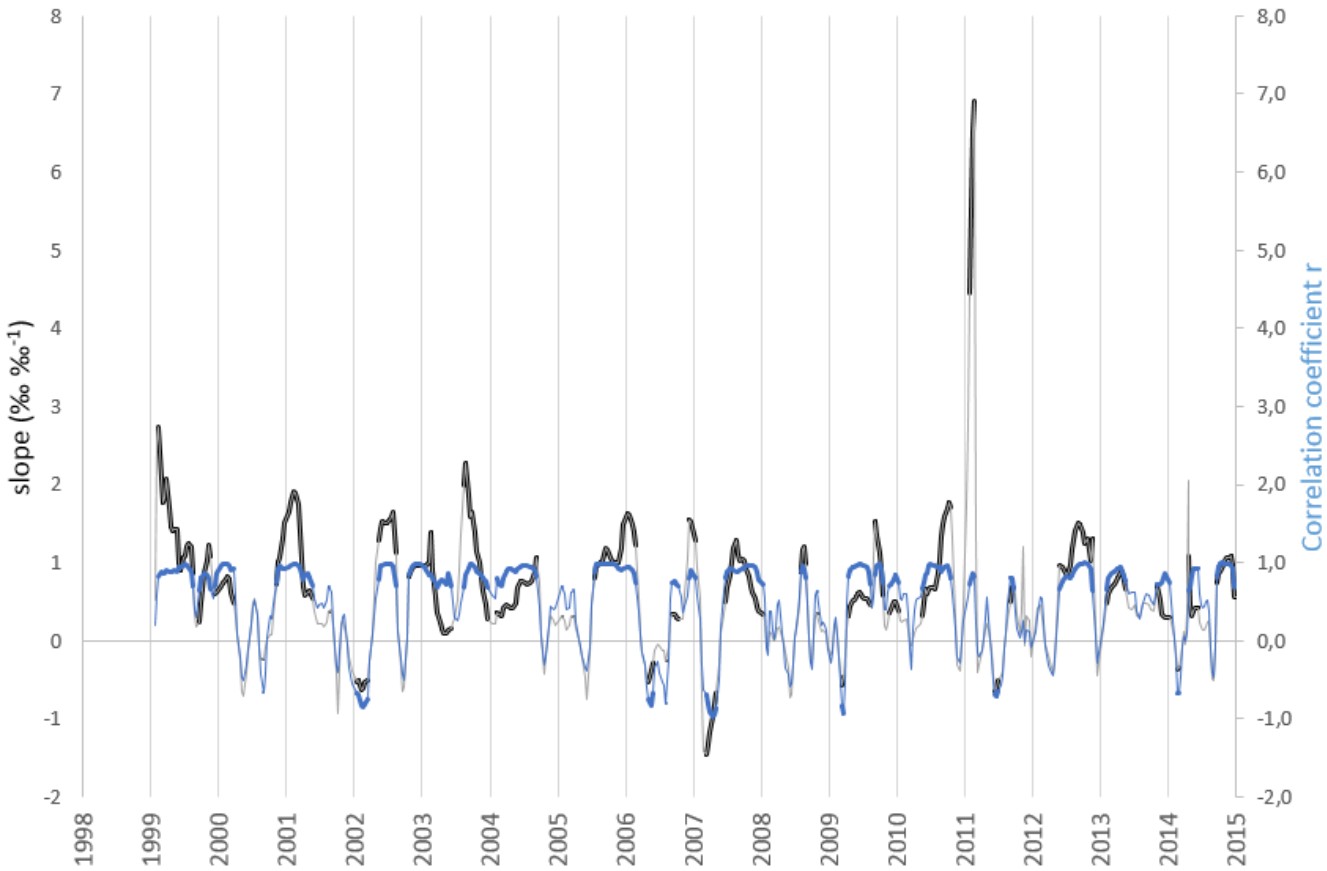

**Figure 11: 10-point running slope ((‰ ‰$^{-1}$) and running correlation coefficient calculated between d-excess versus δ$^{18}$O calculated from the raw data of TA192A. Only significant results are reported (p<0.05). The date associated with the results correspond to the first point of the regression calculation (applied on 10 points).**