# Peer review of "Challenges associated with the climatic interpretation of water stable isotope records from a highly resolved firn core from Adélie Land, coastal Antarctica"

_The Cryosphere, 2018_

## Referee Comment (RC1) · E. Thomas (Referee) · 9 Aug 2018

The paper provides a detailed and thorough examination of stable water isotopes and snow accumulation and chemistry from a shallow (21.3m) ice core in coastal Adelie Land. The record covers a relatively short time (1998-2014), but demonstrates the importance of robust proxy evaluation, particularly at coastal Antarctic sites.

I have a few minor comments and suggestions:

P2 Abstract – There is quite a bit of detail about the method in the abstract, which I felt

was not necessary. Removing lines 7-13 will keep the abstract concise and focus more on your findings. P2 Line 16 - End sentence after . . ..timescale."

Introduction The recent PAGES 2k compilation of ice core snow accumulation data (CP, 2017) highlighted the importance of coastal records.

There was no mention of the Antarctic Peninsula in your summary of drilling efforts at coastal sites? Several shallow, low elevation coastal cores have been drilled in this region.

Climatic interpretation of water stable isotopes records – I felt some reference to previous studies on the role that sea ice plays was missing. I suggest checking the following:

Noone, D., and I. Simmonds (2004), Sea ice control of water isotope transport to Antarctica and implications for ice core interpretation, J. Geophys. Res.,109, D07105, doi:10.1029/2003JD004228.

Bromwich, D. H. (1988), Snowfall at high southern latitudes, Rev. Geo-phys.,26, 149–168.

Bromwich, D. H., and C. J. Weaver (1983), Latitudinal displacement from main moisture source controld18O of snow in coastal Antarctica,Nature,301, 145–147.

P19 Line 23 – remove "does" P21 Line 12 "On average,.."

Discussion

The PAGES 2k compilation, based on ice cores and modelled SMB from RACMO2.3p2, identified a negative trend in SMB in neighbouring Victoria Land. The negative trend since the 1960s is statistically significant (p<0.01) and outside the expected range for the previous 200 years. A negative trend was also identified in the Wilkes Land coast (of which your Adelie land site would fit) which was outside the range of expected variability but not statistically significant.

P23 Line 5-7 Consider rewording . . ."we observe slightly increasing but not significant

trend in the TA core, era and ECHAM5-wiso records".

P24 Reference to other d18O temperature records. Poor correlations with temperature observed in the Antarctic Peninsula and coastal west Antarctica. For example, the relationship between d18O and temperature from ERA-interim at a coastal site in Ellsworth Land (Ferrigno) was 0.44 (p 0.01). At this site the relationship between d18O and temperature was similar to that of the relationship between d18O and sea ice conditions (r= -0.37 between d18O and winter sea ice extent and r=-0.54 between d18O and sea ice concentration).

P25 Line 3 – Should this line be a section heading? P25 Line 20 – Consider rewording – "As discussed earlier, this last item should be considered with caution due to potential numerical artefacts."

Chemistry

P28 Line 16 – There have been several studies suggesting that sea ice is a source of sea salt aerosols, therefore the presence of summer sea ice might not be associated with reduced Na? It might be worth checking or acknowledging these papers.

See Yang, X., Pyle, J. A., and Cox, R. A.: Sea salt aerosol production and bromine release: Role of snow on sea ice, Geophys. Res.Lett., 35, L16815, https://doi.org/10.1029/2008GL034536, 2008.

Rhodes, R. H., Yang, X., Wolff, E. W., McConnell, J. R., and Frey, M. M.: Sea ice as a source of sea salt aerosol to Greenland ice cores: a model-based study, Atmos. Chem. Phys., 17, 9417-9433, https://doi.org/10.5194/acp-17-9417-2017, 2017. (admittedly the later is for Greenland).

---

## Referee Comment (RC2) · B. D. Emanuelsson (Referee) · 29 Aug 2018

**Summary**

In this study, the authors examine the accumulation, isotopic ( $\delta$ 18O and d-excess) and chemical records (MSA, nssSO4) from a shallow firn core (TA192A; 66.78°S; 139.56°E, 602 m a.s.l.) from Adélie Land, Antarctica. The 21.3 m core was retrieved at a coastal high-accumulation site. Local meteorological station data, accumulation stake data, an isotope-enabled GCM and back-trajectory analysis are utilized to examine the signal

that is captured in the core. These highly-resolved records only cover the last two decades, coindiding with Interdecadal Pacific Oscillation's recent negative phase, but provide a rare opportunity to examine climate dynamics from the East Antarctic sector. A sector that in general suffers from a dearth of observations and where historically recovered cores, in general, are from low accumulation (low resolution) sites from the East Antarctic plateau.

I recommend that this study will be accepted but with major revisions.

Comments to the authors

Major comments:

The TA age scale

There is a significant correlation between ECHAM5-wiso  $\delta$ 18O and DDU SAT, but the correlation between TA  $\delta$ 18O and DDU SAT is insignificant. As the lack of correlation consistently appears to be associated with the TA isotope records one can suspect that an error in the age scale introduces an offset between the TA isotopes and station data, and between the TA isotopes and the ECHAM5-wiso data.

"No significant isotope [d18O or d-excess]-temperature relationship can be evidenced at any timescale, ruling out a simple interpretation of in terms of local temperature." (Abstract L.15). "ECHAM5-wiso produces a significant relationship between  $\delta$ 18O and temperature, a feature which is not identified in the TA record." (P22 L17).

I encourage, the authors to investigate if the lack of significant relationship for the TA isotope records with SAT station data, with ECHAM5-wiso SAT and with simulated isotopes can be due to an error in the TA age-scale. The age-scale can be susceptible towards this type of error as there appears to be a lack of age constrains for the core. If one annual layer count is missing this would result in isotopic data being assigned climate data that is one year or more too current. For example, looking at one single year, if a layer-count is missing, values that are truly 2011 would be assigned as 2012

TCD
and thus aligned with too contemporary climate data. These concerns prompted me to try to understand the TA isotopic data, that is, to check if there is an offset associated with the age scale. I created a new depth-age relationship, which includes one more annual count. You didn't include this layer, but you had flagged it as ambiguous ( $\sim$ 0.68 m depth, dashed black line in Fig. 3).

**Correlation maps**

I correlated the TA isotopic data ( $\delta$ 18O and d-excess; estimated from Fig. 7) with ECMWF ERA-Interim (Dee et al. 2011) geopotential 500-hPa (z500), surface air temperature (SAT), and meridional (v850) and zonal (u850) winds, as well as with HadISST sea ice concentration (SIC) (Rayner et al. 2003). These correlations were done both with the original and updated new depth-age relationship. I made plots where the upper set of panels show the modified age scale and the lower panels shows the original (Figs. R1–6).

Here is why I think the updated age-depth relationship, with the  ${\sim}0.68$  m layer count, supersede the original age-scale.

\*The nssSO4 peak is in the 100 ppb range and the nssSO4 peak is associated with a well-defined  $\delta$ 180 peak.

\*The correlation maps between  $\delta$ 18O and the monthly reanalysis fields display higher significance when the updated age-depth relationship is used (Figs. R1–6). (This is a somewhat circular argument, but it provides enough of an indication that the agescale needs to be better constrained.) The higher significance for the updated agedepth relationship is particularly clear for sea ice (Fig. R3). Figure R3a indicate that a negative sea-ice anomaly towards the Southern Ocean in the 120°E–150°W sector and off the coast of Princes Elizabeth Land (65°S, 75°E) is associated with a positive  $\delta$ 18O anomaly. Note that no significance pattern appears in these sectors when the original age-scale is used (Fig. R3b). It is well-known that water isotope records from coastal locations are significantly influenced by regional sea ice conditions (Noone and
Simmonds 2004; Küttel et al. 2012). Sea-ice advance and retreat are affected by (and interact with) large-scale atmospheric circulation modes.

\*The significant pattern with the updated relationship indicates a contemporaneous, in sync, relationship between  $\delta$ 18O and the monthly reanalysis fields. Positive  $\delta$ 18O anomalies tend to be associated with a low-pressure system (clockwise rotation) centered over Tasmania, Australia. The low appears to conduit wind and warm air south (Figs. R1a and R2a). Onshore meridional winds (negative v850) are associated with positive  $\delta$ 18O anomalies, see green shading close to the site. Vapor transport associated with this path tend to be linked with a positive  $\delta$ 18O anomaly as the air comes from a warm open water source region. This is a common correlation pattern for coastal Antarctic ice-core sites (Abram et al. 2011; Thomas et al. 2013). However, commonly with a larger significant region compared to the pattern shown in Fig. R2a. Note that the SAT pattern is significant just off the coast of the site (Fig. R1b), however, only over a limited extent. The v850-wind anomaly close to the site provides an indication that the age-scale and climate data is in sync as the anomaly occur in the atmosphere and thus isn't associated with long memory effects (lead/lags), as can be the case for sea ice and SST. If onshore winds drive warm ocean air toward East Antarctica (positive TA  $\delta$ 18O anomaly) the warm air can potentially linger and cause SAT anomaly on the plateau the following year. Thus when the reanalysis data is misaligned with the isotope data, a significant positive SAT anomaly pattern can appear over the plateau (Fig. R1d). (Alternatively, if the original age scale is correct, you didn't see this positive anomaly over the plateau as you used the DDU station SAT time series instead of reanalysis fields.)

The next set of figures shows d-excess correlation pattern with z500, 2mT, v850, u850 and SIC (Figs. R4–6). Positive d-excess anomalies is associated with anti-cyclone at  $\sim$ 55°S and 90°E (Fig. R4a) this high force winds along the coast towards the site (Fig. R5b). There is a similar pattern of significant zonal easterly winds that approach the site from the Ross Sea side (Fig. R5b). These easterlies appear to be associated
with a weak high over the Ross Ice Shelf and a low-pressure band to the north of the shelf. The easterlies appear to bring vapor to the site from the eastern Ross Sea polynya and sea-ice margin (Fig. R6a). The westerlies associated with high off the East Antarctic coast (~65°S, 90°E) appears to provide a similar conduit of vapor from polynya region (Fig. R6a). The d-excess signal originating from these regions can be anomalously positive as evaporation occur at a high-latitude low RH setting and the transport path towards the site can be associated with kinetic distillation processes as the air parcels are advected over ice, which further increase the positive d-excess anomaly. The cold atmospheric temperatures in combination with the long distance travelled by the air parcel over ice (without re-enrichment from an ocean source), can allow for a greater expression of the Rayleigh distillation and temperature dependence of kinetic effects, which is associated with higher d-excess values (Jouzel and Merlivat 1984; Risi et al. 2013). This is also consistent with your results, namely, the anticorrelation between d-excess and SMB: as the air is advected over ice shelf and seaice it will have experience "rainout" before reaching the site and thus the high d-excess values will be associated with relatively dry conditions. The setting with a positive SIC trend in the 110°E–160°W sector (Fig. R7) can also be important for the d-excess trend as it provide a setting with anomalously northerly sea-ice margin where air can be advected along the coast without contact with an ocean source.

Note that both the eastern Ross Sea and the sea-ice region off the coast of East Antarctica ( $\sim$ 55°S, 90°E) have seen a significant negative sea-ice trend over the examined 1997–2014 period (Fig. R7). This is particularly clear for the eastern Ross Sea. The negative eastern Ross Sea sea-ice trend, therefore, provides a compelling explanation to the positive d-excess trend. I think it could be interesting if you could show whether these transport pathways exist. You can do this by showing the paths associated with the back-trajectories, perhaps as clusters. Note that air originating from these seaice regions doesn't necessarily have to have seen a significant increase, that is the transport path can always have been there it could be that the polynyas just recently became active. However, your findings that back-trajectories from the WAIS+Pacific
region have increased is consistent with a positive d-excess trend, assuming that the air pass over Ross Sea sea ice or shelf prior to reaching the site.

The examined interval, 1997–2014, is short and almost exclusively coincide with the Interdecadal Pacific Oscillation's recent negative phase. Thus, it should be cautioned that your results and these correlation plots might look different if the examination period had been longer. I suggest that you show correlation maps similar to those that I introduced. Local conditions like those presented in Figure 6 in the paper does not necessarily show any influence on the isotopic record. For example correlation plot with reanalysis z500 fields can provide you with information about the pressure systems that are associated with air advection to the site and the isotopic signature. A pressure time series from a nearby station will not provide you with this information. Thus I suggest that you replace Figure 6 with correlation maps. Note that it can also be fruitful to correlate d-excess with SST and RH reanalysis fields, to check if any region stands out. I can share my Matlab analysis code for the correlation maps if that would be helpful? Contact the editor if you would like to get a copy.

The dating process of this core seems to be challenging. However, by not providing any reanalysis correlation maps or investigate the records relationship with standard indices (such as for ENSO and SAM), it feels like you have not exhausted the standard methods used to interpret the isotopic signature of an ice core. Combining the correlation maps with the back-trajectory analysis can be a powerful approach to test the d-excess hypothesis and thus aid in addressing the goals of the study (P7 L7). I don't think it is justified to call the research challenging (referring to the article title) before you have tried additional analysis. The record you present here is interesting so it also deserves a more exciting title.

The introduced update of the age-depth relationship, should not be viewed as a permanent update. Instead, it was meant to raise a concern. It would be good if you can find some age constrains for the core. The data for the closest stake needs to be included in Fig. 3 if it is central for the age-scale. I like the age-scale section otherwise
it has enough detail to ensure reproducibility. Are there any other chemical records from the core, any other shallow cores or pits (that can provide a historic snow surface), or can you use the Acoustic Depth Gauge (ADG) record of snow surface height from the D-10 station? In your 2017 study, you used NH4 and Na, but they might not add any additional information? You couldn't match the isotopes with the S1C1 record, but is it possible with any other records? This might be helpful if the S1C1 age-scale is associated with less uncertainty. In the future, can there be something to gain in terms of resolution to measure these cores on a continuous-flow analysis (CFA) setup or increase the sample resolution of the discrete samples? We were fortunate on the project that I worked on researchers had visited there in the 1970s and retrieved cores. From their pioneering research, we got additional age-constrains from their snow surface at the time, plus beta-counts and we were able to match our isotope record with theirs (Emanuelsson 2016; Winstrup et al. 2017).

**Age-scale thresholds**

The introduced threshold for your annual counts does not capture the older/deeper part of the record. Note how the picks at  $\sim$ 12.2 m,  $\sim$ 11.2 m, and  $\sim$ 10.5 m (currently assign for the year 1999, 2000, and 2001) does not exceed the 100 ppb nssSO4 threshold. There, of course, has to be layer counts for this section, but they are not strictly speaking captured by the introduced rules. Perhaps this can be linked to the low proportion of air parcels that come from the Indian sector at the beginning of the record? You could might thus get less of an ocean signature. Please cite earlier studies that have successfully used nssSO4 and/or MSA for dating (e.g., Steig et al. 2005; Abram et al. 2013).

**Back-trajectories**

Show your results on maps too. Consider cluster analysis for the trajectories or it might suffice to show just the 5-day endpoints. The path can be important though, consider this case. The air is classified as WAIS+Pacific as the endpoint fall in this sector, but
the air advected from this sector can be re-enriched from an ocean source or polynya on its way to the site.

I also suggest that you split up the Indian sector, it covers a too vast region. In my mind, the region from  $0^{\circ}$ -90°E is likely to be associated with a different signature compared to direct onshore moisture transport. Sticklers might also object to that the Indian and WAIS+Pacific sectors intrude into the Atlantic Ocean sector. Also, why do you leave a region south of New Zealand undefined? Is it a concern that the start elevation of 1,000 m a.s.l. does not correspond to the elevation at the site (602 m a.s.l.). One of the findings in Goursaud et al. (2018a) was that the site elevation was important for the d18O-d-excess phase-lag.

"Finally, we associated each daily back-trajectory to daily precipitation  $\delta$ 18O and dexcess values simulated by ECHAM5-wiso in the precipitation, and classified the time series for each variables by back-trajectories sectors." (P22 L3). Did you report any of the results from the d-excess back-trajectory analysis? Is there a way to tag the backtrajectories with TA d-excess data? It would be neat, if you can tag the back-trajectories with monthly TA d-excess values and then show that high d-excess months tend to originate from (or pass over) the Ross Sea and the off the coast of East Antarctica (~65°S, 90°E) negative sea ice trend zones. To pinpoint these area, you might need a finer sector resolution and to also look at the path not only the 5-day back-trajectory endpoints.

**Post-depositional effects**

I think three years is a too short period. Your argument, as you pointed out, is assuming that there is no natural interannual or inter-decadal variability. What ratio do you get if you just split the records in two parts (8 yrs in each)? If you want look at intra-annual (seasonal or monthly) isotopic resolution I still think you would have to consider diffusion. However, I think your analysis and the correlation maps (Figs. R1–6) using annual values shows that you can also obtain interesting information from the annual records.
Note that one approach would be to forward-diffuse the isotope records (Vinther et al. 2010; Küttel et al. 2012). You can use the methodology in Küttel et al. (2012) and then compare the results from using the forward-diffused record and the untouched record to evaluate objectively if there is a difference.

**ECHAM5-wiso**

Could the S1C1core's age-scale be better constrained than the TA core? Beta-counts were used for the S1C1age-scale, which can help to reduce the uncertainty. The significant correlations for the S1C1 with the model data would indicate that this could be the case and the lack thereof for TA could indicate that TA's age-depth relationship is not as well constrained. This provides another indication that an annual-layer count might be missing for the TA age scale. These cores are located only 14 km apart, so they will be exposed to similar climatic conditions. It thus seems to be a hasty conclusion to attribute the lack of similarities for the accumulation records of the two core solely to S1C1. "the absence of similarity between the TA and the S1C1 accumulation reflects the uncertainty in the S1C1 dating resulting from the large spatial variability and from more frequent erosion processes occurring at the S1C1 site" (P24, L1).

For the same reason, I would caution you against being too strong with your criticism against the model simulations here. The analysis that you perform requires that the age-depth is well constrained, that is, that the isotope data is aligned with the GCM data year by year. (The seasonal cycle comparison should not be as sensitive though.)

**SMB**

Are you using a sufficient number of points (stakes and cores) to get conclusive SMB results? Perhaps it would be wiser to leave the SMB part for a later publication? For that study, you could include all the available stakes and cores. I suggest that you home in on one or two novel ideas. I also suggest that you work on focusing and shortening the paper, clarify sections, and work on presenting some findings in figures instead of the text. (I can also become better at this). I like your short summaries at the end of

TCD
each section and they are written in a clear concise way. Strive towards getting this level of clarity throughout. Suggestions: focus on the  $\delta$ 18O and d-excess TA data, interpretation of the signal (explain the d-excess trend), the isotopes relationship with the ECHAM5-wiso model, and back-trajectories. These findings seem novel, robust and interesting. Remove the snow surface sampling archive part, the SMB and the  $\delta$ 18O/d-excess signature analysis? You can still publish these findings elsewhere.

Specific comments:

I suggest that you add a subscript to  $\delta$ 18O and d-excess when you refer to ECHAM5wiso data. That way the reader doesn't have to look back in the text to check if it is the simulated or TA data that you refer to.

**Article title**

You can perhaps change the title. You have to show that this is true first, but here is my suggestion. "Recent positive trend in d-excess in an Adélie Land coastal ice core, Antarctica, is linked to the activation and increased distant advection from the eastern Ross Sea polynya".

Abstract

P1 L2. Provide the name of the core here too: Terre Adélie 192A (TA192A).

P1 L5. It is not necessary to include "hereafter" when you introduce a new acronym. Writing "…reconstructed surface mass balance (SMB)" should suffice. Change this throughout. In the abstract you might even consider avoiding the acronyms altogether to be brief.

P1 L15. Remove the space so it says: "isotope-temperature relationship"

P1 L20. Use another type of dash for intervals, that is, the en-dash: "135  $^{\circ}$ -145  $^{\circ}$ E" and "1998–2014". Change throughout.

Introduction
P3 L22. Change this sentence so it reads? ...are thus essential to provide continuous local to regional sub-annual resolution climate information spanning the last decades. You would ultimately be more interested in information about past climate conditions, not just a couple of decades back. Resent data is important for the interpretation of the isotopic signal as there is an overlap between the observational records and proxy records.

P3 L24. The SMB acronym has already been introduced on line 11.

P3 L27. The ITASE cores (Mayewski 2005; Steig et al. 2005) are not coastal (at least not all of them), unless you define everything that's not on the plateau as coastal. They are similar to coastal cores, however, in that they are exposed to synoptic conditions.

P4 L5. Change to "initiated" instead of "triggered"?

P5 L3. I would remove "until now". Otherwise it sounds like you have resolved this issue here once and for all in this study.

P5 L11. Under-documented, is not a common word (at least not according to google Ngram), replace? Can you say, "remains poorly documented and understood", instead?

P7 L19. Stick to one convention for units, so write "3.9 m s-1" here instead.

P6 L17. Remove "search" here and say something like, highlights the importance of retrieval of more

P6 L22. 139.56°E not S.

P7 L6. I would say resolution instead of scale here.

P7 L7. Remove ", and dated." And start a new sentence and say "these records were used to establish the age scale for the firn core.

P7 L10. Section references can be shortened, e.g., Sect. 2. Material and method

TCD
P7 L18. No need to write "(see Fig. x), (Fig. x) should be fine. Change throughout.

P8 L4. You write ", respectively" here but it's not clear what it refers back to. Rewrite so this sentence becomes clear.

- P8 L8. Write Picarro without all the caps.
- P8 L16. Use the DDU acronym.

P8 L18. When in 1959?

P8 L24. Change to "range" not "ridge". And change the dash symbol for the range.

P9 L3. Change to "... km from the TA drill site" instead.

P9 L10. Move Table 1 to the supporting material?

P9 L15. Change to "back-trajectories" to be consistent with the rest of the text

P9 L19. Change to "DDU intra-annual precipitation variability." instead.

P9 L20. If you use the 0.75°x0.75° grid data, wouldn't the grid point at 139.50°E be the closest point to the drill site, Not 140.25°E? Note that you can choose even higher resolution when downloading the data.

P9 L28. 1500 m a. s. l. not m. a. g. l.?

P10 L24. Change to "...same simulation as Goursaud et al....", instead of "...same simulation than Goursaud et al....".

P10 L26. Can you add one more year of ECHAM5-wiso output data to the analysis to match the analysis period of the core?

P10 L27. End the sentence with a period.

Results

P11 L7. Change to "multi-year study"?
P13 L5. "...obtain a ratio of 0.5 (see Section 3.3)". Did you want to reference the supplementary table again, Table S3? In the supplementary table, it also looks like the ratio for d-excess is 1.1, not 0.5. I think you are a bit too thorough when you write (in the Supplementary Material) for each entry. Write it one time and then after that Fig. S1 and Table S1 etc. should suffice.

P15 L5. You write nssSO4 in many different ways throughout the paper (and define the acronym at several places). I assume they are the same? Stick to one.

P15 L24. You mean p<0.05, not >0.05.

P15 L25. The significance level should be

P20 L14. and L18. I understand why you have put the table and figure together. However, it is probably better to present them separately and refer to them as (Fig S#) and (Table S#) in the text.

P20 L14. Indeed there is a large spike on 7 May. Nevertheless, there are also other anomalously positive events during 2007 that contribute to the annual anomaly.

P20 L20. Please clarify this sentence. This is a case where I think a subscript can be handy and a table to refer to. As you have simulated and TA d-excess and DDU and wiso 2mT, it is hard to keep them apart. Especially in a long sentence like this.

P20 L24. Please provide a p-value.

P21 L12. Change to "On average,".

P21 first paragraph. You say that the calculation are based on the 1998–2014 period, on the 2nd line, that should be enough, so you can remove the "over the period 1998–2014" text from line 13, 13–14, and 15 as it is redundant.

P22 L4. "and classified the time series for each variables by", Change to "variable" instead.

Discussion

P23 L3. "slightly increasing but not significant trend". I'm afraid that you can't call it a trend if it is not significant. If it has some significance, p

smoothing and running correlation?

P26 L13. "As a result, the  $\delta$ 18O-d-excess relationship may be a fingerprint of changes in air mass origins, and particularly of the occurrence of precipitation of air masses coming from the WAIS+Pacific sector". From visually inspecting Figure 7a (and from what you mention in the text) it seems like air originating from WAIS + Pacific and the Ross Sea sectors has increased. If your cite statement above would hold up, how would I see this in Figure 11? The frequency of occurrence of significant relationships (bold line) seems quite even throughout. 2007 and 2011 stand out as anomalies, but so does 1999, 2001, 2003. Or is 2007 more remarkable because it is a negative anomaly? Or do you mean that a spike in the slope signals an abrupt change in air origin, e.g. a sudden change from Indian to WAIS+Pacific? Are there any remarkable conditions associated with the year 1999, 2001 and 2003 too?

P26 L17. "... respectively (consistently with what obtained within each year, see...). "...-1, respectively (this is consistent with the annual means, see...). Does this sound better?

P27 L7. Change to "not only with stake data closest to the drill site".

P27 L11. "However, we cannot draw any conclusion..."

P27 L12. "(Unfortunately, there are no striking features during the records common period, which makes it challenging to match the isotope records)".

P27 L18. "These findings suggest".

P28 L2. "(from ice cores, snow precipitation, and water vapour)'

P28 L5. "We compare the chemical concentrations recorded in the TA firn core with S1C1core, for their common period (1998–2006). The mean concentrations are slightly lower for the..."

P28 L15. "Air originating from near the sea ice margin may contain relatively high d-
excess". In general, I think to say just "air" is fine, that is, the "mass" or "masses" part is often not needed.

P28 L17. "Such a configuration..." this sentence is confusing as it doesn't appear to correspond to the setting described above. The presiding sentence describes a low sea ice setting and the latter describes a positive sea ice scenario, so these sentences don't seem to correspond with one another. "Such a configuration should be associated with high sea-salt concentrations due to increased sea-salt emissions when the sea-ice concertation is low in summer."

P28 L25. Chang to "probably caused by marine air advection".

P28 L26. It might be better to avoid starting sentences with a conjunction ("And" here) in formal writing, as some people might object to it.

P29 L13. Change the highlighted parts of the sentence "...insure that small scale SMB random variability caused by presence of sastrugis, dunes and barchans is negligible..." to "ensure that small-scale SMB random variability caused by presence of sastrugi, dunes and barchans is negligible".

P28 L19. "In contrast, vapour formed over the ocean..."

P29 L1. Change to "To summarize," or "In summary," instead.

Conclusions and perspectives

P29 L11. Change to "The high estimated SMB rate of 74.1  $\pm$  14.1 cm w.e. y-1 limits the effects of diffusion and ensures that records with sub-annual resolution are preserved."

Figures

Change the figure texts so that not everything is in bold. Just keep the fig numbering (Fig. 1, etc) and the letter (a) indicators in bold. Change throughout this section. It is not necessary to write "in" when you specify units, e.g. (in ‰. Change throughout the document.

TCD
Fig. 2. Remove the  $45^{\circ}S$  boundary in the figure, otherwise, it looks like these regions have a northerly limit at this latitude.

Defining sectors:

\*It is confusing that you use both signs (-, +) and letters to indicate the hemispheres. Stick to just letter.

\*I also suggest that you write the intervals as ranges as you do otherwise. That way you can skip the "lat", "lon" text too, S/N and W/E provides you with this information. Example: Ross Sea sector:  $0^{\circ}-75^{\circ}$ S,  $180^{\circ}-240^{\circ}$ E. You write that you don't have any equatorward boundary. Assuming that you don't have any endpoint in the NH you can write this way.

\*Skip the dash in "Ross-Sea sector" as it is a name. I also don't think you need all of the quotation marks when you introduce the names, see if you can write this in a clear way without them.

\*Note also, that you wouldn't usually call a sector that extends so far into the Atlantic Ocean, Indian or WAIS+Pacific.

\*Apply this formatting guidance to where you define sectors elsewhere in the text too, e.g. P8 L25 and P10 L12.

Fig. 3. The  $\delta$ 18O record seems to be shown both as regular value and a smoothed record, but you don't tell the reader how the record has been smoothed. You reference Fig. 3 in the figure text, but this is Fig. 3. For clarity it would be better if you list what the figure show here and describe the methodology elsewhere.

Fig 6. Local sea-ice extent with a % that is in the 124–134% range, this looks odd to me? Fig. S6 should be the same just with the standard deviation too. However, note here how the range of sea ice concertation is different. You write sea ice concentration in the figure text and sea ice extent on the figure axis, please clarify. All that said, I rather see that you scrap this figure and replace it with correlation maps.
Fig. 7. Mention the panels in order, that is, accumulation occurs last, but is mentioned first in the text.

Fig. 9. A letter is missing text inside the figure, it should say "Plateau". Can you increase the length of the bars (a, b) so it will be easier to see differences between the regions? You could also consider splitting a, c and b, c into two separate figures and align a, c so that the years in the panels correspond to each other, that is so that 1998 and 2014 is aligned in a and c. And that the moths are aligned in a similar manner in b, d.

Fig. 11. Maybe start the sentence with "Ten" instead of 10. There is also one parenthesis bracket too many.

Sentence 2. Change to "Significant results are indicated by thick lines (p

one entry start or ends otherwise.

The Cavalieri et al. 1996 reference does not appear in the Reference list.

References

Abram NJ, Mulvaney R, Arrowsmith C (2011) Environmental signals in a highly resolved ice core from James Ross Island, Antarctica. J Geophys Res Atmos 116:1–15. doi: 10.1029/2011JD016147

Abram NJ, Wolff EW, Curran MAJ (2013) A review of sea ice proxy information from polar ice cores. Quat Sci Rev 79:168–183. doi: http://dx.doi.org/10.1016/j.quascirev.2013.01.011

Dee DP, Uppala SM, Simmons AJ, et al (2011) The ERA-Interim reanalysis: configuration and performance of the data assimilation system. Q J R Meteorol Soc 137:553– 597. doi: 10.1002/qj.828

Emanuelsson D (2016) High-Resolution Water Stable Isotope Ice-Core Record: Roosevelt Island, Antarctica: a thesis submitted to the Victoria University of Wellington in fulfilment of the requirements for the degree of Doctor of Philosophy (Geology) / by B. Daniel Emanuelsson. Thesis (Ph.D.)–Victoria University of Wellington, 2016.

Goursaud S, Masson-Delmotte V, Favier V, et al (2018a) Water stable isotope spatiotemporal variability in Antarctica in 1960–2013: observations and simulations from the ECHAM5-wiso atmospheric general circulation model. Clim Past 14:923–946. doi: 10.5194/cp-14-923-2018

Goursaud S, Masson-Delmotte V, Favier V, et al (2018b) Challenges associated with the climatic interpretation of water stable isotope records from a highly resolved firn core from Adélie Land, coastal Antarctica. Cryosph Discuss 2018:1–55. doi: 10.5194/tc-2018-121

Jouzel J, Merlivat L (1984) Deuterium and oxygen 18 in precipitation: Modeling of the
isotopic effects during snow formation. J Geophys Res Atmos 89:11749–11757. doi: 10.1029/JD089iD07p11749

Küttel M, Steig EJ, Ding Q, et al (2012) Seasonal climate information preserved in West Antarctic ice core water isotopes: relationships to temperature, large-scale circulation, and sea ice. Clim Dyn 39:1841–1857. doi: 10.1007/s00382-012-1460-7

Mayewski PA (2005) The International Trans-Antarctic Scientific Expedition (ITASE): An overview. Ann Glaciol 41:180–185.

Noone D, Simmonds I (2004) Sea ice control of water isotope transport to Antarctica and implications for ice core interpretation. J Geophys Res 109:1–13. doi: 10.1029/2003JD004228 Rayner NA, Parker DE, Horton EB, et al (2003) Global analyses of sea surface temperature, sea ice, and night marine air temperature since the late nineteenth century. J Geophys Res Atmos 108:1–37. doi: 10.1029/2002JD002670

Risi C, Landais A, Winkler R, Vimeux F (2013) Can we determine what controls the spatio-temporal distribution of d-excess and 17O-excess in precipitation using the LMDZ general circulation model? Clim Past 9:2173–2193. doi: 10.5194/cp-9-2173-2013

Steig EJ, Mayewski P a, Dixon D a, et al (2005) High-resolution ice cores from USI-TASE (West Antarctica): development and validation of chronologies and determination of precision and accuracy. Ann Glaciol Vol 41, 2005 41:77–84.

Thomas ER, Bracegirdle TJ, Turner J, Wolff EW (2013) A 308 year record of climate variability in West Antarctica. Geophys Res Lett 40:5492–5496. doi: 10.1002/2013GL057782

Vinther BM, Jones PD, Briffa KR, et al (2010) Climatic signals in multiple highly resolved stable isotope records from Greenland. Quat Sci Rev 29:522–538. doi: 10.1016/j.quascirev.2009.11.002

Winstrup M, Vallelonga PT, Kjær H a., et al (2017) A 2700-year timescale and ac-
cumulation reconstruction for Roosevelt Island, West Antarctica. Clim Past Discuss submitted.

**Figures**

Figure R1. Correlation maps between TA  $\delta$ 18O and annual-averaged ECMWF ERA-Interim (Dee et al. 2011) (a, c) z500, and (b, d) SAT. (Upper panels; a, b) show the correlation for the 1997–2014 period using the updated "New" age-depth relationship and (lower panels, c, d) shows the correlation for the 1998–2014 period using the original age-scale. Shading shows correlation coefficients. Black contours enclose regions where the correlation coefficients are significant at the p

**from Goursaud et al. (2018b) Fig.7 for review purposes only.**

Figure R4. Correlation maps between TA d-excess and annual-averaged ECMWF ERA-Interim (Dee et al. 2011) (a, d) z500, (b, e) SAT. (Upper panels; a, b) show correlation maps for the 1997–2014 period using the updated "New" age-depth relationship and the (lower panels; c, d) show correlation maps for the 1998–2014 period using the original age-scale. Shading shows correlation coefficients. Black contours enclose regions where the correlation coefficients are significant at the p

0.05 level.
TA New r( $\delta^{18}$ O, z500) annual ERA-Interim 1997 - 2014

**TA New r( $\delta^{18}$ O, 2mT) annual ERA-Interim 1997 - 2014**

---

## Author Comment (AC1) · 17 Dec 2018

Reply to the editor's comments

Summary In this study, the authors examine the accumulation, isotopic ($\delta$18O and d-excess) and chemical records (MSA, nssSO4) from a shallow firn core (TA192A; 66.78°S; 139.56°E, 602 m a.s.l.) from Adélie Land, Antarctica. The 21.3 m core was retrieved at a coastal high-accumulation site. Local meteorological station data, accumulation stake data, an isotope-enabled GCM and back-trajectory analysis are utilized

to examine the signal that is captured in the core. These highly-resolved records only cover the last two decades, coinciding with Interdecadal Pacific Oscillation's recent negative phase, but provide a rare opportunity to examine climate dynamics from the East Antarctic sector. A sector that in general suffers from a death of observations and where historically recovered cores, in general, are from low accumulation (low resolution) sites from the East Antarctic plateau. I recommend that this study will be accepted but with major revisions. Dear Dr Emanuelsson, many thanks for reviewing and recommending the acceptation of our manuscript, and for all the comments you brought, which will undoubtedly help us to improve our work and make it more robust.

Comments to the authors Major comments:

The TA age scale There is a significant correlation between ECHAM5-wiso $\delta$18O and DDU SAT, but the correlation between TA $\delta$18O and DDU SAT is insignificant. As the lack of correlation consistently appears to be associated with the TA isotope records one can suspect that an error in the age scale introduces an offset between the TA isotopes and station data, and between the TA isotopes and the ECHAM5-wiso data. "No significant isotope [d18O or d-excess]-temperature relationship can be evidenced at any timescale, ruling out a simple interpretation of in terms of local temperature." (Abstract L.15). "ECHAM5-wiso produces a significant relationship between $\delta$18O and temperature, a feature which is not identified in the TA record." (P22 L17). We made a mistake when writing that ECHAM5-wiso produces a significant relationship between $\delta$18O and the temperature (p22), whereas we had written in the manuscript p21 l5: "Neither t2mECH – $\delta$18OECH, nor SMBECH – d-excessECH, nor $\delta$18OECH – d-excessECH relationships are identified in ECHAM5-wiso seasonal or annual outputs.". We thus removed this sentence and replaced it: "Similarly than in the TA firn core, ECHAM5-wiso simulates no $\delta$18OECH – t2mECH correlation, but also no d-excessECH – $\delta$18OECH."

I encourage, the authors to investigate if the lack of significant relationship for the TA isotope records with SAT station data, with ECHAM5-wiso SAT and with simulated

isotopes can be due to an error in the TA age-scale. The age-scale can be susceptible towards this type of error as there appears to be a lack of age constrains for the core. If one annual layer count is missing this would result in isotopic data being assigned climate data that is one year or more too current. For example, looking at one single year, if a layer-count is missing, values that are truly 2011 would be assigned as 2012 and thus aligned with too contemporary climate data. These concerns prompted me to try to understand the TA isotopic data, that is, to check if there is an offset associated with the age scale. I created a new depth-age relationship, which includes one more annual count. You didn't include this layer, but you had flagged it as ambiguous (0.68 m depth, dashed black line in Fig. 3). In our previous paper related to the S1C1 firn core (Goursaud et al., 2017, see Figure 3 attached to this response), we had highlighted the difficulty to date a firn core from Adélie Land based on isotope data only, and thus the necessity of coupling isotope data to chemical species for an annual layer dating. 25 out of the 60 annual $\delta$18O peaks were not identified when compared to chemical summer annual peaks. This absence of isotope annual peak can be due to deposition processes of snow, like drifting snow (e.g. Grazioli et al., 2017), and isotope post-deposition processes (e.g. Jones et al., 2017), for which deposited aerosols are not prone. Moreover, changes in seasonal precipitation and the origin of moisture can result in a different $\delta$18O seasonal cycle than the classical one (displaying a summer peak), associated to distillation processes only. When investigating the ECHAM5-wiso model for the grid point corresponding to the S1C1 firn core (which also corresponds to the TA192A firn core), we noticed that simulated water stable isotopes displayed a high interannual variability of the seasonal cycle (Table S3, Goursaud et al., 2017). For the period 1998-2014, the ECHAM5-wiso model simulates 9 years with maximum values out of the summer time, with 2 in March and 2 in August (Table 1 in the Supplement attached to this response), suggesting that atmospheric dynamic processes are behind this high variability. We thus based our annual layer counting on nssSO4,summer (Table S3, Goursaud et al., 2017), supported by MSA. But as shown in Figure 3, there were 3 uncertain years. We had no absolute horizons for this firn core (neither nuclear

test, nor volcanic horizons). However, stake data measured by the French national observatory GLACIOCLIM were a tool to adjust our dating by comparing it with our reconstructed accumulation. These data were found to be very reliable, as punctual stake data recorded similar interannual variabilities than the mean of the 156 km line data with coefficient correlations equal or higher than 0.8. We initially had 4 uncertain summer layers, the three remaining shown by dashed vertical black line in Figure 3, and the layer we included associated to year 2008 in our resulting dating, that allowed us to obtain the best correlation between our reconstructed accumulation and the stake data. To support our dating, we tested here the three remaining uncertain layers that we numbered in Figure 1 (attached to this response). The layer you pointed, is numbered as "1" on the Figure, and the two deeper ones along the firn are numbered "2" and "3". None of these three layers fill the conditions to count it (Table 2 in the Supplementary attached to this response), that are (i) nssSO4,summer > 200 ppb, or (ii) 100 ppb < nssSO4,summer< 200 ppb and a peak for MSA. An uncertainty is highlighted when one of the condition is close to be filled, or when a peak in $\delta$18O coincides with a peak in nssSO4,summer or MSA, or both. Note that nssSO4,summer peaks at a value of 76.2 ppb at layer "1", thus below the threshold we fixed to count it as an annual summer peak. Our first main constrain was to correlate the stake data which cover the period 2004-2014. Only layer "1" could make a change compared to our submitted dating, as other layers are at depths corresponding to a time before 2004. However, a new reconstruction including this layer gives no correlation with the stake data (r = 0.2 and p-value = 0.16 for the stake data "19.2", r = -0.02 and p-value = -0.06 for the 156 km stake line). We thus excluded it from our annual layer counting. Nevertheless, we simulated, out of curiosity, linear regressions with the ECHAM5-wiso model and find no correlation for the accumulation (r = 0.1 and p-value = 0.7), and a weak correlation for $\delta$18O at the annual scale (r = 0.5 and p-value = 0.045) and at the monthly scale (using 12-points resampling for the firn core record, r = 0.18 and p-value = 0.008). The second method we proposed to refine our dating, was to adjust our dating to improve the correlations between our reconstruction and the simulations (accumulation and

$\delta$18O). Table 3 and 4 in the Supplementary (attached to this response) display the results for the dating from the submitted version, and new tests including layers "2" ("test2"), "3" ("test3"), as well as both "2" and "3" ("test23"). For the accumulation, the simulated regressions widely differ, with correlation coefficients close to 0 for test2 and test23, increasing to 0.36 for test3 and to 0.45 for the original version. To the opposite, the simulated regressions for $\delta$18O are very close with a mean correlation coefficient of 0.26 $\pm$ 0.07. As a result, the correlations between the different dating and the ECHAM5-wiso simulations advocate for our initial dating. At this stage, we thus decided that our initial dating was the most consistent. We added the results of these tests in the Supplementary Material of the manuscript and mentioned it p12 l18-21: "We note an uncertainty in layer counting of 3 years when comparing the outcome of layer counting using chemical records with $\delta$18OTA peaks, which have nonetheless been excluded from our dating as they do not improve the correlations neither between the reconstructed SMB and the stake data, nor between our records and the ECHAM5-wiso simulations (Tables S2 to S4 in the Supplementary Material)." Although these new tests lead us to the conclusion that our dating is the most robust, we looked at the relationship between $\delta$18O recorded in the TA192A firn core ($\delta$18OTA192A) with the near-surface temperature measured at Dumont d'Urville (TDDU, Table 5 in the Supplement attached to this response), and with the 2-meter temperature simulated by the ECHAM5-wiso model (TECH, Table 6 in the Supplement attached to this response). These pieces of information do not have the purpose to call into question our dating, but to test the isotopic thermometer (e.g. the thermodynamic response of the $\delta$18O). We notice that whatever the dating test, the $\delta$18OTA192A-TDDU and $\delta$18OTA192A-TECH remain insignificant (p-value $\geq$0.05), emphasizing our hypothesis that processes out of thermodynamic act on Adélie Land $\delta$18O.

Correlation maps I correlated the TA isotopic data ($\delta$18O and d-excess; estimated from Fig. 7) with ECMWF ERA-Interim (Dee et al. 2011) geopotential 500-hPa (z500), sur-face air temperature (SAT), and meridional (v850) and zonal (u850) winds, as well as with HadISST sea ice concentration (SIC) (Rayner et al. 2003). These correlations

were done both with the original and updated new depth-age relationship. I made plots where the up- per set of panels show the modified age scale and the lower panels shows the original (Figs. R1–6). Many thanks for making such a work, and suggesting the use of correlation maps with the reanalyses to help interpreting the data. Although we did not go for including the 0.54 m layer, and thus the interpretation you made out of this dating, we tried to use the correlation maps to make an interpretation of our Adélie Land isotopic data. We thus first introduced the use of ERAinterim outputs in this arm, completing its description in section 2.2 ("Datasets"), p9 l27: "We also extracted 2-meter temperature (2mT), 10-meter u and v wind components (u10 and v10), and the geopotential height at 500 hPa (z500) over the whole southern hemisphere (50 – 90 °S), in order to investigate potential linear relationships between our records and the large-scale climate variability." We then described the maps in the results section 3.5 ("Influence of synoptic weather on TA records: insights from ECHAM5-wiso simulation, ERA-interim reanalyses, back-trajectories and modes of variability"), adding the question this analysis allows us to answer beforehand, p20 l25: (iv) "Are there significant relationships between our isotopic records and the large-scale climatic variability?". Only maps depicting significant correlations were described in a proper paragraph, p22 l13: "Relationships with the large-scale climatic variability The ERA-interim outputs allow us to investigate whether the large-scale climatic variability influence the isotopic composition of Adélie Land precipitation recorded in the TA firn core. We looked at the simulated linear relationships between the TA isotopic records ($\delta$18OTA and d-excessTA) with the ERA-interim outputs (2mT, u10, v10 and z500, Section 2.2). We here report only significant relationships with absolute correlation coefficients higher than 0.6. For $\delta$18OTA, we found a correlation with 2m-T over the Antarctic plateau (Fig.12a), as well as a correlation with v10 (Fig. 12b) along the westerly wind belt, at $\sim$ (55 °S, 100 °E) and $\sim$ (55 °S, 130 °E) in the Indian Ocean, at $\sim$ (55 °S, 10-50 °E) in the Atlantic Ocean, and a very little area on coastal Dronning Maud Land at $\sim$ (60 °S, 30-40 °E). For d-excessTA, we found a correlation with 2m-T (Fig. 12c) toward the Lambert Glacier at $\sim$ (70-80 °S, 30-40 °E), and an anticorrelation in the south of the Peninsula at $\sim$ (55-65 °S, 250-300

°E). Finally, we noted a correlation between d-excessTA and u10 (Fig. 12d) at a very narrow area of Dronning Maud Land at ∼ (80 °S, 10-20 °E), and an anticorrelation on the westerly wind belt in the Atlantic Ocean at ∼ (55 °S, 40-50 °E). No correlation is found with z500, neither with $\delta$18OTA nor d-excessTA." Finally, interpretations were proposed in the discussion part, first suggesting that the near-surface temperature of the plateau is linked to the $\delta$18OTA (Section 4.2 The $\delta$18O – temperature relationship in coastal Antarctic regions), hypothesis that we strengthened by the significant correlation $\delta$18OTA and the near-surface temperature measured at Dome C over the period covered by the TA core (1998-2014). We used the significant correlation observed between $\delta$18OTA and the 10-meter u wind component to suggest that dynamic processes partly drive the water isotopic composition of Adélie Land, p27 l5: "The $\delta$18O measured in the ice of coastal Adélie Land may thus not allow to reconstruct surface temperatures of this region. However, correlations between $\delta$18OTA and the 2m-temperature by ERA-interim over each grid point of Antarctica (Fig.12a) show significant relationships over the plateau, confirmed by a significant correlation between annual $\delta$18OTA and the near-surface temperature measured at Dome C over the period 1998-2014 (slope of 0.70 $\pm$ 0.29 ‰ °C-1, r=0.53, p<0.05). These results support previous studies suggesting warm intrusions offshore Dumont d'Urville towards Dome C (Naithani et al., 2002). Finally, the significant linear relationships with the u10 wind component above the westerly wind belt and at some coastal Antarctic area (Fig 12b) stress the influence of other processes than thermodynamic drive the isotopic composition of Adélie Land precipitations." Second, we extended our interpretation, using correlations with the second order term, the d-excess, in the following paragraph (Section 4.2 Water stable isotope, a fingerprint of changes in air mass origins). Especially, the correlation maps between d-excessTA and ERA-interim 2-meter temperature and 10-meter u wind component came to support the conjecture that the d-excess signature analyzed in the TA firn core is particular when corresponding to air masses coming from the western sector of Antarctica, p28 l11: "Linear relationships between d-excessTA and ERA-interim outputs come to strengthen the link between the climate variability

of western Antarctic and associated southern oceans, as we note an anticorrelation between d-excessTA and the 2m-T in the south of the Peninsula, the Ellsworth region and the Bellingshausen sea (r>0.6); and an anticorrelation between d-excessTA and the u10 wind component over the coastal Ross sector, consistent with the suggestion of air coming from western Antarctica towards Adélie Land via the Ross sector."

Here is why I think the updated age-depth relationship, with the 0.68 m layer count, supersede the original age-scale. *The nssSO4 peak is in the 100 ppb range and the nssSO4 peak is associated with a well-defined $\delta$18O peak. Please refer to our previous paragraph which shows the inconsistency of taking this peak into account for our dating.

*The correlation maps between $\delta$18O and the monthly reanalysis fields display higher significance when the updated age-depth relationship is used (Figs. R1–6). (This is a somewhat circular argument, but it provides enough of an indication that the age- scale needs to be better constrained.) We agree that our age-scale would need to be better constrained, at least over the period prior to 2004. Notwithstanding, we have no other tools at our knowledge.

The higher significance for the updated age- depth relationship is particularly clear for sea ice (Fig. R3). Figure R3a indicate that a negative sea-ice anomaly towards the Southern Ocean in the 120◦E–150◦W sector and off the coast of Princes Elizabeth Land (65◦S, 75◦E) is associated with a positive $\delta$18O anomaly. Note that no significance pattern appears in these sectors when the original age-scale is used (Fig. R3b). It is well-known that water isotope records from coastal locations are significantly influenced by regional sea ice conditions (Noone and Simmonds 2004; Küttel et al. 2012). Sea-ice advance and retreat are affected by (and interact with) large-scale atmospheric circulation modes. Indeed, Noone and Simmonds (2004) showed how changes in sea-ice, taken independently from other variables, can affect coastal $\delta$18O, and in a more complex way the d-excess. However, this study does not account for the interplay between sea-ice changes and other climate variables, like e.g. katabatic

winds, which also partly drive the atmospheric isotopic composition in Adélie Land, as shown by the recent vapor monitoring processed by Bréant et al. (submitted). In our submitted version, we used the sea-ice concentration averaged over specific longitudinal ranges. Also, we have been advised to rather consider the area where the sea ice concentration is higher than 15 %, what we took into account, as described in section 2.2, p9 l1: "D'Urville summer sea ice extent was estimated by extracting the number of grid points covering the area (50 – 90 °S, 135 – 145 °E) where the sea ice concentration is higher than 15 %, from December to January (included) for each year from 1998 to 2014." We just considered the d'Urville summer sea ice extent, and rather used map correlations to look at the larger scale. We reported our findings in the results section (removing our prior results), first in section 3.3 ("Inter-annual variations"), p16 l25: "Significant increasing trends are detected in the annual values of d-excessTA (0.11 ‰ y-1, r=0.61 and p<0.05) as well as of d'Urville summer sea ice extent (r=0.77, p<0.05)." and p17 l15: "Our record depicts a significant anti-correlation between annual values of SMBTA and d-excessTA (r=-0.59 and p<0.05), as well as a significant correlation between d-excessTA and d'Urville summer sea ice extent (r=0.65 and p<0.05)." With our new results, we also obtain particular high values in 2011 (instead of year 2013), as specified p18 l11: "Year 2011: high MSA, d'Urville summer sea ice extent, and wind speed values." The pattern of the mean seasonal cycle remains unchanged. We thus have changed our interpretation in the discussion part in section 4.3 ("Water stable isotope, a fingerprint of changes in air mass origins"), p28 l21: "Noone and Simmonds (2004) have shown, thanks to climate modelling, that water stable isotopes were conditioned by changes in sea ice extent (as a contraction in sea ice increases the local latent heat and temperature due to open water), but confirmed that a thorough understanding of main mechanisms controlling the d-excess was still needed. Also, earlier studies have suggested the use of the d-excess recorded in ice cores to reconstruct past sea ice extent (e.g. Sinclair et al., 2014). Although we find a significant correlation between the d-excessTA and the d'Urville summer sea ice extent (section 3.3), a correlation map between the annual d-excessTA and the summer sea ice concen-
tration (S16 in the Supplementary Material) show significant correlations with further sea ice areas (e.g. an anticorrelation in the Amundsen sea and correlations in the Belligshausen, Scotia and Lazarev seas)."

*The significant pattern with the updated relationship indicates a contemporaneous, in sync, relationship between $\delta$18O and the monthly reanalysis fields. Positive $\delta$18O anomalies tend to be associated with a low-pressure system (clockwise rotation) centered over Tasmania, Australia. The low appears to conduit wind and warm air south (Figs. R1a and R2a). Onshore meridional winds (negative v850) are associated with positive $\delta$18O anomalies, see green shading close to the site. Vapor transport associated with this path tend to be linked with a positive $\delta$18O anomaly as the air comes from a warm open water source region. This is a common correlation pattern for coastal Antarctic ice-core sites (Abram et al. 2011; Thomas et al. 2013). However, commonly with a larger significant region compared to the pattern shown in Fig. R2a. Note that the SAT pattern is significant just off the coast of the site (Fig. R1b), however, only over a limited extent. With our original dating (Figures R1c, R2c and R2d from your comments), no significant relationship appears between $\delta$18O and z500, or winds (u850 and v850). We thus cannot conclude on schemes of transportation of water vapor at large scale, using the pressure and wind fields simulated by the reanalyses.

The v850-wind anomaly close to the site provides an indication that the age-scale and climate data is in sync as the anomaly occur in the atmosphere and thus isn't associated with long memory effects (lead/lags), as can be the case for sea ice and SST. If onshore winds drive warm ocean air toward East Antarctica (positive TA $\delta$18O anomaly) the warm air can potentially linger and cause SAT anomaly on the plateau the following year. Thus when the reanalysis data is misaligned with the isotope data, a significant positive SAT anomaly pattern can appear over the plateau (Fig. R1d). (Alternatively, if the original age scale is correct, you didn't see this positive anomaly over the plateau as you used the DDU station SAT time series instead of reanalysis fields.) Although we do not observe correlations between $\delta$18OTA and v850 for our initial dating, we observe

a significant correlation between $\delta18$OTA, and the 2m-temperature from the reanalyses over the plateau, confirmed by a significant correlation between $\delta18$OTA and the near-surface temperature measured at Dome C (slope of 0.70 $\pm$ 0.29 ‰ °C-1, r=0.53, pvalue=0.03). This result comforts your idea that warm ocean air driven by winds offshore coastal Adélie Land towards the plateau, resulting in a significant SAT anomaly. This new result was added to the discussion part 4.2 ("$\delta18$O – temperature relationship in coastal Antarctic regions"), supported by a replication of your Figure R1d, p25 l17: "The $\delta18$O measured in the ice of coastal Adélie Land may thus not allow to reconstruct surface temperatures of this region. However, correlations between $\delta18$OTA and the 2m-temperature by ERA-interim over each grid point of Antarctica (Fig.12a) show significant relationships over the plateau, confirmed by a significant correlation between annual $\delta18$OTA and the near-surface temperature measured at Dome C over the period 1998-2014 (slope of 0.70 $\pm$ 0.29 ‰ °C-1, r=0.53, p<0.05). These results support previous studies suggesting warm intrusions offshore Dumont d'Urville towards Dome C (Naithani et al., 2002)."

The next set of figures shows d-excess correlation pattern with z500, 2mT, v850, u850 and SIC (Figs. R4–6). Positive d-excess anomalies is associated with anti-cyclone at 55◦S and 90◦E (Fig. R4a) this high force winds along the coast towards the site (Fig. R5b). There is a similar pattern of significant zonal easterly winds that approach the site from the Ross Sea side (Fig. R5b). These easterlies appear to be associated with a weak high over the Ross Ice Shelf and a low-pressure band to the north of the shelf. The easterlies appear to bring vapor to the site from the eastern Ross Sea polynya and sea-ice margin (Fig. R6a). With the original dating, the significant correlations you pointed disappear, especially with z500 at (55 °S; 90 °E), and the easterlies in the neighborhood of the Ross sector.

The westerlies associated with high off the polynya and sea-ice margin (Fig. R6a). The westerlies associated with high off the East Antarctic coast (65◦S, 90◦E) appears to provide a similar conduit of vapor from polynya region (Fig. R6a). The

d-excess signal originating from these regions can be anomalously positive as evaporation occur at a high-latitude low RH setting and the transport path towards the site can be associated with kinetic distillation processes as the air parcels are advected over ice, which further increase the positive d-excess anomaly. The cold atmospheric temperatures in combination with the long distance travelled by the air parcel over ice (without re-enrichment from an ocean source), can allow for a greater expression of the Rayleigh distillation and temperature dependence of kinetic effects, which is associated with higher d-excess values (Jouzel and Merlivat 1984; Risi et al. 2013). This is also consistent with your results, namely, the anti- correlation between d-excess and SMB; as the air is advected over ice shelf and sea- ice it will have experience "rainout" before reaching the site and thus the high d-excess values will be associated with relatively dry conditions. The setting with a positive SIC trend in the 110◦E–160◦W sector (Fig. R7) can also be important for the d-excess trend as it provide a setting with anomalously northerly sea-ice margin where air can be advected along the coast without contact with an ocean source. As aforementioned, we observe a significant correlation between the d-excessTA and the sea ice extent with our original dating, not only for the Ross Sea, but also the Bellingshausen, Scotia (in the North of Peninsula) and Lazarev seas (face to Dronning Maud Land). Nonetheless, we tested your hypothesis by fitting a linear relationship between the annual d-excessTA and the annual sea ice area where the concentration is lower than 15 % over the region (60 – 70 °S; 150 – 210 °E), as a surrogate for Ross Polynya. But we found no significant correlation. This was reported in the section 4.3 ("Water stable isotope, a fingerprint of changes in air mass origins"), after discussing the link between the d-excessTA and the sea ice extent, p29 l8: "As we suggested a particular d-excess signature in the TA firn core, associated with air masses coming from the western sector, we tested the possibility for the d-excessTA to imprint changes in the Ross polynya. We thus estimated it, by counting the annual sea ice concentration over the polygon (60 – 70 °S; 150 – 210 °E), lower than 15 %. But we find no significant correlation between this estimated Ross polynya and the d-excessTA over the period 1998-2014."

Note that both the eastern Ross Sea and the sea-ice region off the coast of East Antarctica (55◦S, 90◦E) have seen a significant negative sea-ice trend over the examined 1997–2014 period (Fig. R7). This is particularly clear for the eastern Ross Sea. The negative eastern Ross Sea sea-ice trend, therefore, provides a compelling explanation to the positive d-excess trend. Interestingly, we observe positive correlations between d-excessTA and the sea ice concentration for areas associated with positive trends in sea ice concentration over the period 1998-2014, while we observe negative correlations for areas with negative trends in in sea ice concentration over the period 1998-2014. This, indeed, provides an additional proof that the deuterium excess has a high potential for reconstituting the origin of air masses, as it seems very sensitive to sea ice conditions. However these differences in the sign of the correlation between d-excessTA and the sea ice concentration highlight by another mean, the need for more comprehensive studies about the processes driving this signal. We reported these results just after describing the correlation map between the d-excessTA and the the sea ice concentration, p29 l3: "We also noted a coincidence between the sign of the correlation of the relationship between the d-excessTA and the the sea ice concentration, and the sea ice extent trend over the period 1998-2014 (S17 in the Supplementary Material), especially positive correlations (negative) associated to positive sea ice concentration trends. These findings call for mechanistic studies to understand the different processes behind d-excess associated to each air mass origins."

I think it could be interesting if you could show whether these transport pathways exist. You can do this by showing the paths associated with the back-trajectories, perhaps as clusters. Hysplit provide a cluster analysis. However, daily back-trajectories over the period 1998-2014 are too many. We thus make a k-means clustering analysis over the last point of whole back-trajectories, and obtained two clusters, one above the Indian Ocean, and another one in coastal West Antarctic Ice Sheet, as reported in the result section 3.5 p23 l14 (""): "A k-mean clustering over the last points of the whole back-trajectories indicate two main origins, in the Indian Ocean (62.4 °S, 131.7 °E) and in the coastal West Antarctic Ice Sheet (73.4 °S, 227.5 °E)."

Note that air originating from these sea- ice regions doesn't necessarily have to have seen a significant increase, that is the transport path can always have been there it could be that the polynyas just recently became active. However, your findings that back-trajectories from the WAIS+Pacific region have increased is consistent with a positive d-excess trend, assuming that the air pass over Ross Sea sea ice or shelf prior to reaching the site. Our new results from this response show that the increase in d-excess would rather be linked to the increase of back-trajectories from the western sector than the Ross sea polynya. But a proper explanation remains to be investigated by other studies (as significant correlations between the d-excessTA and the sea ice concentrations are also to be noted).

The examined interval, 1997–2014, is short and almost exclusively coincide with the Interdecadal Pacific Oscillation's recent negative phase. Thus, it should be cautioned that your results and these correlation plots might look different if the examination period had been longer. This feature was added in the conclusion, p35 l13: "Ways forward include a better documentation of the spatio-temporal variability in SMB and water stable isotopes using a matrix of coastal firn core records spanning longer periods over the last decades (17 points being small to assess linear relationships, and record climate shifts, e.g. the IPO shift occurring in 1998 (Turner et al., 2016))"

I suggest that you show correlation maps similar to those that I introduced. Local conditions like those presented in Figure 6 in the paper does not necessarily show any influence on the isotopic record. For example correlation plot with reanalysis z500 fields can provide you with information about the pressure systems that are associated with air advection to the site and the isotopic signature. A pressure time series from a nearby station will not provide you with this information. Thus I suggest that you replace Figure 6 with correlation maps. We left Figure 6 that we think necessary to illustrate the inter-annual variability of the observations. We took into account your advice, selecting correlation maps (Figure 12), describing them and including them in our interpretation (see previous response).

[Figure]

Note that it can also be fruitful to correlate d-excess with SST and RH reanalysis fields, to check if any region stands out. I can share my Matlab analysis code for the correlation maps if that would be helpful? Contact the editor if you would like to get a copy. Many thanks for suggesting to share your code. However, we decided to keep this idea for a study focusing on the processes behind the d-excess signature.

The dating process of this core seems to be challenging. However, by not providing any reanalysis correlation maps or investigate the records relationship with standard indices (such as for ENSO and SAM), it feels like you have not exhausted the standard methods used to interpret the isotopic signature of an ice core. We tested potential linear relationships between the isotopic TA records and the modes of variability. We first introduced the used of specific indices in the methods, section 2.2 ("Datasets"), p11 l15: "Modes of variability We tested either the main modes of variability were imprinted in our recorded, especially: - the Southern Annual Mode (SAM) using the index defined by Marshall (2003), and archived on the National Center for Atmospheric Research website (Marshall, Gareth & National Center for Atmospheric Research Staff (Eds). Last modified 19 Mar 2018. "The Climate Data Guide: Marshall Southern Annular Mode (SAM) Index (Station-based)." Retrieved from https://climatedataguide.ucar.edu/climate-data/marshall-southern-annular-mode-sam-index-station-based.) - the El Niño Southern Oscillation (ENSO) using el Niño 3.4 index defined by the Climate Prediction Center of NOAA's National Centers for Environmental Prediction, and archived on their website (Trenberth, Kevin & National Center for Atmospheric Research Staff (Eds). Last modified 06 Sep 2018. "The Climate Data Guide: Nino SST Indices (Nino 1+2, 3, 3.4, 4; ONI and TNI)." Retrieved from https://climatedataguide.ucar.edu/climate-data/nino-sst-indices-nino-12-3-34-4-oni-and-tni.) - the Interdecadal Pacific Oscillation (IPO), using the IPO Tripole Index (TPI) defined by Henley et al. (2015) based on filtered HadISST and ERSSTv3b sea surface temperature data and archived on the internet (Accessed on 09 20 2018 at "https://www.esrl.noaa.gov/psd/data/timeseries/IPOTPI".). - the Amundsen Sea Low pressure center (ASL) archived one the National Center for Atmospheric Research

website (Hosking, Scott & National Center for Atmospheric Research Staff (Eds). Last modified 19 Mar 2018. "The Climate Data Guide: Amundsen Sea Low indices." Retrieved from https://climatedataguide.ucar.edu/climate-data/amundsen-sea-low-indices.)" We then reported the results of the linear relationships between our records and theses modes of variability in section 3.5 (retitled "Influence of synoptic weather on TA records: insights from ECHAM5-wiso simulation, ERA-interim reanalyses, back-trajectories and modes of variability"), p23 l19: "Note that no significant relationship is obtained between the TA records and any mode of variability."

Combining the correlation maps with the back-trajectory analysis can be a powerful approach to test the d-excess hypothesis and thus aid in addressing the goals of the study (P7 L7). I don't think it is justified to call the research challenging (referring to the article title) before you have tried additional analysis. The record you present here is interesting so it also deserves a more exciting title. We have followed your advices and nevertheless, still remain limited in our interpretation. We thus prefer keeping this title as the message we would like to emphasize is the current limitation of this isotopic interpretation, at least for Adélie Land.

The introduced update of the age-depth relationship, should not be viewed as a permanent update. Instead, it was meant to raise a concern. It would be good if you can find some age constrains for the core. Many thanks to have raised concerns about the age-depth profile. Unfortunately, we do not have more constrains.

The data for the closest stake needs to be included in Fig. 3 if it is central for the age-scale. SMB for the closest stake area is central for the age-scale, but not for the first step consisting in an annual layer counting, and illustrated by Fig.3. Its use comes to a second step consisting in comparing the reconstructed accumulation from the TA with the SMB from this stake area. This is illustrated by Fig. 4.

I like the age-scale section otherwise it has enough detail to ensure reproducibility. Are there any other chemical records from the core, any other shallow cores or pits (that

can provide a historic snow surface), or can you use the Acoustic Depth Gauge (ADG) record of snow surface height from the D-10 station? We are very confident in the SMB data from the closest stake area which is highly correlated with the 156 km stake line data. It covers only the period 2004-2014. Unfortunately, no other shallow cores or pits have been drilled and analyzed in the surroundings. And we have no knowledge of Acoustic Depth Gauge record of surface height from the D10 station.

In your 2017 study, you used NH4 and Na, but they might not add any additional information? You couldn't match the isotopes with the S1C1 record, but is it possible with any other records? This might be helpful if the S1C1 age-scale is associated with less uncertainty. NH4 and Na are associated with more uncertainties than nssSO4 and MSA: NH4 depends not only on the population of penguins, but also on conditions enabling the degradation of urea (Legrand et al., 1998), while Na shows peaks during winter, a feature very specific to Adélie Land and not fully explained. The S1C1 firn core is associated with an uncertainty of 8 years, its accumulation is not correlated with the ECHAM simulation, and its variability is different from the Glacioclim stake data over the short period 2004-2007. Unfortunately, there are no other data.

In the future, can there be something to gain in terms of resolution to measure these cores on a continuous-flow analysis (CFA) setup or increase the sample resolution of the discrete samples? For water stable isotopes, a CFA setup is currently installed, so it is planned for the next months.

We were fortunate on the project that I worked on researchers had visited there in the 1970s and retrieved cores. From their pioneering research, we got additional age-constrains from their snow surface at the time, plus beta-counts and we were able to match our isotope record with theirs (Emanuelsson 2016; Winstrup et al. 2017). We looked at your work we were not aware of. Our interpretation regarding air masses coming the western sector is in line with what you suggested, leading us to complete section 4.3 ("Water stable isotope, a fingerprint of changes in air mass origins") p29 l20: "These dry air masses might origin from the Amundsen Bellingshausen sea (Winstrup

et al., 2017; Emanuelsson et al., 2018), but cannot be directly linked to the Amundsen sea cyclonic, as we obtain no significant relationship with the ASL center pressure indices."

Age-scale thresholds The introduced threshold for your annual counts does not capture the older/deeper part of the record. Note how the picks at 12.2 m, 11.2 m, and 10.5 m (currently assign for the year 1999, 2000, and 2001) does not exceed the 100 ppb nssSO4 threshold. There, of course, has to be layer counts for this section, but they are not strictly speaking captured by the introduced rules. You are true. Thanks for noticing this mistake. So our dating is reproducible, we changed the rules we had fixed, p13 l3" For depths lower than 10 m w.e., summer (December-January) peaks were identified (i) from nssSO4 values higher than 100 ppb, synchronous with MSA peaks (with no threshold), and (ii) for nssSO4 values higher than 200 ppb (with or without a simultaneous MSA peak). Double nssSO4 peaks were counted as one summer (e.g. 2012, 2003, and 2001). For depths higher than 10 m w.e., summer peaks were identified for nssSO4 values higher than 27 ppb."

Perhaps this can be linked to the low proportion of air parcels that come from the Indian sector at the beginning of the record? You could might thus get less of an ocean signature. There is no trend in the proportion of air parcels coming from this sector that would allow to make such a suggestion.

Please cite earlier studies that have successfully used nssSO4 and/or MSA for dating (e.g., Steig et al. 2005; Abram et al. 2013). This method has been used for decades (e.g. Wagenhach et al., 1994), and the seasonal cycles of these species depend on local observations of the seasonal cycles of associated aerosols. We thus prefer referring to our previous publication (as already done) which gives the appropriated citations (i.e. related to atmospheric observations at Dumont d'Urville).

Back-trajectories Show your results on maps too. Consider cluster analysis for the trajectories or it might suffice to show just the 5-day endpoints. The path can be important

though, consider this case. The air is classified as WAIS+Pacific as the endpoint fall in this sector, but the air advected from this sector can be re-enriched from an ocean source or polynya on its way to the site. Please refer to prior answers.

I also suggest that you split up the Indian sector, it covers a too vast region. In my mind, the region from 0ᵉ–90ᵉE is likely to be associated with a different signature compared to direct onshore moisture transport. Sticklers might also object to that the Indian and WAIS+Pacific sectors intrude into the Atlantic Ocean sector. Our arm was to discriminate different types of back-trajectories, especially continental vs maritime, and East vs West trajectories, which explains large defined zones: the plateau and the newly named eastern sector for the EAIS, the western sector for WAIS, and the Ross sea sector in between those two zones, with a particular climate still under debate regarding its association to EAIS or WAIS for recent past climate variability (Stenni et al., 2017). Narrower sectors might be defined in a new study focusing on processes.

Also, why do you leave a region south of New Zealand undefined? It was a mistake as we should have taken into account this region. We thus included in the eastern sector (previous "Indian Ocean"), changing the boundary of the regions in in the description (section 2.2 "dataset"), p10 l21: "(i) the eastern sector: $(0 - 66 °S, 0 - 180 °E)$, (ii) the Plateau: $(66 - 90 °S, 0 - 180 °E)$, (iii) the Ross sea sector: $(0 - 75 °S, 180 - 240 °E)$, and finally (iv) the western sector: $(0 - 75 °S, 180 - 240 °E)$, and $(50 - 90 °S, 240 - 360 °E)$." We also recalculated back-trajectories with end points in this region, as relationships with this amount. Quantitative results were modified, but note that it does not change the main results (ie trends and significant relationships).

Is it a concern that the start elevation of 1,000 m a.s.l. does not correspond to the elevation at the site (602 m a.s.l.). One of the findings in Goursaud et al. (2018a) was that the site elevation was important for the d18O-d-excess phase-lag. This is an important point we raised in a previous study, and that we cannot have answered until now, as current analysed ice cores were drilled at sites lower than 1 000 m asl. However, the new ASUMA ice core measured in water stable isotopes and under study

was drilled at a site above this threshold. We thus hope that its study will provide a step forward in the comprehension of the $\delta$18O- d-excess phase lag.

"Finally, we associated each daily back-trajectory to daily precipitation $\delta$18O and d-excess values simulated by ECHAM5-wiso in the precipitation, and classified the time series for each variables by back-trajectories sectors." (P22 L3). Did you report any of the results from the d-excess back-trajectory analysis? Yes, we reported these results p24 l14: "The d-excess mean seasonal cycles substantially differ by their amplitude: for air masses coming from the western sector, it is 11.8 ‰ with outstanding values in March and October (minima) and in May (higher than the mean plus two standard deviations), whereas it varies between 3.2 ‰ and 3.6 ‰ for the other sectors."

Is there a way to tag the back- trajectories with TA d-excess data? It would be neat, if you can tag the back-trajectories with monthly TA d-excess values and then show that high d-excess months tend to originate from (or pass over) the Ross Sea and the off the coast of East Antarctica ( 65◦S, 90◦E) negative sea ice trend zones. To pinpoint these area, you might need a finer sector resolution and to also look at the path not only the 5-day back-trajectory endpoints. We had already tagged monthly averaged back-trajectories with either $\delta$18OTA or d-excessTA but unfortunately, nothing stand out. For Ross Sea, please refer to the test we made with our estimation of Ross polynya which brought no significant correlation with the TA isotopic records.

Post-depositional effects I think three years is a too short period. However, this number corresponds to the only peaks we have a (weak) doubt on.

Your argument, as you pointed out, is assuming that there is no natural interannual or inter-decadal variability. What ratio do you get if you just split the records in two parts (8 yrs in each)? Dividing the mean amplitude of the 8 first annual amplitudes by the mean of 8 next ones, we obtain a ratio of 1.06.

If you want look at intra-annual (seasonal or monthly) isotopic resolution I still think you would have to consider diffusion. Please refer to our calculations of diffusion lengths

(in months) using the method of Küttel et al. (2012), we reported as an answer in the next lines.

However, I think your analysis and the correlation maps (Figs. R1–6) using annual values shows that you can also obtain interesting information from the annual records. We thank you once more for having pointed these tools to extent our analysis.

Note that one approach would be to forward-diffuse the isotope records (Vinther et al. 2010; Küttel et al. 2012). You can use the methodology in Küttel et al. (2012) and then compare the results from using the forward-diffused record and the untouched record to evaluate objectively if there is a difference. We applied this methodology and reported the results in the section 3.1 ("Firn core chronology"): P13 l19 "For the TA, we also estimated the diffusion length (Küttel et al., 2012), and found mean diffusion lengths of 1.4 ± 0.3 months for $\delta$18OTA (with a maximum of 1.9 months in 2007), and 1.6 ± 0.5 months for d-excessTA (with a maximum of 2.4 months in 2007)."

ECHAM5-wiso Could the S1C1core's age-scale be better constrained than the TA core? Beta-counts were used for the S1C1age-scale, which can help to reduce the uncertainty. The significant correlations for the S1C1 with the model data would indicate that this could be the case and the lack thereof for TA could indicate that TA's age-depth relationship is not as well constrained. This provides another indication that an annual-layer count might be missing for the TA age scale. These cores are located only 14 km apart, so they will be exposed to similar climatic conditions. It thus seems to be a hasty conclusion to attribute the lack of similarities for the accumulation records of the two core solely to S1C1. "the absence of similarity between the TA and the S1C1 accumulation reflects the uncertainty in the S1C1 dating resulting from the large spatial variability and from more frequent erosion processes occurring at the S1C1 site" (P24, L1). Although the S1C1 firn core was dated with more constrains, it is associated with more uncertainty (8 years). Indeeed, it was really more difficult to date it. Also, people on the field can have attest of stastrugi and rugosity for the S1C1 drilling site contrary to the TA192A, evidence of wind drifting and scouring. It is also displayed by the correlation coefficient of the closest stake from the S1C1 and TA192A data with the 156 km mean accumulation (added in the Supplementary Material of the manuscript, and in the Supplement attached to this response). Finally we are very confident with the match between the reconstructed accumulation and the stake data.

For the same reason, I would caution you against being too strong with your criticism against the model simulations here. The analysis that you perform requires that the age-depth is well constrained, that is, that the isotope data is aligned with the GCM data year by year. (The seasonal cycle comparison should not be as sensitive though.) We did not write that the model was wrong but listed the potential causes of model/data mismatch, p32 l2: "This could be related to (i) to post-deposition processes associated with wind scoring or snow metamorphism not resolved in ECHAM5-wiso, (ii) the key role of very local atmospheric circulation effects related to katabatic wind processes, not resolved in large-scale atmospheric reanalyses and simulations, (iii) or the difficulties of ECHAM5-wiso to resolve the processes associated with the ocean boundary vapour d-excess", that we develop within the paragraphs. Finally, the only solution to decipher wich of the model or the data is not a (realist) climate signal is to obtain more isotope data, as we can have concluded p32 l9: "and thus calls for more isotopic measurements (from ice cores, snow precipitation, and water vapour) in Adélie Land to reduce uncertainties."

SMB Are you using a sufficient number of points (stakes and cores) to get conclusive SMB results? There are 91 stake data on the 156 stake line from Dumont d'Urville towards Dome C (Agosta et al., 2012), with a field mean accumulation highly correlated with the three closest stake data from the TA192A drilling site (r): - the "18.3" stake point, 1 km from the TA192A drilling site, and r = 0.91; - the "19.2" stake point, 83 m from the TA192A drilling site, and r = 0.86; - the "20.3" stake point, 998 m from the TA192A drilling site, and r = 0.85; Correlation coefficients with the TA192A firn core given in fig. 4 argue for the robustness of our reconstruction (over the overlapping period 2004-2014). And that is one of the key messages of the section 4.1 ("SMB")

we develop, defending here that our firn core thus imprinted a regional SMB signal. A second key message (l5 to l10) is that, over the period covered by the TA192A (1998-2014), we observe no trend. This is an important point, as shown by Thomas et al. (2017). Of course we are less confident out of the interval time from 2004 to 2014, but there is an absence of trend not only in the TA firn core but also in ECHAM.

Perhaps it would be wiser to leave the SMB part for a later publication? As drawn in our introduction, our arm through our current studies, is to infer SMB information from water stable isotopes. Consequently, this paragraph is important to remind it, and we made it clearer by adding the following paragraph p27 l6: "The anticorrelation between the d-excess$_{TA}$ and SMB$_{TA}$ shows the possibility to use water isotope firn core records from Adélie Land to complete the document of the SMB spatio-temporal variability. Dry air masses from the western sector may be associated with particular high d-excess values. The remaining uncertainty in the dating and the extraction of a pure signal limited our investigation.".

For that study, you could include all the available stakes and cores. Unfortunately, there are no more stakes and cores.

I suggest that you home in on one or two novel ideas. I also suggest that you work on focusing and shortening the paper, clarify sections, and work on presenting some findings in figures instead of the text. (I can also become better at this). We have decided not to shorten the manuscript, because we did not find one prominent message about the interpretation of our water stable isotope records. Thus, at this stage, we think that all the pieces of information we can have extracted from this study might help to deepen the question.

I like your short summaries at the end of each section and they are written in a clear concise way. Strive towards getting this level of clarity throughout. Suggestions: focus on the $\delta$18O and d-excess TA data, interpretation of the signal (explain the d-excess trend), the isotopes relationship with the ECHAM5-wiso model, and back-trajectories.

These findings seem novel, robust and interesting. Remove the snow surface sampling archive part, the SMB and the $\delta$18O/d-excess signature analysis? You can still publish these findings elsewhere. Many thanks for these advices, but please take into consideration our point.

Specific comments: I suggest that you add a subscript to $\delta$18O and d-excess when you refer to ECHAM5- wiso data. That way the reader doesn't have to look back in the text to check if it is the simulated or TA data that you refer to. Done

Article title You can perhaps change the title. You have to show that this is true first, but here is my suggestion. "Recent positive trend in d-excess in an Adélie Land coastal ice core, Antarctica, is linked to the activation and increased distant advection from the eastern Ross Sea polynya". The new tests we made do not argue in favor of this statement.

Abstract P1 L2. Provide the name of the core here too: Terre Adélie 192A (TA192A). Done.

P1 L5. It is not necessary to include "hereafter" when you introduce a new acronym. Writing "... reconstructed surface mass balance (SMB)" should suffice. Change this throughout. I removed it throughout the manuscript.

In the abstract you might even consider avoiding the acronyms altogether to be brief. We removed all acronyms in the abstract.

P1 L15. Remove the space so it says:"isotope-temperature relationship" Done.

P1 L20. Use another type of dash for intervals, that is, the en-dash: "135 ◦–145 ◦E" and "1998–2014". Change throughout. Done.

Introduction P3 L22. Change this sentence so it reads? ...are thus essential to provide continuous local to regional sub-annual resolution climate information spanning the last decades. We modified the sentence to: "...are thus essential to provide continuous climate information spanning the last decades at sub-annual resolution, at local but

also regional scales."

You would ultimately be more interested in information about past climate conditions, not just a couple of decades back. Recent data is important for the interpretation of the isotopic signal as there is an overlap between the observational records and proxy records. From a SMB point of view, it also allows to expand the documentation of the spatio-temporal variability and the comprehension of the underlying mechanisms.

P3 L24. The SMB acronym has already been introduced on line 11. Done.

P3 L27. The ITASE cores (Mayewski 2005; Steig et al. 2005) are not coastal (at least not all of them), unless you define everything that's not on the plateau as coastal. They are similar to coastal cores, however, in that they are exposed to synoptic conditions. We removed the citation, and did not replace it by another one, as those given in the following sentence illustrates this purpose: "in Dronning Maud Land (e.g. Isaksson and Karlén, 1994; Graf et al., 2002; Altnau et al., 2015) and the Weddell Sea Sector (Mulvaney et al., 2002). Fewer annually resolved water stable isotope records have been obtained from ice cores in other regions, such as the Ross Sea sector (Bertler et al., 2011), Law Dome (Morgan et al., 1997; Delmotte et al., 2000; Masson-Delmotte et al., 2003), Adélie Land (Yao et al., 1990; Ciais et al., 1995; Goursaud et al., 2017), and Princess Elizabeth region (Ekaykin et al., 2017)".

P4 L5. Change to "initiated" instead of "triggered"? Done.

P5 L3. I would remove "until now". Otherwise it sounds like you have resolved this issue here once and for all in this study. Done.

P5 L11. Under-documented, is not a common word (at least not according to google Ngram), replace? Can you say, "remains poorly documented and understood", instead? Done.

P7 L19. Stick to one convention for units, so write "3.9 m s-1" here instead. Done.

P6 L17. Remove "search" here and say something like, highlights the importance of

Interactive
comment
retrieval of more Done.

P6 L22. 139.56âŮȩE not S. Done

P7 L6. I would say resolution instead of scale here. Done.

P7 L7. Remove ", and dated." And start a new sentence and say "these records were used to establish the age scale for the firn core. Done.

P7 L10. Section references can be shortened, e.g., Sect. 2. Material and method Done.

P7 L18. No need to write "(see Fig. x), (Fig. x) should be fine. Change throughout. Done.

P8 L4. You write ", respectively" here but it's not clear what it refers back to. Rewrite so this sentence becomes clear. We completed the sentence as follows: "Samples devoted to ionic concentration measurements were stored in the cold room until concentrations of sodium (Na+), sulfate (SO42-), and methane sulfonate (MSA) were analyzed by ion chromatography equipped with a CS12 and an AS11 separator column, for cations and anions respectively."

P8 L8. Write Picarro without all the caps. P8 L16. Use the DDU acronym. Done. P8 L18. When in 1959? March. We precised it in the text.

P8 L24. Change to "range" not "ridge". And change the dash symbol for the range. Done.

P9 L3. Change to ". . . km from the TA drill site" instead. Done.

P9 L10. Move Table 1 to the supporting material? We did not remove Table 1 to the supplementary material so references of all data we used are cited in the manuscript.

P9 L15. Change to "back-trajectories" to be consistent with the rest of the text Done.

P9 L19. Change to "DDU intra-annual precipitation variability." instead. Done.

P9 L20. If you use the 0.75âŮ̧ex0.75âŮ̧e grid data, wouldn't the grid point at 139.50âŮ̧eE be the closest point to the drill site, not 140.25âŮ̧eE? Note that you can choose even higher resolution when downloading the data. We made this mistake as we were thinking about observations at Dumont d'Urville (66.7 °S, 140.0 °E). We thus extracted the precipitation for the grid point 0.75 ° western (i.e. 139.5 °E). We did not substantial changes neither in the distribution of the precipitation along the year (see Table 7 in the Supplement attached to this response, and replaced in table S7 of the Supplementary Material of the manuscript), nor in the mean precipitation of 46.0 $\pm$ 26.9 cm w.e. y-1 (replaced p15 l14).

P9 L28. 1500 m a. s. l. not m. a. g. l.? Yes indeed. We changed it.

P10 L24. Change to "... same simulation as Goursaud et al. ...", instead of ". . .same simulation than Goursaud et al. . . .". Done.

P10 L26. Can you add one more year of ECHAM5-wiso output data to the analysis to match the analysis period of the core? Done. It did not change any results of our study.

P10 L27. End the sentence with a period. Did you rather mean a point?

Results

P11 L7. Change to "multi-year study"? Done.

P13 L5. "...obtain a ratio of 0.5 (see Section 3.3)". Did you want to reference the supplementary table again, Table S3? We meant the next supplementary material, so we replaced Section 3.3 by the corresponding supplementary table 6.

In the supplementary table, it also looks like the ratio for d-excess is 1.1, not 0.5. I think you are a bit too thorough when you write (in the Supplementary Material) for each entry. Write it one time and then after that Fig. S1 and Table S1 etc. should suffice. We made a mistake and corrected this ratio to 1.1 in the text.

P15 L5. You write nssSO4 in many different ways throughout the paper (and define the

acronym at several places). I assume they are the same? Stick to one. We checked that the same notation nssSO4 was used through all the manuscript and changed it when it was not the case.

P15 L24. You mean p<0.05, not >0.05. We changed it to p<0.05.

P15 L25. The significance level should be <0.001 than, if r=1? Be consistent with the significant levels throughout, that is, include the <0.001 level too. In the Material and methods section, we precised the significance level p11 l13: "Note also that linear relationships are considered significant when the p-value < 0.05."

P15 L27. Change to "... in Fig.6. In Figures 6 and 7,..". Done.

P16 L1. Change to "Figs. S5 and S6" instead. Done.

P16 L5. "The sea ice trend is the largest in summer, they disappear if we discard the value observed in 2013." This sentence seems ambiguous, as it can be interpreted as it is only the trend in summer that is lost if 2013 is omitted. Split the sentences into two parts. We changed the sentences to: "Significant increasing trends are detected in the annual values of TA d-excess (0.11 ‰ y-1, r=0.61 and p<0.05) as well as of local and Indian Ocean sea-ice concentrations (0.10 % y-1, r=0.53 and r=0.52 respectively, p<0.05) with the largest sea-ice trend in summer. The sea-ice disappear if we discard the value observed in 2013."

P16 L18. Delete the "at all" part of the sentence. Done.

P16 L17 It is not clear what ", respectively" refers back to here. We detailed "r=0.65 and r=0.55 for $\delta18O$ and d-excess respectively".

P16 L26. It is just one value. You can present the p-value here instead of providing the range. We gave the exact p-value of 0.08.

P16 L27. Is it anti-correlated or positively correlated? This is hard to follow. We resentenced: "Although we are cautious with the short DDU precipitation time series

(with only 19 points, and p-values = 0.08, cell in italics in Table 5), it shows a positive relationship, similar to the one identified in the TA record. We conclude that the positive correlation observed in the TA records is specific to the coastal Adélie Land region, what is unusual in an Antarctic context."

P20 L4. "This relationship is valid for all seasons." Done.

P20 L7 "m.s-1" the period is not needed. We actually removed what was related to wind speed as we did not use it in the manuscript.

P20 L12. Correct figure reference, (Fig. S8). Done.

P20 L14. and L18. I understand why you have put the table and figure together. However, it is probably better to present them separately and refer to them as (Fig S#) and (Table S#) in the text. Done

P20 L14. Indeed there is a large spike on 7 May. Nevertheless, there are also other anomalously positive events during 2007 that contribute to the annual anomaly. It is the only so high value during the year (with a value of 43.3 per mille!). The second highest value is 27 per mille and all other values are below 20 per mille. As written previously, there are only 4 other so high d-excess values simulated over the period 1998-2014 (ie > 30 per mille, maximum d-excess mean + standard deviation simulated by ECHAM5-wiso over Antarctica at the monthly scale).

P20 L20. Please clarify this sentence. This is a case where I think a subscript can be handy and a table to refer to. As you have simulated and TA d-excess and DDU and wiso 2mT, it is hard to keep them apart. Especially in a long sentence like this. The sentence was rewritten as: "Neither t2mECH – $\delta$18OECH, nor SMBECH – d-excessECH, nor $\delta$18OECH – d-excessECH relationships are identified in ECHAM5-wiso seasonal or annual outputs."

P20 L24. Please provide a p-value. Rather than providing a p-value, we specified that the correlation is significant: "a systematic positive significant correlation..."

P21 L12. Change to "On average,". Done.

P21 first paragraph. You say that the calculation are based on the 1998–2014 period, on the 2nd line, that should be enough, so you can remove the "over the period 1998–2014" text from line 13, 13–14, and 15 as it is redundant. Done.

P22 L4. "and classified the time series for each variables by", Change to "variable" instead. Done.

Discussion

P23 L3. "slightly increasing but not significant trend". I'm afraid that you can't call it a trend if it is not significant. If it has some significance, p<0.1, you can perhaps call it weak. Given the additional 2014 year output from the ECHAM5-wiso and the lack of significant trend, we changed the sentence to: "For our study period (17 years for the TA record, and the ECHAM5-wiso simulation), we observe no significant trend."

P23 L18. I believe it is just called sastrugi when plural too. Done.

P24 L3. Are you referring to the wrong supplementary figure here? P24 L16. Is this p-value correct? Do you use too many zeros? We copied the data as given in the reference. However, we harmonized in our manuscript with our notation and thus replaced this value by 0.05. We specified this threshold for considering the relationships significant p11 l8: "Note also that linear relationships are considered significant when the p-value < 0.05."

P25 L3. The formatting of this header has been lost. Done.

P25. L27. Did you use 10 points (see Fig. 11) or 12 (see the referenced line) for the smoothing and running correlation? We used a 10-points running correlation. Many thanks for pointing this absent-mindedness.

P26 L13. "As a result, the $\delta$18O-d-excess relationship may be a fingerprint of changes in air mass origins, and particularly of the occurrence of precipitation of air masses

coming from the WAIS+Pacific sector". From visually inspecting Figure 7a (and from what you mention in the text) it seems like air originating from WAIS + Pacific and the Ross Sea sectors has increased. If your cite statement above would hold up, how would I see this in Figure 11? The frequency of occurrence of significant relationships (bold line) seems quite even throughout. 2007 and 2011 stand out as anomalies, but so does 1999, 2001, 2003. Or is 2007 more remarkable because it is a negative anomaly? Or do you mean that a spike in the slope signals an abrupt change in air origin, e.g. a sudden change from Indian to WAIS+Pacific? Are there any remarkable conditions associated with the year 1999, 2001 and 2003 too? We observe that anomalies in the $\delta$18O-d-excess slope, i.e. values out of the range defined by mean $\pm$ 2 standard deviations ([-0.81;2.48] ‰ ‰1), are associated to years 2007 (-1.46 ‰ ‰1) and 2011 (6.9 ‰ ‰1), identified as particular from air mass back-trajectories. Checking, we realized that we had made a mistake for year 2002 which is actually within the range by mean $\pm$ 2 standard deviations. Slopes calculated for years 1999, 2001 and 2003 are also within this range. Although we note concomitant specificities in $\delta$18O-d-excess slopes and air masses from the western sector, we could not explain processes behind these features. The origin of the air masses might not be the only condition to explain these particular isotopic signatures, but also specific moisture conditions at this location or along the trajectory. We were limited to investigate processes, as for this, we would need data allowing the identification of events (typically daily data).

P26 L17. ". . . respectively (consistently with what obtained within each year, see. . .). ". . .-1, respectively (this is consistent with the annual means, see. . .). Does this sound better? Done.

P27 L7. Change to "not only with stake data closest to the drill site". Done.

P27 L11. "However, we cannot draw any conclusion. . ." Done.

P27 L12. "(Unfortunately, there are no striking features during the records common period, which makes it challenging to match the isotope records)". Done.

P27 L18. "These findings suggest". Done.

P28 L2. "(from ice cores, snow precipitation, and water vapour)' Done.

P28 L5. "We compare the chemical concentrations recorded in the TA firn core with S1C1core, for their common period (1998–2006). The mean concentrations are slightly lower for the. . ." Done.

P28 L15. "Air originating from near the sea ice margin may contain relatively high d-excess". In general, I think to say just "air" is fine, that is, the "mass" or "masses" part is often not needed. We prefer to keep using "air masses" which is more rigorous physically as it characterizes an air parcel by its temperature and water vapor content.

P28 L17. "Such a configuration. . ." this sentence is confusing as it doesn't appear to correspond to the setting described above. The presiding sentence describes a low sea ice setting and the latter describes a positive sea ice scenario, so these sentences don't seem to correspond with one another. "Such a configuration should be associated with high sea-salt concentrations due to increased sea-salt emissions when the sea-ice concertation is low in summer." In this configuration, we meant that sea-salts concentrations should be high compared to air originating from ocean. We thus reworded: "Such a configuration should be associated to low sea-salt concentrations due to the presence of the sea ice, as shown by atmospheric studies (Legrand et al., 2016)."

P28 L25. Chang to "probably caused by marine air advection". Done.

P28 L26. It might be better to avoid starting sentences with a conjunction ("And" here) in formal writing, as some people might object to it. We removed it.

P29 L13. Change the highlighted parts of the sentence "...insure that small scale SMB random variability caused by presence of sastrugis, dunes and barchans is negligible. . ." to "ensure that small-scale SMB random variability caused by presence of sastrugi, dunes and barchans is negligible". Done.

[Figure]

P28 L19. "In contrast, vapour formed over the ocean. . ." Done.

P29 L1. Change to "To summarize," or "In summary," instead. Done.

Conclusions and perspectives P29 L11. Change to "The high estimated SMB rate of 74.1 $\pm$ 14.1 cm w.e. y-1 limits the effects of diffusion and ensures that records with sub-annual resolution are preserved." Done.

Figures Change the figure texts so that not everything is in bold. Just keep the fig numbering (Fig. 1, etc) and the letter (a) indicators in bold. Change throughout this section. It is not necessary to write "in" when you specify units, e.g. (in ‰Ċhange throughout the document. Done.

Fig. 2. Remove the 45âŮȩS boundary in the figure, otherwise, it looks like these regions have a northerly limit at this latitude. Done.

Defining sectors: *It is confusing that you use both signs (-, +) and letters to indicate the hemispheres. Stick to just letter. Done.

*I also suggest that you write the intervals as ranges as you do otherwise. That way you can skip the "lat", "lon" text too, S/N and W/E provides you with this information. Example: Ross Sea sector: 0âŮȩ–75âŮȩS, 180âŮȩ–240âŮȩE. You write that you don't have any equatorward boundary. Assuming that you don't have any endpoint in the NH you can write this way. Done.

*Skip the dash in "Ross-Sea sector" as it is a name. I also don't think you need all of the quotation marks when you introduce the names, see if you can write this in a clear way without them. We removed the dash in "Ross Sea sector". However, we remained the quotation marks as it refers to the text in the figure.

*Note also, that you wouldn't usually call a sector that extends so far into the Atlantic Ocean, Indian or WAIS+Pacific. We changed the names of the regions: "Indian" to "the eastern sector", and "WAIS+Pacific to "the western sector.

*Apply this formatting guidance to where you define sectors elsewhere in the text too, e.g. P8 L25 and P10 L12. Done.

Fig. 3. The $\delta$18O record seems to be shown both as regular value and a smoothed record, but you don't tell the reader how the record has been smoothed. You reference Fig. 3 in the figure text, but this is Fig. 3. For clarity it would be better if you list what the figure show here and describe the methodology elsewhere. Smoothing of $\delta$18O should have been removed, as we had made tests for dating, but finally did not use it. We thus replicated the figure without.

Fig 6. Local sea-ice extent with a % that is in the 124–134% range, this looks odd to me? Fig. S6 should be the same just with the standard deviation too. However, note here how the range of sea ice concertation is different. You write sea ice concentration in the figure text and sea ice extent on the figure axis, please clarify. All that said, I rather see that you scrap this figure and replace it with correlation maps. As mentioned before, we changed the definition used for the sea ice: whereas we looked at the mean concentration (which is, by the way, over 233 in the NSIDC data; so we forgot to divide the outputs by 233 for the figure), we defined the sea ice extent face to Dumont d'Urville as the number of grid point with a sea ice concentration higher than 15 % between 135 and 145 °E. The figure was changed with recalculated values, consistently with this last definition.

Fig. 7. Mention the panels in order, that is, accumulation occurs last, but is mentioned first in the text. Done.

Fig. 9. A letter is missing text inside the figure, it should say "Plateau". Can you increase the length of the bars (a, b) so it will be easier to see differences between the regions? You could also consider splitting a, c and b, c into two separate figures and align a, c so that the years in the panels correspond to each other, that is so that 1998 and 2014 is aligned in a and c. And that the moths are aligned in a similar manner in b, d. Subplots a and b were removed to show only a sector diagram with mean

[Figure]

percentage of back-trajectories for each sector.

Fig. 11. Maybe start the sentence with "Ten" instead of 10. There is also one paren­thesis bracket too many. Sentence 2. Change to "Significant results are indicated by thick lines (p<0.05)". It is confusing that the secondary axes exceed 1.0. Also, you might want to change so it's not using a comma for the axis text, (e.g.1.0 instead of 1,0). Done.

Supplementary material The CNRS affiliation is missing for Vincent Favier, Suzanne Preunkert, and Michel Legrand. Show figures and tables as separate items. Done.

Start by list the tables Table S1, S2, S3, . . ., then continue with the figures in a new section (Fig. S1). Done.

Table S7 ". . .and deuterium excess ("dxs") from. . .", change to ". . .and d-excess from. . ." to be consistent with the rest of the document.

Place the text associated with figures below the figures. Done.

The year 2014 seems to be missing in Fig. S8. Add one more year of ECHAM5-wiso output data to the analysis to match the analysis period. Done.

Article references Please, format the references list so that there is wider line space between individual article entries (like in the Reference list below). It is hard for the reader to identify where one entry start or ends otherwise. Done.

The Cavalieri et al. 1996 reference does not appear in the Reference list. We added it.

Figures attached to this response

Figure 1: Annual layer counting of the TA192A firn ice core with the numbering of the years of uncertainty tested for the dating. Figure 2: Correlation coefficient between the sea-ice concentration extracted from the Nimbus-7 Scanning Multichannel Microwave Radiometer (SMMR) and Defense Meteorological Satellite Program Special Sensor Mi­crowave/Imagers - Special Sensor Microwave Image/Sounder (DMSP SSM/I-SSMIS)

passive microwave data (http://nsidc.org/data/nsidc-0051) and stable water isotopes records from the TA192A ice core (using the original age-depth relationship), $\delta$18O in (a), and d-excess in (b). Figure 3: First empirical orthogonal function of moisture uptake in the boundary layer and the upper troposphere simulated by the moisture source diagnostic Watersip at the monthly scale over the period 1998-2014 (Sodemann, H., and Stohl, A.: Asymmetries in the moisture origin of Antarctic precipitation, Geophysical research letters, 36, 1-5, 2009.)

Author's changes in the manuscript We took into consideration your suggestion as well

as E. Thomas'. We also add a data availability paragraph referring to the PANGAEA data archive. Please find below the resulting revised manuscript. 
[revised manuscript text omitted]
 $1.4 \pm 0.3$ months for $\delta$18OTA (with a maximum of 1.9 months in 2007), and $1.6 \pm 0.5$ months for d-excessTA (with a maximum of 2.4 months in 2007). These results suggest that (i) potential post-deposition effects in the TA can be neglected. Notwithstanding, a potential loss of seasonal amplitude in the other average time series compared to the most recent seasonal cycles cannot be discarded, and has to be considered in the comparison of seasonal amplitudes, from one core to the other, in the comparison with the seasonal amplitude of precipitation $\delta$18O time series, and with ECHAM5-wiso outputs. 3.2 Mean values Mean climate from instrumental data Before reporting the mean values from TA records, we describe the available meteorological data. A time-averaged statistical description of the available meteorological data measured at DDU, the station closest to the drilling site, is given in Table 2 for the whole available measurement period prior to 2015 (1957—2014), and over the period covered by our TA records, 1998-2014. For all the considered parameters (near-surface temperature, wind direction, wind speed, humidity, and surface pressure), the time-averaged values differ by less than 8% (the maximum deviation being for the wind direction) over the period 1998-2014 compared to the whole available period. Standard deviations calculated over these two time periods also differ by less than one respective standard deviation unit, except for wind direction which shows

[revised manuscript text omitted]

Remarkable years Using only annual SMB, water stable isotope and chemistry TA records, we finally searched for remarkable years, defined here as deviating from the 1998—2014 mean value by at least 2 standard deviations. We highlight three remarkable years (red-shaded for high values and blue-shaded for low values, Fig. 6 and 7):  Year 2007: very low SMBTA in in TA data.  Year 2009: remarkably high $\delta$18OTA values.  Year 2011: high MSA, d'Urville summer sea ice extent, and wind speed values. We had previously noted that years 2003 and 2004 are associated with very high Na+ values and add that year 1999 experienced low nssSO4 values. and year 2013 high local sea ice concentration. The remarkable large sea ice concentration was also observed at a larger scale by Reid et al. (2015).

To summarizeIn summary, we identify increasing trends in d-excessTA and sea -ice concentration, no significant correlation between TA $\delta$18OTA and DDU near-surface temperature, and an anti-correlation between d-excessTA and SMBTA. We also note 2 remarkable years in SMBTA ("dry" 2007) and $\delta$18OTA ("high" 2009). Finally, no sys-

Interactive
comment

[revised manuscript text omitted]

mean valuelevel in 2007 (see Fig. S4118 in the Supplementary Material). Only year 2007 is remarkable in both the data (low reconstructed SMB) and the model. We thus explored more deeply the model. The highest d-excess value was simulated the 7th of May (see Table S8129a in the Supplementary Material). When comparing from the 6th to the 8th of May in 2007, with daily averages over the period 1979-20143, the model simulates similar near-surface temperature, but particularly low precipitation, and wind components (zonal and meridional). Despite the small precipitation amount, the daily isotopic anomaly is sufficiently large to drive the annual anomaly (see Fig. S5912b in the Supplementary Material). D-excess values higher than 30 ‰ (threshold chosen as it corresponds to the maximum d-excess mean + standard deviation simulated by ECHAM5-wiso over Antarctica at the monthly scale, see Fig. S613 in the Supplementary Material) occur only 4 other timer over 1998—2014. We nevertheless remain cautious with theseis values which could be due to a numerical artefact. Neither $\delta$18Ot2mECH – $\delta$18OECHtemperature, nor SMBECHd-excess – d-excessECH - SMB, nor $\delta$18OECH – d-excessECH relationships are identified in ECHAM5-wiso seasonal or annual outputs for precipitation and d-excess. Likewise, no significant relationship could be identified between d-excessTA and SMB simulated by ERA using both annual and seasonal values. A systematic positive correlation is identified between d-excess and $\delta$18O, except in summer. It is the strongest in austral spring, with a correlation coefficient of 0.75 (with a slope of 0.61 ‰ ‰1).

Simulated seasonal cycles for $\delta$18O and d-excess We now explore the simulated seasonal variations in $\delta$18OECH and d-excessECH over the period of simulation (1998-20143, see Table 6). The peaks in the simulated $\delta$18OECH predominantly occur in spring and summer (25 % and 63 % respectively), while it only happens 12 % of the time in winter and never in autumn. The simulated d-excessECH peaks most often in autumn (69 %) and secondarily in winter (31 %), but never during the other seasons. As a result, the model simulates more regular isotopic seasonal cycles with $\delta$18O maxima during spring to summer seasons, and d-excess maxima during autumn to winter seasons, than identified in the TA record.

Relationships with the large-scale climatic variability The ERA-interim outputs allow us to investigate whether the large-scale climatic variability influence the isotopic composition of Adélie Land precipitation recorded in the TA firn core. We looked at the simulated linear relationships between the TA isotopic records ($\delta$18OTA and d-excessTA) with the ERA-interim outputs (2mT, u10, v10 and z500, Section 2.2). We here report only significant relationships with absolute correlation coefficients higher than 0.6. For $\delta$18OTA, we found a correlation with 2m-T over the Antarctic plateau (Fig.12a), as well as a correlation with v10 (Fig. 12b) along the westerly wind belt, at $\sim$ (55 °S, ; 100 °E) and $\sim$ (55 °S, ; 130 °E) in the Indian Ocean, and at $\sim$ (55 °S, ; 10-50 °E) in the Atlantic Ocean, and a very little area on coastal Dronning Maud Land at $\sim$ (60 °S ,; 30-40 °E). For d-excessTA, we found a correlation with 2m-T (Fig. 12c) toward the Lambert Glacier at $\sim$ (70-80 °S ,; 30-40 °E), and an anticorrelation in the south of the Peninsulat at $\sim$ (55-65 °S,; 250-300 °E). Finally, we noted a correlation between d-excessTA and u10 (Fig. 12d) at a very narrow area of Dronning Maud Land at $\sim$ (80 °S,; 10-20 °E), and an anticorrelation on the westerly wind belt in the Atlantic Ocean at $\sim$ (55 °S,; 40-50 °E). No correlation is found with z500, neither with $\delta$18OTA nor d-excessTA. Note that no significant relationship is obtained between the TA records and any mode of variability.

Origin of air masses Finally, we used the HYSPLIT back-trajectory model to count the proportion (in percentage) of air mass back-trajectories, based on daily calculations over the period 1998-2014, and averaged at the annual and seasonal scale, from four different regions (see Section 2.3): the plateau, the eastern Atlantic Ocean and th Indian Ocean (eastern sector), the Ross Sea sector (hereafter RSS), and the West Antarctic Ice Sheet with the Pacific Ocean and the western Atlantic Ocean (hereafter WAIS+Pacificwestern sector), as displayed on Fig. 92. IOn average, the highest annual proportion of air masses comes from the eastern sectorIndian Ocean (49.854.1 $\pm$ 6.48.3 % over the period 1998-2014) and the East Antarctic plateau (32.54.3 $\pm$ 4.93.8 % over the period 1998-2014), while a small proportion of air masses come from the WAIS and the Pacific Oceanwestern sectors (9.76 $\pm$ 3.76 % over the period 1998-

2014), and from the RSS (3.66.3 ± 2.41.3 % over the period 1998-2014). A k-mean clustering over the last points of the whole back-trajectories indicate two main origins, in the Indian Ocean (62.4 °S, 131.7 °E) and in the coastal West Antarctic Ice Sheet (73.4 °S, 227.5 °E). Inter-annual variations in back trajectories (Fig. 9ba and 9b) reveal a positive trend for the fraction of air masses coming from western sector WAIS+Pacific (slope of 0.416.5 ± 0.162.1 % y-1, r=0.5563 and p<0.05), and remarkable years: 1999, which was identified as a remarkable high $\delta$18O value and low nssSO4 in our TA records, is here associated with a minimum of back-trajectories from the Plateau, and year 2011 which was associated with particular high MSA in our TA record, shows a minimum particular low proportion of air massesback-trajectories coming from the Indian regionestern sector and maxima ofwhile particular high proportion of air masses back-trajectories coming from the Ross and the western sectorsWAIS+
[revised manuscript text omitted]
.; and rRare dry air masses may also comeing from the western sector, and that thiswith a signal is preserved in d-excess., These different cases we propose here are illustrated by Figure XX. Last pointa is leadled by the following hints: (i) the positive trends both for the TA d-excess and the percentage of air masses coming from the western sectorWAIS+Pacific region (ii) an anti-correlation between the seasonal cycles of the TA d-excess and percentages of air masses coming from the western sectorWAIS+Pacific region (iii) the high simulated d-excess amplitude simulated by ECHAM5-wiso for air masses coming from the western sectorWAIS+Pacific sector, reflecting outstanding values occurring in autumn and winter times (iv) The particular case of the 7th of May in 2007 with very high d-excess values simulated by ECHAM5-wiso, corresponding to an air mass trajectory from the western sectorWAIS+Pacific sector As pointed discussed earlier, the last item should be consideredis however to take with cautionus as the four other remarkable d-excess values (i.e. higher than 30 ‰ simulated by the ECHAM5-wiso are associated to air masses coming from other regions (S613 in the Supplementary Material), and also to the fact that such high values it could be due to potential a numerical artefacts. Linear relationships between d-excessTA and ERA-interim outputs come to 
[revised manuscript text omitted]

Agosta, C., Favier, V., Genthon, C., Gallée, H., Krinner, G., Lenaerts, J. T., and van den Broeke, M. R.: A 40-year accumulation dataset for Adelie Land, Antarctica and its application for model validation, Climate dynamics, 2012, 38; 1-2, p. 75-86.

Altnau, S., Schlosser, E., Isaksson, E., and Divine, D.: Climatic signals from 76 shallow firn cores in Dronning Maud Land, East Antarctica, The Cryosphere, 2015, 9; 3, p. 925-944.

Amory, C., Naaim-Bouvet, F., Gallée, H., and Vignon, E.: Brief communication: Two well-marked cases of aerodynamic adjustment of sastrugi, The Cryosphere, 2016, 10; 2, p. 743-750.

Amory, C., Gallée, H., Naaim-Bouvet, F., Favier, V., Vignon, E., Picard, G., Trouvilliez, A., Piard, L., Genthon, C., and Bellot, H.: Seasonal variations in drag coefficient over a sastrugi-covered snowfield in coastal East Antarctica, Boundary-Layer Meteorology, 2017, 164; 1, p. 107-133.

Bertler, N., Mayewski, P., and Carter, L.: Cold conditions in Antarctica during the Little Ice Age—Implications for abrupt climate change mechanisms, Earth and Planetary Science Letters, 2011, 308; 1, p. 41-51.

Bertler, N. A., Conway, H., Dahl-Jensen, D., Emanuelsson, D. B., Winstrup, M., Vallelonga, P. T., Lee, J. E., Brook, E. J., Severinghaus, J. P., and Fudge, T. J.: The Ross Sea Dipole–temperature, snow accumulation and sea ice variability in the Ross Sea region, Antarctica, over the past 2700 years, Climate of the Past, 2018, 14; 2, p. 193-214.

Bonne, J. L., Steen‐Larsen, H. C., Risi, C., Werner, M., Sodemann, H., Lacour, J. L., Fettweis, X., Cesana, G., Delmotte, M., and Cattani, O.: The summer 2012 Greenland heat wave: In situ and remote sensing observations of water vapor isotopic composition during an atmospheric river event, Journal of Geophysical Research: Atmospheres, 2015, 120; 7, p. 2970-2989.

Bouchard, A., Rabier, F., Guidard, V., and Karbou, F.: Enhancements of satellite data assimilation over Antarctica, Monthly Weather Review, 2010, 138; 6, p. 2149-2173.

Bréant, C., Leroy Dos Santos, C., Casado, M., Fourré, E., Goursaud, S., Masson-Delmotte, V., Favier, V., Agosta, C., Cattani, O., Prié, F., Golly, B., Orsi, A., and Martinerie, P.: Coastal water vapor isotopic composition driven by katabatic wind variability in summer at Dumont d'Urville, coastal East Antarctica, Earth and Planetary Science Letters, submitted.

Bromwich, D. H., and Fogt, R. L.: Strong trends in the skill of the ERA-40 and NCEP–NCAR reanalyses in the high and midlatitudes of the Southern Hemisphere, 1958–2001, Journal of Climate, 2004, 17; 23, p. 4603-4619.

[Figure]

Bromwich, D. H., Fogt, R. L., Hodges, K. I., and Walsh, J. E.: A tropospheric assessment of the ERA-40, NCEP, and JRA-25 global reanalyses in the polar regions, Journal of Geophysical Research: Atmospheres, 2007, 112; D10, p. 1-21.

Bromwich, D. H., Nicolas, J. P., and Monaghan, A. J.: An assessment of precipitation changes over Antarctica and the Southern Ocean since 1989 in contemporary global reanalyses, Journal of Climate, 2011, 24; 16, p. 4189-4209.

Caiazzo, L., Becagli, S., Frosini, D., Giardi, F., Severi, M., Traversi, R., and Udisti, R.: Spatial and temporal variability of snow chemical composition and accumulation rate at Talos Dome site (East Antarctica), Science of the Total Environment, 2016, 550, p. 418-430.

Casado, M., Landais, A., Masson-Delmotte, V., Genthon, C., Kerstel, E., Kassi, S., Arnaud, L., Picard, G., Prie, F., and Cattani, O.: Continuous measurements of isotopic composition of water vapour on the East Antarctic Plateau, Atmospheric Chemistry and Physics, 2016, 16; 13, p. 8521-8538.

Cavalieri, D., Parkinson, C., Gloersen, P., and Zwally, H.: updated yearly: Sea Ice Concentrations from Nimbus-7 SMMR and DMSP SSM/I-SSMIS Passive Microwave Data, Version 1 [1979-2014]. Boulder, Colorado USA. NASA National Snow and Ice Data Center Distributed Active Archive Center, 1996.

[revised manuscript text omitted]

Wagenhach, D., Graf, W., Minikin, A., Trefzer, U., Kipfstuhl, J., Oerter, H., and Blindow, N.: Reconnaissance of chemical and isotopic firn properties on top of Berkner Island, Antarctica, Annals of Glaciology, 1994, 20, p. 307-312.
Werner, M., Langebroek, P. M., Carlsen, T., Herold, M., and Lohmann, G.: Stable water isotopes in the ECHAM5 general circulation model: Toward high‐resolution isotope modeling on a global scale, Journal of Geophysical Research: Atmospheres, 2011, 116; D15, p.

Yao, T., Petit, J., Jouzel, J., Lorius, C., and Duval, P.: Climatic record from an ice margin area in East Antarctica, Annals of Glaciology, 1990, 14; 1, p. 323-327.

Please also note the supplement to this comment:
https://www.the-cryosphere-discuss.net/tc-2018-121/tc-2018-121-AC1-supplement.pdf
* * *
[Figure]

**Fig. 1.** Figure 1: Annual layer counting of the TA192A firn ice core with the numbering of the years of uncertainty tested for the dating

**Fig. 2.** Figure 2a: Correlation coefficient between the sea-ice concentration and TA $\delta 18O$

**Fig. 3.** Figure 2b: Correlation coefficient between the sea-ice concentration and TA dexcess

**Fig. 4.** Figure 3: First empirical orthogonal function of moisture uptake in the boundary layer and the upper troposphere simulated by Watersip

[Figure]

**Supplement:**

Table 1: Occurrence of months associated to a $\delta^{18}O$ annual maximum simulated by the ECHAM5-wiso model

| Month | January | March | August | November | December |
|---|---|---|---|---|---|
| Occurrence | 6 | 2 | 2 | 5 | 2 |

Table 2: Peak number (as shown in Figure 1), depth (in m snow equivalent), $nssSO_{4,summer}$ (in ppb), MSA (in ppb), and $\delta^{18}O$ (in ‰) for each peak associated to an uncertainty in our dating.

| Peak number | Depth (m s.e.) | $nssSO_{4,summer}$ (ppb) | MSA (ppb) | $\delta^{18}O$ (‰) |
|---|---|---|---|---|
| 1 | 0.54 | 76.2 | 4.0 | -16.7 |
| 2 | 6.78 | 109.2 | 0.2 | -15.3 |
| 3 | 12.5 | 10.4 | 2.1 | -17.9 |

Table 3: Parameters of the linear relationship of the annual reconstructed accumulations including uncertain peaks 2 ("test2"), 3 ("test3") and both 2 and 3 ("test23") as shown in Fig.1, and the original reconstructed accumulation given in the submitted version ("original), with the simulated accumulation from the ECHAM5-wiso over the same cover period than the resulting reconstructions: the slope, the correlation coefficient ("r"), and the p-value and the standard error ("stderr'). The line in bold highlights the reconstruction which gives the best correlation.

| | slope (cm w.e. $y^{-1}$ (cm w.e. $y^{-1}$) $^{-1}$) | r | pvalue | stderr |
|---|---|---|---|---|
| test2 | 0.03 | 0.06 | 0.82 | 0.20 |
| test3 | 0.20 | 0.36 | 0.14 | 0.42 |
| test23 | -0.01 | -0.02 | 0.93 | 0.14 |
| **original** | **0.28** | **0.45** | **0.07** | **0.22** |

Table 4: Parameters of the linear relationship of the $\delta^{18}O$ annual averages including uncertain peaks 2 ("test2"), 3 ("test3") and both 2 and 3 ("test23") as shown in Fig.1, and the original reconstructed accumulation given in the submitted version ("original), with the simulated accumulation from the ECHAM5-wiso over the same cover period than the resulting reconstructions: the slope, the correlation coefficient ("r"), the p-value and the standard error ("stderr').

| | Slope ($‰‰^{-1}$) | r | pvalue | stderr |
|---|---|---|---|---|
| original | 0.16 | 0.32 | 0.21 | 0.12 |
| test1 | 0.14 | 0.33 | 0.19 | 0.10 |
| test2 | 0.11 | 0.28 | 0.25 | 0.09 |
| test3 | 0.18 | 0.37 | 0.14 | 0.11 |
| test23 | 0.11 | 0.27 | 0.26 | 0.09 |

Table 5: Parameters of the linear relationship of the $\delta^{18}O$ annual averages including uncertain peaks 2 ("test2"), 3 ("test3") and both 2 and 3 ("test23") as shown in Fig.1, and the original reconstructed accumulation given in the submitted version ("original), with the near-surface temperature measured at Dumont d'Urville over the same cover period than the resulting reconstructions: the slope, the correlation coefficient ("r"), the p-value and the standard error ("stderr').

|          | slope (‰ °C$^{-1}$) | r    | pvalue | stderr |
|----------|---------------------|------|--------|--------|
| original | 0.59                | 0.30 | 0.24   | 0.48   |
| test1    | 1.05                | 0.49 | 0.05   | 0.48   |
| test2    | 0.87                | 0.36 | 0.16   | 0.58   |
| test3    | 0.57                | 0.29 | 0.26   | 0.49   |
| test23   | 0.87                | 0.36 | 0.16   | 0.58   |

Table 6: Parameters of the linear relationship of the $\delta^{18}O$ annual averages including uncertain peaks 2 ("test2"), 3 ("test3") and both 2 and 3 ("test23") as shown in Fig.1, and given in the submitted version ("original), with the simulated temperature by the ECHAM5-wiso model over the same cover period than the resulting reconstructions: the slope, the correlation coefficient ("r"), the p-value and the standard error ("stderr').

| | slope (‰ °C$^{-1}$) | r | pvalue | stderr |
|---|---|---|---|---|
| original | 0.10 | 0.26 | 0.31 | 0.10 |
| test1 | 0.08 | 0.23 | 0.38 | 0.09 |
| test2 | 0.08 | 0.27 | 0.29 | 0.08 |
| test3 | 0.10 | 0.26 | 0.32 | 0.09 |
| test23 | 0.08 | 0.27 | 0.29 | 0.08 |

Table 7: Percentage of annual precipitation for the summer, from December to February ("DJF"), the autumn, from March to May ("MAM"), the winter, from June to September ("JJAS") and the spring, from October to November ("ON"), within each year from 1998 to 2014 simulated by ERA-interim.

| Year | DJF | MAM | JJAS | ON |
|------|------|------|------|------|
| 1998 | 36.3 | 29.8 | 27.1 | 6.8 |
| 1999 | 26.5 | 22.4 | 38.3 | 12.8 |
| 2000 | 20.1 | 17.4 | 48.7 | 13.8 |
| 2001 | 23.6 | 36.4 | 31.2 | 8.7 |
| 2002 | 34.2 | 21.2 | 34.8 | 9.8 |
| 2003 | 30.8 | 15.1 | 47.0 | 7.1 |
| 2004 | 19.9 | 32.2 | 42.5 | 5.4 |
| 2005 | 20.1 | 21.3 | 28.5 | 30.2 |
| 2006 | 27.3 | 39.9 | 25.5 | 7.3 |
| 2007 | 25.0 | 37.9 | 29.2 | 8.0 |
| 2008 | 19.6 | 33.9 | 37.2 | 9.4 |
| 2009 | 22.7 | 26.8 | 47.2 | 3.3 |
| 2010 | 33.1 | 22.4 | 27.5 | 17.0 |
| 2011 | 38.7 | 18.6 | 25.4 | 17.3 |
| 2012 | 19.0 | 22.7 | 48.9 | 9.3 |
| 2013 | 37.7 | 23.6 | 33.4 | 5.4 |
| 2014 | 33.5 | 35.6 | 26.0 | 5.0 |

---

## Author Comment (AC2) · 17 Dec 2018

Reply to the editor's comments The paper provides a detailed and thorough examination of stable water isotopes and snow accumulation and chemistry from a shallow (21.3m) ice core in coastal Adelie Land. The record covers a relatively short time (1998-2014), but demonstrates the importance of robust proxy evaluation, particularly at coastal Antarctic sites. We thank Elizabeth Thomas for reviewing our paper, and assigning importance to our work.

I have a few minor comments and suggestions: P2 Abstract – There is quite a bit of detail about the method in the abstract, which I felt was not necessary. Removing lines 7-13 will keep the abstract concise and focus more on your findings. Done.

P2 Line 16 - End sentence after . . ..timescale." Done.

Introduction The recent PAGES 2k compilation of ice core snow accumulation data (CP, 2017) highlighted the importance of coastal records. We added this important feature, after reporting coastal investigations: "However, the recent 2k temperature and SMB reconstructions for Antarctic (Stenni et al., 2017; Thomas et al., 2017) highlighted the need for more coastal records." P4 l4

There was no mention of the Antarctic Peninsula in your summary of drilling efforts at coastal sites? Several shallow, low elevation coastal cores have been drilled in this region. Indeed, we had forgotten to mention AP, and thus added it when reporting investigated coastal regions, citing the lastly work of F.Fernandoy: "such as the Peninsula (Fernandoy et al., 2018)" p4 l2

Climatic interpretation of water stable isotopes records – I felt some reference to previous studies on the role that sea ice plays was missing. I suggest checking the following: Noone, D., and I. Simmonds (2004), Sea ice control of water isotope transport to Antarctica and implications for ice core interpretation, J. Geophys. Res.,109, D07105, doi:10.1029/2003JD004228. Bromwich, D. H. (1988), Snowfall at high southern latitudes, Rev. Geo-phys.,26, 149– 168. Bromwich, D. H., and C. J. Weaver (1983), Latitudinal displacement from main moisture source controld18O of snow in coastal Antarctica,Nature,301, 145–147. Indeed, Noone and Simmonds (2004) showed how changes in sea-ice, taken independently from other variables, can affect coastal $\delta$18O, and in a more complex way the d-excess. However, this study does not account for the interplay between sea-ice changes and other climate variables, like e.g. katabatic winds, which also partly drive the atmospheric isotopic composition in Adélie Land, as shown by the recent vapor monitoring processed by Bréant et al. (submitted). In our

submitted version, we used the sea-ice concentration averaged over specific longitudi-
nal ranges. Also, we have been advised to rather consider the area where the sea ice
concentration is higher than 15 %, what we took into account, as described in section
2.2, p9 l1: "D'Urville summer sea ice extent was estimated by extracting the number of
grid points covering the area (50 − 90 °S, 135 − 145 °E) where the sea ice concentra-
tion is higher than 15 %, from December to January (included) for each year from 1998
to 2014." We just considered the d'Urville summer sea ice extent, and rather used map
correlations to look at the larger scale. We reported our findings in the results sec-
tion (removing our prior results), first in section 3.3 ("Inter-annual variations"), p16 l25:
"Significant increasing trends are detected in the annual values of d-excessTA (0.11 ‰
y-1, r=0.61 and p<0.05) as well as of d'Urville summer sea ice extent (r=0.77, p<0.05)."
and p17 l15: "Our record depicts a significant anti-correlation between annual values
of SMBTA and d-excessTA (r=-0.59 and p<0.05), as well as a significant correlation
between d-excessTA and d'Urville summer sea ice extent (r=0.65 and p<0.05)." With
our new results, we also obtain particular high values in 2011 (instead of year 2013),
as specified p18 l11: "Year 2011: high MSA, d'Urville summer sea ice extent, and wind
speed values." The pattern of the mean seasonal cycle remains unchanged. We thus
have changed our interpretation in the discussion part in section 4.3 ("Water stable
isotope, a fingerprint of changes in air mass origins"), p28 l21: "Noone and Simmonds
(2004) have shown, thanks to climate modelling, that water stable isotopes were con-
ditioned by changes in sea ice extent (as a contraction in sea ice increases the local
latent heat and temperature due to open water), but confirmed that a thorough under-
standing of main mechanisms controlling the d-excess was still needed. Also, earlier
studies have suggested the use of the d-excess recorded in ice cores to reconstruct
past sea ice extent (e.g. Sinclair et al., 2014). Although we find a significant corre-
lation between the d-excessTA and the d'Urville summer sea ice extent (section 3.3),
a correlation map between the annual d-excessTA and the summer sea ice concen-
tration (S16 in the Supplementary Material) show significant correlations with further
sea ice areas (e.g. an anticorrelation in the Amundsen sea and correlations in the

Belligshausen, Scotia and Lazarev seas)."

P19 Line 23 – remove "does" Done.

P21 Line 12 "On average,.." Done. Discussion The PAGES 2k compilation, based on ice cores and modelled SMB from RACMO2.3p2, identified a negative trend in SMB in neighbouring Victoria Land. The negative trend since the 1960s is statistically significant (p<0.01) and outside the expected range for the previous 200 years. A negative trend was also identified in the Wilkes Land coast (of which your Adelie land site would fit) which was outside the range of expected variability but not statistically significant. We thank you for reminding the reference. In the manuscript, we thus cited Thomas et al. (2017) for the unprecedented negative trend of SMB observed in Victoria Land for the last 50 years, but not for Wilkes Land as not significant, p. 23 l.16: "Especially, Thomas et al. (2017) report an unprecedented negative trend observed in Victoria Land for the last 50 years (1961-2010)."

P23 Line 5-7 Consider rewording . . ."we observe slightly increasing but not significant trend in the TA core, era and ECHAM5-wiso records". We reworded the sentence: "For our study period (17 years for the TA record, and the ECHAM5-wiso simulation), we observe no significant trend."

P24 Reference to other d18O temperature records. Poor correlations with temperature observed in the Antarctic Peninsula and coastal west Antarctica. For example, the relationship between d18O and temperature from ERA-interim at a coastal site in Ellsworth Land (Ferrigno) was 0.44 (p 0.01). At this site the relationship between d18O and temperature was similar to that of the relationship between d18O and sea ice conditions (r= -0.37 between d18O and winter sea ice extent and r=-0.54 between d18O and sea ice concentration). We thank you for completing our brief state-of-the-art of coastal dO18-temperature relationship. We can have added results from Thomas et al. (2013) p.24 l.27: "In coastal West Antarctica, Thomas et al. (2013) also reported a significant but weak correlation between the the $\delta$18O recorded in an ice core drilled

on the Bryan coast and the near-surface temperature simulated by ERA-interim, over the period 1979-2009."

P25 Line 3 – Should this line be a section heading? P25 Line 20 – Indeed. Thanks for making us notice.

Consider rewording – "As discussed earlier, this last item should be considered with caution due to potential numerical artefacts." Done.

Chemistry P28 Line 16 – There have been several studies suggesting that sea ice is a source of sea salt aerosols, therefore the presence of summer sea ice might not be associated with reduced Na? It might be worth checking or acknowledging these papers. See Yang, X., Pyle, J. A., and Cox, R. A.: Sea salt aerosol production and bromine release: Role of snow on sea ice, Geophys. Res.Lett., 35, L16815, https://doi.org/10.1029/2008GL034536, 2008. Rhodes, R. H., Yang, X., Wolff, E. W., McConnell, J. R., and Frey, M. M.: Sea ice as a source of sea salt aerosol to Greenland ice cores: a model-based study, Atmos. Chem. Phys., 17, 9417-9433, https://doi.org/10.5194/acp-17-9417-2017, 2017. (admittedly the later is for Greenland). In general, the year-round atmospheric sodium record at DDU exhibits a summer maximum. As already discussed by Wagenbach et al. (1998) such a seasonal pattern is reversed compared to those observed at other coastal Antarctic sites where the largest concentrations occur in winter. This finding pointed out the sea ice as an efficient source of sea-salt aerosol in winter. The outstanding summer maximum of sea-salt concentrations only detected at DDU is related to the particularity of this site, located on a small island with open ocean immediately around from December to February. This particularity of DDU was confirmed by an absence of summer sodium maximum observed in January (Jourdain and Legrand, 2002) and more recently during summers (2011/2012, and 2012/2013) (Legrand et al., 2016) when exceptionally severe summer sea ice conditions had prevailed at the site. It should be emphasized that this summer sea ice is soft and does not permit emissions of sea-salt aerosol, unlike the case in winter. In contrary, its presence offshore DDU leads to an accumulation

of sea ice pack between the numerous small islands of the Archipelago. This likely limits the effect of breaking waves around the site in emitting locally large particles. References Bréant, C., Leroy Dos Santos, C., Casado, M., Fourré, E., Goursaud, S., Masson-Delmotte, V., Favier, V., Agosta, C., Cattani, O., Prié, F., Golly, B., Orsi, A., and Martinerie, P.: Coastal water vapor isotopic composition driven by katabatic wind variability in summer at Dumont d'Urville, coastal East Antarctica, Earth and Planetary Science Letters, submitted.

Fernandoy, F., Tetzner, D., Meyer, H., Gacitúa, G., Hoffmann, K., Falk, U., Lambert, F., and MacDonell, S.: New insights into the use of stable water isotopes at the northern Antarctic Peninsula as a tool for regional climate studies, The Cryosphere, 2018, 12; 3, p. 1069-1090.

Noone, D., and Simmonds, I.: Sea ice control of water isotope transport to Antarctica and implications for ice core interpretation, Journal of Geophysical Research: Atmospheres, 2004, 109; D7, p. 1-13.

Sinclair, K. E., Bertler, N. A., Bowen, M. M., and Arrigo, K. R.: Twentieth century sea‐ice trends in the Ross Sea from a high‐resolution, coastal ice‐core record, Geophysical Research Letters, 2014, 41; 10, p. 3510-3516.

Stenni, B., Curran, M. A., Abram, N. J., Orsi, A., Goursaud, S., Masson-Delmotte, V., Neukom, R., Goosse, H., Divine, D., and Van Ommen, T.: Antarctic climate variability on regional and continental scales over the last 2000 years, Climate of the Past, 2017, 13; 11, p. 1609-1634.

Thomas, E. R., Bracegirdle, T. J., Turner, J., and Wolff, E. W.: A 308 year record of climate variability in West Antarctica, Geophysical Research Letters, 2013, 40; 20, p. 5492-5496.

Thomas, E. R., van Wessem, J. M., Roberts, J., Isaksson, E., Schlosser, E., Fudge, T. J., Vallelonga, P., Medley, B., Lenaerts, J., and Bertler, N.: Regional Antarctic snow

accumulation over the past 1000 years, Climate of the Past, 2017, 13; 11, p. 1491-1513.

Author's changes in the manuscript We took into consideration your suggestion as well as D. Emanuelsson's review. We also add a data availability paragraph referring to the PANGAEA data archive. Please find below the resulting revised manuscript.  

[revised manuscript text omitted]
.; and rRare dry air masses may also comeing from the western sector, and that thiswith a signal is preserved in d-excess., These different cases we propose here are illustrated by Figure XX. Last pointa is leadled by the following hints: (i) the positive trends both for the TA d-excess and the percentage of air masses coming from the western sectorWAIS+Pacific region (ii) an anti-correlation between the seasonal cycles of the TA d-excess and percentages of air masses coming from the western sectorWAIS+Pacific region (iii) the high simulated d-excess amplitude simulated by ECHAM5-wiso for air masses coming from the western sectorWAIS+Pacific sector, reflecting outstanding values occurring in autumn and winter times (iv) The particular case of the 7th of May in 2007 with very high d-excess values simulated by ECHAM5-wiso, corresponding to an air mass trajectory from the western sectorWAIS+Pacific sector As pointed discussed earlier, the last item should be consideredis however to take with cautionus as the four other remarkable d-excess values (i.e. higher than 30 ‰ simulated by the ECHAM5-wiso are associated to air masses coming from other regions (S613 in the Supplementary Material), and also to the fact that such high values it could be due to potential a numerical artefacts. Linear relationships between d-excessTA and ERA-interim outputs come to 
[revised manuscript text omitted]

Bréant, C., Leroy Dos Santos, C., Casado, M., Fourré, E., Goursaud, S., Masson-Delmotte, V., Favier, V., Agosta, C., Cattani, O., Prié, F., Golly, B., Orsi, A., and Martinerie, P.: Coastal water vapor isotopic composition driven by katabatic wind variability in summer at Dumont d'Urville, coastal East Antarctica, Earth and Planetary Science Letters, submitted.

Bromwich, D. H., and Fogt, R. L.: Strong trends in the skill of the ERA-40 and NCEP–

NCAR reanalyses in the high and midlatitudes of the Southern Hemisphere, 1958–2001, Journal of Climate, 2004, 17; 23, p. 4603-4619.

Bromwich, D. H., Fogt, R. L., Hodges, K. I., and Walsh, J. E.: A tropospheric assessment of the ERA-40, NCEP, and JRA-25 global reanalyses in the polar regions, Journal of Geophysical Research: Atmospheres, 2007, 112; D10, p. 1-21.

Bromwich, D. H., Nicolas, J. P., and Monaghan, A. J.: An assessment of precipitation changes over Antarctica and the Southern Ocean since 1989 in contemporary global reanalyses, Journal of Climate, 2011, 24; 16, p. 4189-4209.

Caiazzo, L., Becagli, S., Frosini, D., Giardi, F., Severi, M., Traversi, R., and Udisti, R.: Spatial and temporal variability of snow chemical composition and accumulation rate at Talos Dome site (East Antarctica), Science of the Total Environment, 2016, 550, p. 418-430.

Casado, M., Landais, A., Masson-Delmotte, V., Genthon, C., Kerstel, E., Kassi, S., Arnaud, L., Picard, G., Prie, F., and Cattani, O.: Continuous measurements of isotopic composition of water vapour on the East Antarctic Plateau, Atmospheric Chemistry and Physics, 2016, 16; 13, p. 8521-8538.

Cavalieri, D., Parkinson, C., Gloersen, P., and Zwally, H.: updated yearly: Sea Ice Concentrations from Nimbus-7 SMMR and DMSP SSM/I-SSMIS Passive Microwave Data, Version 1 [1979-2014]. Boulder, Colorado USA. NASA National Snow and Ice Data Center Distributed Active Archive Center, 1996.

[revised manuscript text omitted]

Wagenbach, D., Ducroz, F., Mulvaney, R., Keck, L., Minikin, A., Legrand, M., Hall, J. S., and Wolff, E. W.: Sea-salt aerosol in coastal Antarctic regions, J. Geophys. Res.-Atmos., 1998, 103; D9, p. 10961-10974.

Werner, M., Langebroek, P. M., Carlsen, T., Herold, M., and Lohmann, G.: Stable water

isotopes in the ECHAM5 general circulation model: Toward high‐resolution isotope modeling on a global scale, Journal of Geophysical Research: Atmospheres, 2011, 116; D15, p.

Yao, T., Petit, J., Jouzel, J., Lorius, C., and Duval, P.: Climatic record from an ice margin area in East Antarctica, Annals of Glaciology, 1990, 14; 1, p. 323-327.

---

## Author Response (AR2)

Dear editor,

Many thanks for having consulted another reviewer for the dating of the ice core. As asked, we added our tests in the supplementary materials, and refer to it in main text, in section 3.1 "Firn core chronology", page 13 lines 13-19: "Peaks in $\delta^{18}O_{TA}$ or d-excess$_{TA}$ were not used in our layer counting, so that our age scale is independent of a climatic interpretation of water stable isotopes (e.g. assumption of synchronicity between temperature seasonal cycles and water stable isotope records). We note an uncertainty in layer counting of 3 years when comparing the outcome of layer counting using chemical records with $\delta^{18}O_{TA}$ peaks, which have nonetheless been excluded from our dating, as they do not improve the correlations, neither between the reconstructed SMB and the stake data, nor between our records and the ECHAM5-wiso simulations (Tables S1 to S3 in the Supplementary Material)."

To the comments made by the third reviewer concerning the chemistry, here is our response:
Calcium and magnesium were measured together with sodium. The level of calcium is of a few ppb (less than 5 ppb) and this species has a double origin (sea-salt and long-range transport of terrigeneous aerosol). Both low level and double origin of calcium render delicate its use to date Antarctic snow and ice (Legrand and Mayewski, 1997). Magnesium is present at higher level than calcium, and under present-day climatic conditions only originates from sea-salt. Its measurement does not provide additional information with respect to sodium. Particle counts and liquid conductivity were not measured here. The powerfulness of particles counts for dating is not obvious (insoluble organic particles coming from the ocean, volcanic glass, long-range transport of terrigeneous aerosol). The liquid conductivity strongly depends on the proton level (related to the presence of acidic compounds like nitrate, non-sea-salt sulfate and MSA) (Legrand, 1980;Legrand and Delmas, 1984) and to weaker extent to sea-salt concentration. Therefore again no additional information can be derived from such a measurement.

We hope that this revised version will be appropriate for publication in The Cryosphere.

Best regards,
Sentia Goursaud

Legrand, M.: Mesure de l'acidité et de la conductivité électrique des précipitations antarctiques, 1980.
Legrand, M., and Delmas, R.: The ionic balance of Antarctic snow: a 10-year detailed record, Atmospheric Environment (1967), 18, 1867-1874, 1984.
Legrand, M., and Mayewski, P.: Glaciochemistry of polar ice cores: a review, Reviews of geophysics, 35, 219-243, 1997.

---

## Author Response (AR3)

Dear editor,

Many thanks for the acceptance of our manuscript for public review and discussion in TC.

Best regards,
Sentia Goursaud